# ANISOTROPIC RANDOM FEATURE REGRESSION IN HIGH DIMENSIONS

**Gabriel C. Mel**
Neurosciences Graduate Program
Stanford University
`meldefon@gmail.com`

**Jeffrey Pennington**
Brain Team, Google Research
`jpennin@google.com`

## ABSTRACT

In contrast to standard statistical wisdom, modern learning algorithms typically find their best performance in the overparameterized regime in which the model has many more parameters than needed to fit the training data. A growing number of recent works have shown that random feature models can offer a detailed theoretical explanation for this unexpected behavior, but typically these analyses have utilized isotropic distributional assumptions on the underlying data generation process, thereby failing to provide a realistic characterization of real-world models that are designed to identify and harness the structure in natural data. In this work, we examine the high-dimensional asymptotics of random feature regression in the presence of structured data, allowing for arbitrary input correlations and arbitrary alignment between the data and the weights of the target function. We define a partial order on the space of weight-data alignments and prove that generalization performance improves in response to stronger alignment. We also clarify several previous observations in the literature by distinguishing the behavior of the sample-wise and parameter-wise learning curves, finding that sample-wise multiple descent can occur at scales dictated by the eigenstructure of the data covariance, but that parameter-wise multiple descent is limited to double descent, although strong anisotropy can induce additional signatures such as wide plateaus and steep cliffs. Finally, these signatures are related to phase transitions in the spectrum of the feature kernel matrix, and unlike the double descent peak, persist even under optimal regularization.

## 1 INTRODUCTION

While deep learning races ahead with its impressive list of practical successes, it has often done so by building larger models and training on ever larger datasets. For example, recent large language models leverage billions (Brown et al., 2020; Raffel et al., 2019) or even trillions (Fedus et al., 2021) of parameters. As the state-of-the-art pushes the computational frontier, model development, experimentation, and exploration become increasingly costly, both in terms of the computational resources and person-hours.

To facilitate the development of improved models, it is becoming ever more necessary to develop theoretical models that can help guide model development in these high-dimensional scenarios. Indeed, our theoretical understanding is so limited that even linear regression, the most basic machine learning method of all, is still yielding surprises in high dimensions (Hastie et al., 2019).

While linear regression can capture many salient features of high-dimensional *data*, it is not a reasonable model for high-dimensional *models*. In particular, there is no natural data-independent notion of model size: the number of parameters in the model is intimately tied (usually equal) to the dimensionality of the datapoints. This limitation means that linear regression may not accurately

capture the effect of overparameterization, which appears to be a crucial ingredient in modern deep learning models (Zhang et al., 2016).

Beyond linear regression, the next simplest model is arguably the random feature model of Rahimi et al. (2007), which turns out to be tractable in high dimensions while also allowing for a natural data-independent lever to vary model size – the number of random features. Random feature models have also gained independent interest through their relationship to neural kernels and Gaussian processes (Neal, 1996; Lee et al., 2017; Novak et al., 2018; Lee et al., 2019; Jacot et al., 2018). While recent works have examined the asymptotic performance of random feature models, most have done so in the restricted setting of isotropic covariates and target functions (Mei & Montanari, 2019; d'Ascoli et al., 2020; Adlam & Pennington, 2020a;b). Based on the conjectured Gaussian equivalence principle of Goldt et al. (2020b), a few recent works have begun relaxing these simplifying assumptions in an effort to describe more natural data distributions (Loureiro et al., 2021; d'Ascoli et al., 2021). In this work, we extend the high-dimensional random feature linearization technique of Tripuraneni et al. (2021a;b) to allow for anisotropic target functions, revealing a host of novel phenomena related to the alignment of the data and the target weights.

## 1.1 CONTRIBUTIONS

Our primary contributions are to:

1. Derive theoretical predictions for the test error, bias, and variance of random feature regression for anisotropic Gaussian covariates under general linear target functions in the high-dimensional asymptotic limit (Section 3.1);

2. Prove that overparameterization reduces the total test error, bias, and variance, even in the presence of anisotropy (Section 3.2);

3. Define a partial order on the space of weight-data alignments and prove that generalization performance improves in response to stronger alignment (Definition 2.1, Section 3.3);

4. Demonstrate that anisotropy can induce sample-wise multiple descent, but prove that parameter-wise multiple descent is limited to double descent (Section 3.4);

5. Provide a theoretical explanation for steep cliffs observed in parameter-wise learning curves that persist under optimal regularization, and relate them to phase transitions in the spectrum of the feature kernel matrix (Section 3.4).

## 1.2 RELATED WORK

Our analysis builds on a series of works that have studied the exact high-dimensional asymptotics of the test error for a growing class of model families and data distributions. Early work in this direction examined minimum-norm interpolated least squares and ridge regression in the high-dimensional random design setting (Belkin et al., 2019; Dobriban et al., 2018; Hastie et al., 2019), finding surprising phenomena such as double descent and a negative optimal ridge constant (Kobak et al., 2020). These analyses were generalized in (Richards et al., 2021; Wu & Xu, 2020; Hastie et al., 2019) to allow for anisotropic covariance matrices and weights that generate the targets. This anisotropic configuration was studied in further detail in (Mel & Ganguli, 2021), uncovering nuanced behavior such as sample-wise multiple descent and a sequence of phase transitions in the performance and optimal regularization.

While linear regression can provide insight into generalization performance in high dimensions, it cannot faithfully reproduce the features of non-linear models, such as the effects of overparameterization. Random feature models, on the other hand, can capture many such features while still maintaining analytical tractability. Recent results have developed a detailed understanding of isotropic random feature regression in the high-dimensional asymptotic limit, providing a precise characterization of peaks in the test error and the benefits of overparameterization (Mei & Montanari, 2019; d'Ascoli et al., 2020; Adlam & Pennington, 2020a). The origin of these peaks was explained by means of a fine-grained decomposition of the bias and variance in (Adlam & Pennington, 2020b; Lin & Dobriban, 2020). While most prior work examining the high-dimensional asymptotics of random feature models have leveraged isotropy of the covariate distribution, this assumption was recently relaxed in (Liao et al., 2020) for random Fourier feature models, in (Tripuraneni et al., 2021a;b) in the

study of covariate shift, and in several works based on the Gaussian equivalence principle of Goldt et al. (2020a;b) (Gerace et al., 2020; Loureiro et al., 2021).

Though most of these works did not pursue a detailed investigation of the anisotropy itself, while finalizing this manuscript we became aware of concurrent work (d'Ascoli et al., 2021) which does focus on the interplay of the anisotropic input features and target functions, as we do here. Whereas d'Ascoli et al. (2021) leverage the Gaussian equivalence principle and the replica method from statistical physics, we use a completely different set of techniques stemming from the literature on random matrix theory and operator-valued free probability (Pennington & Worah, 2018; 2019; Adlam et al., 2019; Adlam & Pennington, 2020a; Louart et al., 2018; Péché et al., 2019; Far et al., 2006; Mingo & Speicher, 2017), which provides a complementary perspective that could be made completely rigorous. d'Ascoli et al. (2021) investigates the role of anisotropy not just under squared loss, as we do here, but also under logistic loss, and obtains good numerical support for the Gaussian equivalence conjecture in that setting. In contrast, our narrower and deeper focus on squared loss allows us not only to uncover a host of novel phenomena, including multiple descent, steep error cliffs, and spectral phase changes, but also to rigorously prove several propositions about their existence and properties. We provide a more in-depth discussion of these and other related works in App. A.

Our work relies heavily on the concept of alignment (see Def. 2.1) between the data covariance and the target vector coefficients. An analogous definition appears in (Tripuraneni et al., 2021a;b) in the context of covariate shift, leading to a superficial similarity of results. However, the implications and interpretations are otherwise quite different, so we refrain from repeatedly commenting on these recurring connections.

## 2 PRELIMINARIES

### 2.1 PROBLEM SETUP AND NOTATION

We study random feature regression (Rahimi et al., 2007) as a model for learning an unknown linear function of the data,

$$y(\mathbf{x}_i) = \beta^\top \mathbf{x}_i / \sqrt{n_0} + \epsilon_i \,, \tag{1}$$

from $m$ independent samples $(\mathbf{x}_i, y_i) \in \mathbb{R}^{n_0} \times \mathbb{R}$, $i = 1, \ldots, m$, where the covariates are Gaussian, $\mathbf{x}_i \sim \mathcal{N}(0, \Sigma)$, and where $\epsilon_i \sim \mathcal{N}(0, \sigma_\epsilon^2)$ is additive noise (present on the training samples only).

We focus on the high-dimensional proportional asymptotics in which the input feature dimension $n_0$, the hidden layer size $n_1$, and the number of samples $m$ all tend to infinity at the same rate, with $\phi := n_0/m$ and $\psi := n_0/n_1$ held constant. The *overparameterization ratio* $\phi/\psi = n_1/m$ is a measure of the normalized complexity of the random feature model.

Interestingly, under this model in our high-dimensional setup, any nonlinear component of the signal to be learned behaves like additive noise, so the linear function in Eq. (1) does not sacrifice generality (Mei & Montanari, 2019; Adlam & Pennington, 2020a).

Following many works on high-dimensional regression (see e.g. Dobriban et al. (2018)), we assume the coefficient vector $\beta$ is random, $\beta \sim \mathcal{N}(0, \Sigma_\beta)$, though allowing for deterministic $\beta$ would be a straightforward extension and would simply involve replacing $\Sigma_\beta \to \beta\beta^\top$ throughout.

Denoting the training dataset as $X = [\mathbf{x}_1, \ldots, \mathbf{x}_m]$ and the test point as $\mathbf{x}$, the random features are,

$$F := \sigma(WX/\sqrt{n_0}) \quad \text{and} \quad f := \sigma(W\mathbf{x}/\sqrt{n_0}) \,, \tag{2}$$

where $W \in \mathbb{R}^{n_1 \times n_0}$ is a random weight matrix with i.i.d. standard Gaussian entries, and $\sigma(\cdot) : \mathbb{R} \to \mathbb{R}$ is an activation function that is applied elementwise. These random features define the kernel

$$K(\mathbf{x}_1, \mathbf{x}_2) := \frac{1}{n_1} \sigma(W\mathbf{x}_1/\sqrt{n_0})^\top \sigma(W\mathbf{x}_2/\sqrt{n_0}) \,, \tag{3}$$

which can be used to compute the model's predictions on a point $x$ as $\hat{y}(\mathbf{x}) = YK^{-1}K_\mathbf{x}$.

Here we have defined the training labels as $Y := [y(\mathbf{x}_1), \ldots, y(\mathbf{x}_m)]$, and the regularized kernel matrix as $K := K(X, X) + \gamma I_m$, as well as $K_\mathbf{x} := K(X, \mathbf{x})$ for $\gamma \geq 0$.

The training error is given by,

$$E_{\text{train}}^\Sigma = \mathbb{E}_\beta \mathbb{E}[(y(X) - \hat{y}(X))^2] = \mathbb{E}[(\beta^\top X/\sqrt{n_0} - YK^{-1}K(X, X))^2] \,, \tag{4}$$

and the test error (without label noise on the test point) can be written as

$$E_\Sigma = \mathbb{E}_\beta \mathbb{E}_\mathbf{x} \mathbb{E}[(y(\mathbf{x}) - \hat{y}(\mathbf{x}))^2] - \sigma_\epsilon^2 = \mathbb{E}_\mathbf{x} \mathbb{E}[(\beta^\top \mathbf{x}/\sqrt{n_0} - YK^{-1}K_\mathbf{x})^2], \tag{5}$$

where the outermost expectation over $\beta$ has been suppressed since the quantity concentrates sharply around its mean and the inner expectation is computed over all the randomness from training, *i.e.* $W$, $X$, and $\epsilon$. The test error can be decomposed into its bias and variance components as

$$E_\Sigma = \underbrace{\mathbb{E}_\mathbf{x} \mathbb{E}[(\mathbb{E}[\hat{y}(\mathbf{x})] - y(\mathbf{x}))^2]}_{B_{\Sigma_\beta}} + \underbrace{\mathbb{E}_\mathbf{x}[\mathbb{V}[\hat{y}(\mathbf{x})]]}_{V_{\Sigma_\beta}}. \tag{6}$$

The bias and variance are computed with respect to all the randomness from training, *i.e.* $W$, $X$, $\epsilon$, which differs from common practice in the statistics literature, but which is necessary to obtain a decomposition that is intuitive and unambiguous when the predictive function relies on randomness from multiple sources (Adlam & Pennington, 2020b; Lin & Dobriban, 2020).

As noted by (Hastie et al., 2019; Wu & Xu, 2020; Mel & Ganguli, 2021), the test error of linear regression depends on the geometry of $(\Sigma, \Sigma_\beta)$ (or $(\Sigma, \beta)$ in the case of nonrandom $\beta$), and we will find the same to be true for random feature regression. As such, we decompose the training and $\beta$ covariance matrices into eigenbases as $\Sigma = \sum_{i=1}^{n_0} \lambda_i \mathbf{v}_i \mathbf{v}_i^\top$ and $\Sigma_\beta = \sum_{i=1}^{n_0} \lambda_i^\beta \mathbf{v}_i^\beta \mathbf{v}_i^{\beta \top}$, where the eigenvalues are in nondecreasing magnitude, *i.e.* $\lambda_1 \le \lambda_2 \le \ldots \le \lambda_{n_0}$ and $\lambda_1^\beta \le \lambda_2^\beta \le \ldots \le \lambda_{n_0}^\beta$. Following (Tripuraneni et al., 2021a;b), we then define the *overlap coefficients* as

$$q_i := \mathbf{v}_i^\top \Sigma_\beta \mathbf{v}_i = \sum_{j=1}^{n_0} (\mathbf{v}_j^\beta \cdot \mathbf{v}_i)^2 \lambda_j^\beta, \tag{7}$$

to measure the alignment of $\Sigma_\beta$ with the $i$th eigendirection of $\Sigma$. Note that for deterministic $\beta$ this definition simply gives the square of the component of $\beta$ in the $i$th eigendirection, i.e. $q_i = (\mathbf{v}_i^\top \beta)^2$.

## 2.2 ASSUMPTIONS

In order to define the limiting behavior of the test error in Eq. (5), it is necessary to impose some regularity conditions on $\Sigma$ and $\Sigma_\beta$.[1] As in (Wu & Xu, 2020; Tripuraneni et al., 2021a;b), the limiting spectra of these matrices cannot be considered independently because their eigenspaces may align. This alignment is most conveniently described in an eigenbasis of $\Sigma$.

**Assumption 1.** *We define the joint spectral distribution (JSD) as*

$$\mu_{n_0} := \frac{1}{n_0} \sum_{i=1}^{n_0} \delta_{(\lambda_i, q_i)} \tag{8}$$

*and assume it converges in distribution to some $\mu$, a distribution on $\mathbb{R}_+^2$ as $n_0 \to \infty$. We refer to $\mu$ as the limiting joint spectral distribution (LJSD), and emphasize that this defines the relevant limiting properties of the covariate and weight distributions[2].*

Often we use $(\lambda, q)$ for random variables sampled jointly from $\mu$ and denote the marginal of $\lambda$ under $\mu$ with $\mu_{\text{data}}$. Since the conditional expectation $\mathbb{E}[q|\lambda]$ is an important object in our study, we assume the following for simplicity.

**Assumption 2.** *$\mu$ is either absolutely continuous or a finite sum of delta masses and the expectations of $\lambda$ and $q$ are finite. Moreover, $\mathbb{E}_\mu[\lambda q] = 1$.*

The normalization condition $\mathbb{E}_\mu[\lambda q] = 1$ is not necessary for our analysis, but, as in (Mel & Ganguli, 2021), it enforces consistent signal-to-noise ratios across different target functions and thereby facilitates meaningful comparisons of the test error.

---

[1]In fact, the necessary assumptions are identical to those of Tripuraneni et al. (2021a;b) since $\Sigma_\beta$ plays an analogous role to the test covariance $\Sigma^*$ of that work.

[2]The JSD depends not only on $\Sigma$ and $\Sigma_\beta$ but also on a choice of eigendecomposition for $\Sigma$ when it has repeated eigenvalues; however, as in (Tripuraneni et al., 2021a;b), all possible choices lead to the same formulas and conclusions.

As we will eventually consider the high-dimensional limit, it is convenient to define the asymptotic covariance scale as $s = \lim_{n_0 \to \infty} \frac{1}{n_0} \text{tr}(\Sigma) = \mathbb{E}_\mu[\lambda]$ under the limiting behavior specified in Assumption 1.

One important case is when $\Sigma_\beta = I_{n_0}$, in which case the LJSD degenerates to $\mu_\emptyset$ defined by

$$\mu_\emptyset(\lambda, q) := \mu_{\text{data}}(\lambda)\delta_1(q). \tag{9}$$

Following Tripuraneni et al. (2021a;b), we also enforce the following standard regularity assumptions on the activation functions to ensure the existence of the moments and derivatives we compute.

**Assumption 3.** *The activation function $\sigma : \mathbb{R} \to \mathbb{R}$ is assumed to be differentiable almost everywhere. We assume there exists a universal constant $C$ such that, $|\sigma(x)|, |\sigma'(x)| = O(\exp(Cx))$.*

### 2.3 A SIMPLE FAMILY OF ANISOTROPIC DISTRIBUTIONS

The assumptions above allow for a wide class of possible asymptotic covariance structures and spectra. To allow for concrete examples that have intuitive interpretations, we introduce the following simple family of distributions that capture multi-scale structure in the input, which we refer to as $d$-scale LJSDs. In particular, for $d \in \mathbb{N}$ and real $0 < \alpha \le 1$ and $\theta$, we define

$$\mu_{\alpha,\theta}^{d\text{-scale}} := \sum_{j=0}^{d-1} \frac{1}{d} \delta_{(C\alpha^j, D\alpha^{\theta j})}, \tag{10}$$

where $C := \left( \frac{1}{d} \frac{1-\alpha^d}{1-\alpha} \right)^{-1}$ and $D := \frac{1}{C} \left( \frac{1}{d} \frac{1-\alpha^{(\theta+1)d}}{1-\alpha^{(\theta+1)}} \right)^{-1}$ enforce the normalization conditions $s = \mathbb{E}_{\mu_{\alpha,\theta}^{d\text{-scale}}}[\lambda] = 1$ and $\mathbb{E}_{\mu_{\alpha,\theta}^{d\text{-scale}}}[q\lambda] = 1$. These simple covariance models capture the hierarchy of scales that often characterize the structure of natural datasets. Note that by setting $\alpha = 1$ and $s = 1$, the distribution reduces to the isotropic case with identity covariance. For the nontrivial setting in which $\alpha < 1$, the data exhibits a sequence of $d$ distinct scales where each scale is a factor $\alpha$ smaller than the previous one. The exponent $\theta$ parameterizes the strength of the weight-data alignment in an intuitive way. When $\theta = 0$, there is no alignment, and $\Sigma_\beta$ is proportional to the identity. When $\theta > 0$, there is strong alignment, as the large eigendirections of the training distribution correspond to large eigendirections of $\Sigma_\beta$. When $\theta < 0$, there is anti-alignment, as the large eigendirections of the training distribution correspond to small eigendirections of $\Sigma_\beta$.

### 2.4 DEFINITION OF THE STRENGTH OF WEIGHT-DATA ALIGNMENT

As the preceding discussion has suggested and as we will see explicitly in the following section, the LJSD $\mu$ captures all of the information about the pair of covariance matrices $(\Sigma, \Sigma_\beta)$ that is relevant for describing the asymptotic test error, bias, and variance. As the alignment between $\Sigma$ and $\Sigma_\beta$ can vary in strength among the different eigendirections, the concept of alignment is inherently multi-dimensional. Nevertheless, by requiring strong alignment along the larger eigendirections, it is possible to define the following natural partial order on the space of possible weight-data alignments[3]

**Definition 2.1.** *Let $\mu_1$ and $\mu_2$ be LJSDs with the same marginal distribution of $\lambda$. If the asymptotic overlap coefficients are such that $\mathbb{E}_{\mu_1}[\lambda q | \lambda] / \mathbb{E}_{\mu_2}[\lambda q | \lambda] = \mathbb{E}_{\mu_1}[q | \lambda] / \mathbb{E}_{\mu_2}[q | \lambda]$ is nondecreasing in $\lambda$, we say that $\mu_1$ is more strongly aligned than $\mu_2$ and write $\mu_1 \le \mu_2$. Comparing against the case of isotropic weight distribution, $\mu_\emptyset$, we say $\mu_1$ is aligned when $\mu_1 \le \mu_\emptyset$ and anti-aligned when $\mu_1 \ge \mu_\emptyset$.*

The parameter $\theta$ of the $d$-scale model in Eq. (10) provides a quantitative measure of alignment under Definition 2.1, formalizing the intuition given in Section 2.3. As we will see in Section 3.3, the asymptotic generalization performance of random feature regression is strictly ordered under this definition of alignment strength.

---

[3]Note that this definition is nearly identical to that of Tripuraneni et al. (2021a;b), with the concept of *hardness* replaced by *alignment*.

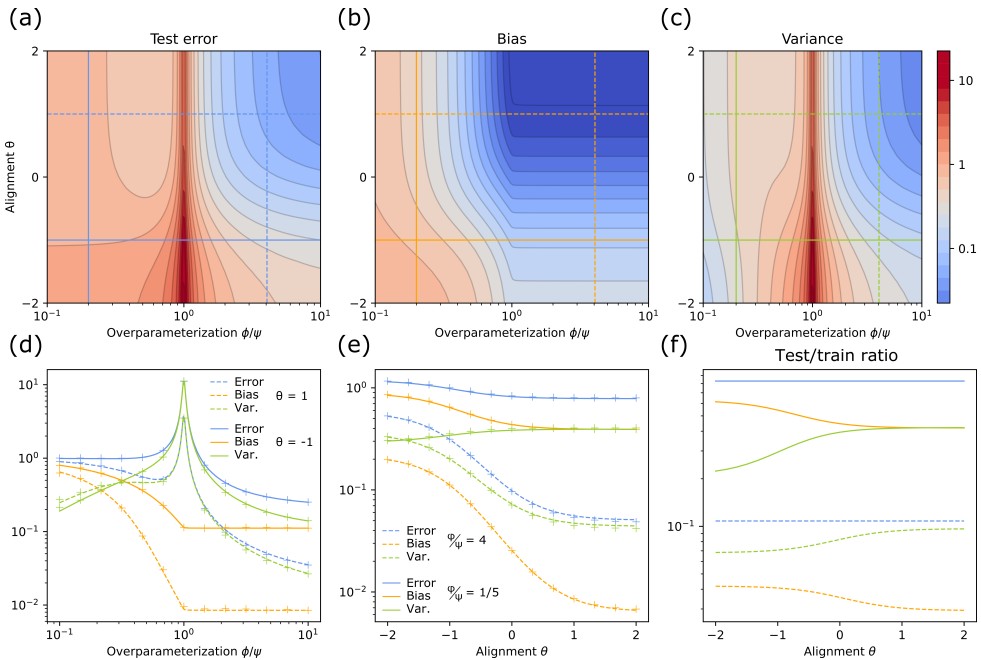

Figure 1: Test error, bias, and variance as a function of the overparameterization ratio ($\phi/\psi = n_1/m$) and the alignment $\theta$ for the 2-scale LJSD (Eq. (10)) with $\phi = n_0/m = 10/9, \sigma = \tanh, \gamma = 10^{-8}$ and $\sigma_\varepsilon^2 = 1/100$. (a) The total test error exhibits the characteristic double descent behavior for all shift powers. (b) The bias is a nonincreasing function of $\phi/\psi$ for all shift powers, as in Proposition 3.1. (c) The variance is the source of the peak, and is a nonincreasing function of $\phi/\psi$ for all shift powers in the overparameterized regime, as in Proposition 3.2. Blue, orange, and green lines in (a,b,c) indicate locations of vertical and horizontal slices shown in (d,e,f). (d) 1D horizontal slices of (a,b,c) more clearly demonstrate the monotonicity in $\phi/\psi$ predicted by Propositions 3.1 and 3.2. (e) 1D vertical slices demonstrate the monotonicity in $\theta$ of the test error and the bias predicted by Proposition 3.3, and illustrate that the variance may not decrease in response to stronger alignment (solid green trace). (f) Test error, bias, and variance normalized by the training error (ie. $\frac{E_\mu + \sigma_\varepsilon^2}{E_{\text{train}}/E_{\text{train}}(\theta=0)}$; similarly for $B_\mu, V_\mu$). Same legend as (e). Constancy of the blue traces illustrates how the train and test error respond identically to alignment. Simulations for $m = 4000$ (d,e; crosses) agree well with formulas.

## 3 MAIN RESULTS

Our main result characterizes the high-dimensional asymptotic limits of the test error, bias, and variance of the nonlinear random feature model of Section 2. Before stating the result, we introduce some additional notation that captures the effect of the nonlinearity $\sigma$,

$$\eta := \mathbb{V}_{z \sim \mathcal{N}(0,s)}[\sigma(z)], \quad \rho := \left(\tfrac{1}{s}\mathbb{E}_{z \sim \mathcal{N}(0,s)}[z\sigma(z)]\right)^2, \quad \zeta := s\rho, \quad \omega := s(\eta/\zeta - 1). \tag{11}$$

where, as above, $s = \lim_{n_0 \to \infty} \frac{1}{n_0}\text{tr}(\Sigma) = \mathbb{E}_\mu[\lambda]$. The constant $\omega \geq 0$ is a measure of the degree of nonlinearity, with $\omega = 0$ corresponding to linear activation functions (see Lemma B.1). Analogously to Tripuraneni et al. (2021a;b), we also introduce two functionals of $\mu$, which capture all the relevant spectral information needed for describing the test loss,

$$\mathcal{I}_{a,b} := \phi\, \mathbb{E}_\mu\left(\lambda^a\,(\phi + x\lambda)^{-b}\right) \quad \text{and} \quad \mathcal{I}_{a,b}^\beta := \phi\, \mathbb{E}_\mu\left(q\lambda^a\,(\phi + x\lambda)^{-b}\right). \tag{12}$$

### 3.1 FORMULAS FOR ASYMPTOTIC BIAS, VARIANCE, AND TEST ERROR

**Theorem 3.1.** *Under Assumptions 1-3, as $n_0, n_1, m \to \infty$ with $\phi = n_0/m$ and $\psi = n_0/n_1$ held constant, the training error $E_{\text{train}}^\Sigma$ tends toward the value of $E_{\text{train}}^\mu$ where*

$$E_{\text{train}}^\mu = -\gamma^2\left(\partial_\gamma(\tau_1\mathcal{I}_{1,1}^\beta) + \sigma_\varepsilon^2\partial_\gamma\tau_1\right), \tag{13}$$

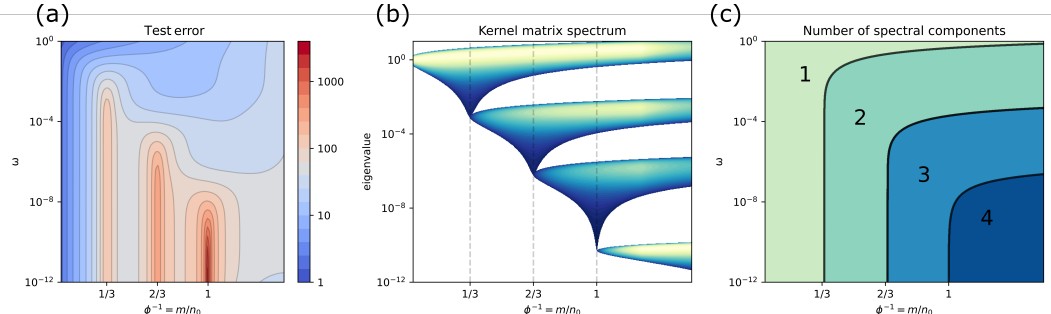

Figure 2: Sample-wise multiple descent for anisotropic data as a function of nonlinearity strength $\omega$. All panels refer to a 3-scale covariance model with $\alpha = 10^3$, and $\psi = \frac{1}{2}, s = \rho = 1, \gamma = 10^{-13}$ and $\sigma_\varepsilon^2 = 10$. (a) Error exhibits multiple peaks as a function of the the sampling density $\phi^{-1}$. As $\omega$ is increased, peaks are attenuated and eventually disappear in sequence starting from those due to the weakest data scales (at $m = n_0$) to those due to the strongest data scales (at $m = n_0/3$). (b) Limiting spectral density of the kernel matrix $K(X, X) = F^T F/n_1$. Vertical slices correspond to the spectral density at the corresponding value of $\phi^{-1}$ for $\omega = 10^{-10}$. Peaks in the error curves in (a) occur at critical values $m = \frac{1}{3}n_0, \frac{2}{3}n_0, n_0$ where a new spectral component first appears. (c) Consistent with this, sample-wise learning curves exhibiting multiple descent at low $\omega$ pass through several transitions in the number of spectral components, while those at higher $\omega$ with fewer peaks pass through fewer component transitions.

*and the bias, variance, and test error $B_{\Sigma_\beta}, V_{\Sigma_\beta}, E_{\Sigma_\beta}$ tend toward $B_\mu, V_\mu, E_\mu$ with*

$$E_\mu = \frac{E_{train}^\mu}{\gamma^2 \tau_1^2} - \sigma_\varepsilon^2, \quad B_\mu = \phi \mathcal{I}_{1,2}^\beta, \quad and \quad V_\mu = E_\mu - B_\mu, \tag{14}$$

*where $\rho, \omega$ are defined in (11), $\mathcal{I}_{a,b}, \mathcal{I}_{a,b}^\beta$ are defined in (12), and $x$ is the unique nonnegative real root of $x = \frac{1-\gamma\tau_1}{\omega + \mathcal{I}_{1,1}}$ with $\tau_1 = \frac{\sqrt{(\psi-\phi)^2 + 4x\psi\phi\gamma/\rho} + \psi - \phi}{2\psi\gamma}$.*

**Remark 3.1.** *As noted in (Adlam & Pennington, 2020a; Hastie et al., 2019) for isotropic covariates, the test error is simply related to the training error through the generalized cross-validation (GCV) formula (Golub et al., 1979). Curiously, because $\tau_1$ is independent of $\beta$, it follows that the test error depends on $\beta$ entirely through its effect on the training error; in contrast, the bias-variance decomposition retains explicit $\beta$ dependence, even conditional on the training error.*

The results in Theorem 3.1 depend on a single scalar self-consistent equation for $x$, $x = \frac{1-\gamma\tau_1}{\omega + \mathcal{I}_{1,1}}$, which is in fact the same as the one appearing in Tripuraneni et al. (2021a;b), and which significantly simplifies the expressions relative to those recently obtained using the replica method (d'Ascoli et al., 2021). Owing to its simplicity, this equation admits straightforward analysis that we pursue in the Appendix, yielding numerous inequalities and bounds that ultimately prove the propositions presented throughout this section. We will occasionally refer to the ridgeless limit of Theorem 3.1, which is given in Corollary G.1. Finally, as a consistency check, taking $\sigma(x) = x$ and $\psi \to 0$ in Theorem 3.1 yields an expression for the test error, bias, and variance for linear regression that agrees with the results of (Wu & Xu, 2020; Mel & Ganguli, 2021) (see Section E).

## 3.2 THE BENEFITS OF OVERPARAMETERIZATION

While there is abundant empirical evidence that overparameterization can improve the generalization performance of practical models (see e.g. (Zhang et al., 2016)), rigorous explanations for this behavior have been offered solely in the setting of isotropic covariates and target functions. It is therefore natural to wonder whether the benefits of overparmeterization persist in the presence of anisotropy, and indeed recent theoretical work has provided numerical evidence hinting that overparameterization is beneficial (d'Ascoli et al., 2021). The following two results prove this to be the case.

First, we show that the bias decreases (or stays constant) in response to an increase in the number of random features, regardless of any anisotropy in the covariates or target function weights.

**Proposition 3.1.** *In the setting of Theorem 3.1, the bias $B_\mu$ is a nonincreasing function of the overparameterization ratio $\phi/\psi$.*

The same is true for the variance in the overparameterized regime. Note that our proof requires the ridgeless setting, but numerical simulations suggest that the result may hold generally (see Fig. 1).

**Proposition 3.2.** *In the ridgeless limit and in the overparameterized regime ($\psi < \phi$), the variance $V_\mu$ is a nonincreasing function of overparameterization ratio $\phi/\psi$.*

In Fig. 1, Propositions 3.1 and 3.2 are illustrated. Moving left to right in panels (a)-(d) leads to models with more parameters. The monotonicity of the bias across the whole range of parameterization is evident, as is the monotonicity of the variance in the overparmaeterized regime.

### 3.3 WEIGHT-DATA ALIGNMENT REDUCES THE BIAS AND TEST ERROR

A number of recent works have confirmed the intuition that generalization performance should improve if the weights of the target function align with the large eigendirections of the training covariance, with formal theoretical arguments in the case of linear regression (Hastie et al., 2019; Mel & Ganguli, 2021), and with informal numerical simulations for random feature models (d'Ascoli et al., 2021). The following result proves that this informal observation holds in full generality, not only for the total test error, but for the bias and the bias-to-variance ratio as well.

**Proposition 3.3.** *Let $\mu_1, \mu_2$ be two LJSDs such that $\mu_1 \leq \mu_2$ (see Definition 2.1). Then $B_{\mu_1} \leq B_{\mu_2}$, $E_{\mu_1} \leq E_{\mu_2}$, and $B_{\mu_1}/V_{\mu_1} \leq B_{\mu_2}/V_{\mu_2}$.*

We illustrate Proposition 3.3 in Fig. 1 for the 2-scale LJSD of Eq. (10). Following vertical lines upward in panels (a-c) or the x-axis left-to-right in (e,f) yields stronger alignment and a corresponding decrease in the bias and test error. Panel (f) shows the alignment-dependence of the bias, variance, and test error when normalized by the training error. As suggested by Remark 3.1, the normalized test error is independent of the alignment strength, and as intuition would suggest and as predicted by Proposition 3.3, the normalized bias decreases in response to stronger alignment.

### 3.4 ANISOTROPY INDUCES STRUCTURED LEARNING CURVES

Strong anisotropy has been observed to induce structured learning curves in the context of linear regression (Mel & Ganguli (2021)). It is natural to wonder whether these effects generalize to the case of nonlinear random features. Figure 2 shows that this is indeed the case for sample-wise learning curves, where $\phi^{-1} = m/n_0$ is varied while $\psi = n_0/n_1$ is held fixed. Horizontal slices through Fig. 2 (a) exhibit sample-wise multiple descent. Fig. 2 (b) and (c) show that these peaks are associated with phase transitions at critical values of $\phi^{-1}$ where the spectrum of the kernel matrix $\frac{1}{n_1}F^T F$ acquires a new component. Increased nonlinearity $\omega$ attenuates this effect.

In contrast, the following proposition shows that parameter-wise multiple descent does not occur even in the presence of strong anisotropy, as long as $\mu$ is *aligned*.

**Proposition 3.4.** *If $\mu$ is* aligned *(see Definition 2.1), then, in the ridgeless limit, the test error has at most two interior critical points as a function of the overparameterization ratio $\phi/\psi$.*

**Remark 3.2.** *Since $k$-fold descent requires at least $k$ critical points, we conclude that multiple ($k > 2$) descent does not occur, even in the presence of covariate anisotropy, so long as $\mu$ is aligned.*

Fig. 3(a,b) shows parameter-wise learning curves for various values of $\theta$ in the 2- and 3-scale models. As predicted by Proposition 3.4, the curves for aligned LJSDs ($\theta \geq 0$) have at most two critical points (not shown). In fact, even for anti-aligned LJSDs ($\theta < 0$), most curves also have at most two critical points; however, for small enough $\theta$, an additional critical point emerges, and indeed the upper-most traces in Fig. 3(b) showcase a second peak in the learning curves.

Even though parameter-wise multiple descent is not possible for aligned LJSDs, strong anisotropy can nevertheless induce other signatures, including wide plateaus and steep cliffs. To understand the origin of these effects, consider the ridgeless limit, for which the self-consistent equation for $x$ is,

$$\omega \frac{x}{\phi} + \mathbb{E}_\mu \frac{\lambda}{\lambda + \frac{\phi}{x}} = \frac{1}{\phi} \min\left(1, \frac{\phi}{\psi}\right) . \tag{15}$$

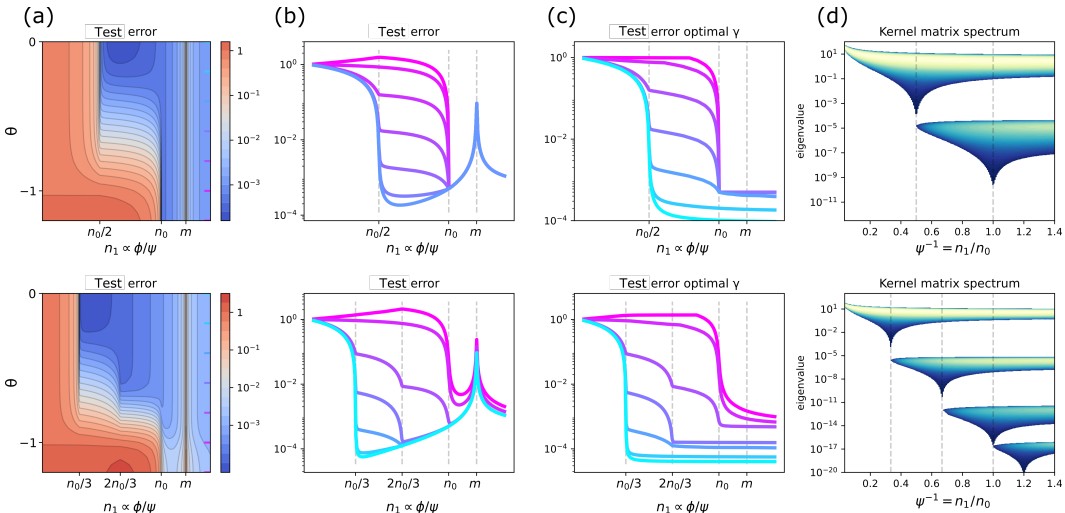

Figure 3: Strong anisotropy causes steep cliffs in the test error as a function of the overparameterization ratio $\phi/\psi = n_1/m$. $\phi = 5/6, \gamma = 10^{-22}, s = \rho = 1, \omega = 10^{-16}$ and $\sigma_\varepsilon^2 = 10^{-4}$. Top row: $d = 2$-scale model with $\alpha = 10^{-5}$; bottom row: 3-scale model with $\alpha = 10^{-6}$. (**a,b**) Test error exhibits the characteristic peak at the interpolation threshold, but additionally displays steep cliffs at critical values of $n_1$ corresponding to multiples of $n_0/d$ (b, dashed lines; $m$ also shown). The $\theta$ value associated to each trace is indicated by the colored ticks in (a). (**c**) Unlike the peak at the interpolation threshold, cliffs in the error traces in (b) persist under optimal regularization (error values from analytical formulas were numerically optimized over $\gamma$). (**d**) The spectrum of the kernel matrix $K(X, X) = \frac{1}{n_1} F^\top F$ undergoes a phase transition at each of the critical values of $n_1$ where a cliff occurs (dashed lines). The phase transition associated to the interpolation threshold at $n_1 = m$ is also visible in the lower panel, but is out of range (at $\approx 10^{-17}$) in the upper panel.

For small $\omega$, the left side becomes relatively insensitive to changes in $x$ whenever $\lambda_- \ll \phi/x \ll \lambda_+$ for some wide spectral gap with boundary $(\lambda_-, \lambda_+)$. It follows that, in the underparameterized regime $(\phi < \psi)$, the derivative $\partial(\phi/\psi)/\partial x$ is small, or, equivalently, $\partial x/\partial(\phi/\psi)$ is large. This strong sensitivity of $x$ to changes in the overparameterization ratio $\phi/\psi$ induces similarly strong changes to the error because $\partial E_\mu/\partial x$ is bounded. As a result, the parameter-wise learning curves exhibit sharp downward cliffs in the underparameterized regime for sufficiently small $\omega$ and sufficiently large spectral gaps $\lambda_+/\lambda_-$. We make this argument more precise in Appendix F.

Fig. 3(a) illustrates these cliffs in the context of the $d = 2$- and 3-scale LJSD. Along horizontal slices, the error turns sharply downward as a function of $n_1 \propto \phi/\psi$ at integer multiples of $n_0/d$. As can be seen in Fig. 3(b), when the alignment is increased and $\beta$ overlaps more with large $\lambda$s, the first error cliff, corresponding to the largest scales in the data, strengthens relative to others (cliff at $\phi/\psi = 1/2$ in the 2-scale model and $\phi/\psi = 1/3$ in the 3-scale model). Fig. 3(c) shows that, unlike the peak at the interpolation threshold, cliffs persist under optimal regularization. Finally, Fig. 3(d) illustrates how steep decreases in the error are associated with the appearance of a new component in the spectrum of the kernel matrix at critical values of $\phi/\psi$.

## 4 CONCLUSIONS

We presented an exact calculation of the limiting test error, bias, and variance for random feature kernel regression with anisotropic covariates and target function weights. We defined a partial order over weight-data alignments and proved that stronger alignment decreases the bias and test error. We also proved that the benefits of overparameterization persist in the anisotropic setting, and that weight-data alignment limits parameter-wise multiple descent to double descent. In contrast, we demonstrated that anisotropy can induce sample-wise multiple descent, and parameter-wise cliffs, and that their structure is dictated by the eigenstructure of the data covariance. Future directions include extending our results to the non-asymptotic regime, accommodating feature learning and more general neural network models, and investigating the impact of anisotropy for other loss functions.

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

# Supplementary Material: Anisotropic Random Feature Regression in High Dimensions

## TABLE OF CONTENTS: SUPPLEMENTARY MATERIAL

## A   ADDITIONAL DISCUSSION OF RELATED WORK

Since preparing the initial version of this manuscript, we became aware of several related recent and concurrent works. Our main contribution relative to these works is the detailed phenomenology explored throughout the main text, but here we provide a more detailed discussion of some of the connections between the current paper and other work, highlighting the main similarities and differences.

### A.1   ASYMPTOTIC ERROR FORMULAS AND THEOREM 3.1

Alternative asymptotic formulas for the test error of anisotropic random feature regression have been presented in a set of concurrent works, yielding overlapping results with Theorem 3.1. The result that is most closely related is that of d'Ascoli et al. (2021), who also study the random feature model with anisotropic input data and target function. They derive formulas for arbitrary convex loss function (generalizing our setup) and then focus on two specialized learning scenarios: (1) linear target function with additive noise and quadratic loss, corresponding to the case studied here; and (2) discrete class labels $\text{sign}(\beta^T x/\sqrt{n_0})$ with label-flipping noise and logistic cost function. Their main results rely on a Gaussian equivalence theorem for anisotropic data, which they derive assuming the Gaussian equivalence principle of Goldt et al. (2020a;b), and then proceed via a standard replica calculation. Another salient paper is that of Loureiro et al. (2021), who study generic feature maps for student-teacher models. A Gaussian covariate model is proposed and rigorous asymptotic solutions are derived for it using Gaussian comparison inequalities, which are shown to agree with calculations from the replica method. The model is general enough to facilitate comparisons to realistic datasets, and numerical evidence and universality arguments support the utility of the model for exactly describing the random feature model we examine here, among many other applications.

Our technical approach proceeds in a substantially different manner, using tools from random matrix theory and operator-valued free probability, rather than statistical physics techniques or the replica method. The results could be made entirely rigorous, though here we simply present the pertinent calculations and defer justification of the underlying linearization techniques to future work and to (Tripuraneni et al., 2021a;b). Our analysis ultimately yields final expressions with a relatively simple form, involving only a single scalar self-consistent equation, which lends itself to more straightforward downstream calculations and analysis (e.g. Propositions 3.1-3.4 and Corollary G.1). Finally, beyond the total error, we also derive formulas for the bias and variance, which aid significantly in the interpretation of the phenomenology, and are novel results. Interestingly, the order parameter $Q$ from d'Ascoli et al. (2021) (and others) is interpreted as the variance of the student's outputs, but actually differs from the variance defined in Eq. (6). The reason is that the bias-variance decomposition is defined *conditionally* on $\mathbf{x}$. Because the conditional mean is nonzero, i.e. $\mathbb{E}[\hat{y}|\mathbf{x}] \neq 0$, $Q$ actually corresponds to an uncentered second moment, and corresponds to our term $E_3$ (defined in Eq. (S147)) which differs from the total variance by the non-trivial additive term $E_4 = \mathbb{E}_{\mathbf{x}}[\mathbb{E}[\hat{y}|\mathbf{x}]^2]$ (defined in Eq. (S162)). A more thorough discussion of these and related concepts is given by Adlam & Pennington (2020b).

### A.2   GAUSSIAN EQUIVALENTS

Our calculations utilize the concept of Gaussian equivalents, in particular a linear-signal-plus-noise surrogate $F^{\text{lin}}$ for the random feature matrix $F$. This approach originates from Karoui (2010) in the context of kernel random matrices of the form $K_{ij} = \sigma(X_i^\top X_j/n_0)$ or $K_{ij} = \sigma(\|X_i - X_j\|^2/n_0)$ and from Pennington & Worah (2019); Adlam et al. (2019); Péché et al. (2019) for the covariance matrices $F^\top F/n_1$ studied here. This linearization technique was further developed by Adlam et al. (2019) for anisotropic covariates (and in the presence of bias), where it was shown to be sufficient for predicting the training error for random feature ridge regression with isotropic linear target functions. In the setting of spherical data and weights, Mei & Montanari (2019) extended these results to cover the test error as well. In this case, many of the main technical results could build directly from Karoui (2010) owing to a decomposition of kernel inner product matrices (Mei & Montanari, 2019, Section C.4) that ultimately relies on the orthogonality of Gegenbauer polynomials and cannot (immediately) be extended to the Gaussian settings studied here and elsewhere. For random feature regression with the neural tangent kernel (which subsumes the standard random feature setting), a proof of the linearization was outlined for the test error in the setting of isototropic Gaussian covariates by Adlam

& Pennington (2020a). As mentioned previously, an extension to anisotropic Gaussian covariates was later developed by Tripuraneni et al. (2021a;b), which is the basis for our analysis in this work.

In a parallel and largely independent line of work stemming from Goldt et al. (2020a), a nearly identical approach is developed under the name of the *Gaussian equivalence property*, under which a possibly nonlinear target function and/or prediction are replaced with simple linear Gaussian equivalents. This is a crucial step in the analysis in Goldt et al. (2020b); Gerace et al. (2020); d'Ascoli et al. (2021) and related work. For example, Gerace et al. (2020) use this principle (in fact, a stronger version they refer to as "replicated Gaussian equivalence") in order to perform their replica analysis of the isotropic random feature model. Subsequently, in the isotropic setting, this principle was rigorously justified using the Lindeburg exchange method under a variety of technical assumptions on the data distribution, weight distributions, nonlinear activation function, and target function (Hu & Lu, 2021). Goldt et al. (2020b) relaxes some of these conditions and provides extensive tests of the resulting formulas on real-world datasets. To perform the analysis for *anisotropic* input data and target function weights, as is pursued in d'Ascoli et al. (2021), an anisotropic extension of the Gaussian equivalence theorem is required. Substantial numerical evidence and theoretical arguments are presented by d'Ascoli et al. (2021); Loureiro et al. (2021) for the validity of this extension, but to the best of our knowledge a rigorous proof in this context has not been established.

### A.3 WEIGHT-DATA ALIGNMENT

One of the basic conclusions of our study of anisotropy is that weight-data alignment generally improves performance. Similar observations appear in several recent works, albeit in slightly different contexts. For example, Ghorbani et al. (2019) study the random feature model with isotropic inputs, but anisotropic *weights*, in the case of a fixed quadratic target function and derives an asymptotic formula for the test error in the population limit (ie. $m \gg n_0, n_1$). For wide networks, $n_1 \gg n_0$, the error simplifies and is exactly proportional to a simple measure of alignment between the random feature weights and the target, that is loosely related to the measure we propose in Definition 2.1.

Ghorbani et al. (2021) also study the random feature model in the population limit, and makes the assumption that the target function is sensitive to a much lower dimensional subspace of the input by positing sub-linear scaling of the dimensionality of the relevant subspace. They show that increasing the power of the input data in this subspace generally decreases test error and the number of random features required to learn a function of fixed complexity. Although the learning contexts and the final scaling limits for $m, n_0, n_1$ are distinct, these phenomena parallel our main result on alignment (see e.g. Fig. 3b for illustration in the context of the $d$-scale model).

A main contribution of the current paper is the partial order on the space of weight-data alignments, which allows us to prove that the total error and the bias decrease in response to stronger alignment (Proposition 3.3). Our results in this vein are most directly related to those of d'Ascoli et al. (2021), who informally observe a basic relationship between weight-data alignment and performance, though the impact of alignment is also investigated elsewhere, e.g. Loureiro et al. (2021, Fig. 2). While these works informally examine concept of alignment, the conclusions about it derive from numerical evaluation of the formulas, and as such the generality of some of the results remains unclear and some of the underlying phenomena are partially obfuscated. For example, it is not clear why the "isotropic" and "misaligned" curves cross each other in of d'Ascoli et al. (2021, Fig. 2c): naively, one might expect the misaligned model to always perform worse. Our results provide a nice perspective on this behavior: owing to the differing covariate distributions, the two forms of alignment are incomparable under the partial order.

## B USEFUL INEQUALITIES

Here we include the statements and proofs of several auxiliary inequalities that we use throughout the Supplementary Material.

### B.1 BASIC PROPERTIES OF THE SELF-CONSISTENT EQUATION FOR $x$

We begin by reviewing the basic inequalities, first given in (Tripuraneni et al., 2021a;b). The definitions of the following quantities can be found in Theorem 3.1.

**Lemma B.1** (Adapted from (Tripuraneni et al., 2021a;b)). *We have the following bounds:* $\omega, \tau_1, \bar{\tau}_1, x, \mathcal{I}_{a,b}, \mathcal{I}_{a,b}^\beta \geq 0$ *and* $\frac{\partial x}{\partial \gamma} \leq 0$.

*Proof.* As shown in (Pennington & Worah, 2018) for the unit-variance case, a simple Hermite expansion argument establishes the relation $\eta \geq \zeta$, which implies $\omega = s(\eta/\zeta - 1) \geq 0$. From Appendix G.4.1, $\tau_1$ and $\bar{\tau}_1$ are traces of positive semi-definite matrices and are therefore nonnegative. From the same equations, it follows that $x = \gamma \rho \tau_1 \bar{\tau}_1 \geq 0$. Nonnegativity of $x$ implies $\mathcal{I}_{a,b} \geq 0$ and $\mathcal{I}_{a,b}^\beta \geq 0$ from their definitions in (12). Finally, using the nonnegativity of $\omega$, $\tau_1$, $\bar{\tau}_1$, $x$, and $\mathcal{I}_{a,b}$, the expression for $\frac{\partial x}{\partial \gamma}$ in Theorem 3.1 immediately gives,

$$\frac{\partial x}{\partial \gamma} = -\frac{x}{\gamma + \rho\gamma(\frac{\psi}{\phi}\tau_1 + \bar{\tau}_1)(\omega + \phi\mathcal{I}_{1,2})} \leq 0. \tag{S1}$$

$\square$

Next we show that the self-consistent equation $x = \frac{1-\gamma\tau_1}{\omega+\mathcal{I}_{1,1}}$ appearing in Theorem 3.1 and defined in (S237) admits a unique positive real solution for $x$.

**Lemma B.2** (Adapted from (Tripuraneni et al., 2021a;b)). *There is a unique real $x \geq 0$ satisfying* $x = \frac{1-\gamma\tau_1}{\omega+\mathcal{I}_{1,1}}$.

*Proof.* Let $t = 1/x \geq 0$ and define,

$$h(t) = t\left(\frac{\rho(\psi-\phi) + \sqrt{\rho^2(\psi-\phi)^2 + 4\gamma\rho\phi\psi/t}}{2\rho\psi} - 1\right) + \omega + \mathcal{I}_{1,1}(1/t), \tag{S2}$$

which is a rewriting of eqn. (S237), so it suffices to show that $h$ admits a unique real positive root. To that end, first observe that $\lim_{t\to 0} \mathcal{I}_{1,1}(1/t) = 0$ and $\lim_{t\to\infty} \mathcal{I}_{1,1}(1/t) = s$ so that

$$h(0) = \omega > 0 \quad \text{and} \quad \lim_{t\to\infty} h(t)/t = -\min\{1, \phi/\psi\} < 0, \tag{S3}$$

which together imply that $h$ has an odd number of positive real roots. Next, we show that $h$ is concave for $t \geq 0$:

$$h''(t) = -\frac{2\phi}{t^3}\left(\frac{\gamma^2\rho\phi\psi}{(\rho^2(\psi-\phi)^2 + 4\gamma\rho\phi\psi/t)^{3/2}} + \mathcal{I}_{2,3}(1/t)\right) \tag{S4}$$

$$\leq 0, \tag{S5}$$

which implies that $h$ has at most two positive real roots. Therefore, we conclude that $h$ has exactly one positive real root. To provide a bounding interval for this root, we first observe that,

$$\lim_{t\to\infty} h(t) - \left(-\min\{1, \phi/\psi\}t + \omega + s + \frac{\gamma\phi}{\rho|\psi-\phi|}\right) = 0, \tag{S6}$$

so that $h(t)$ can be upper- and lower-bounded by linear functions ,

$$\omega - \min\{1, \phi/\psi\}t \leq h(t) \leq \omega + s + \frac{\gamma\phi}{\rho|\psi-\phi|} - \min\{1, \phi/\psi\}t. \tag{S7}$$

The roots of these linear functions bound the root of $h$, so we have

$$\frac{\min\{1, \phi/\psi\}}{\frac{\gamma\phi}{\rho|\psi-\phi|} + \omega + s} \leq x \leq \frac{\min\{1, \phi/\psi\}}{\omega}. \tag{S8}$$

$\square$

## B.2 $\mathcal{I}$ AND $\mathcal{I}^\beta$ INEQUALITIES

We now establish some useful properties of the $\mathcal{I}$ and $\mathcal{I}^\beta$ functionals defined in (12). To begin, we note that simple algebraic manipulations establish the following raising and lowering identities:

$$\mathcal{I}_{a-1,b-1} = \phi\mathcal{I}_{a-1,b} + x\mathcal{I}_{a,b} \quad \text{and} \quad \mathcal{I}^\beta_{a-1,b-1} = \phi\mathcal{I}^\beta_{a-1,b} + x\mathcal{I}^\beta_{a,b}. \tag{S9}$$

Next, we consider how the partial order of LJSDs given in Definition 2.1 leads to inequalities on the $\mathcal{I}^\beta$ functionals. Letting $(\mathcal{I}^\beta_{a,b})_1$ and $(\mathcal{I}^\beta_{a,b})_2$ to denote the corresponding functionals with the LJSDs $\mu_1$ and $\mu_2$ respectively, we can establish the following useful lemma.

**Lemma B.3** (Adapted from (Tripuraneni et al., 2021a;b)). *Let $\mu_1 \leq \mu_2$, so $\mu_1$ is more strongly aligned than $\mu_2$ (recall Definition 2.1). Suppose the functions $f, g, h : \mathbb{R} \to \mathbb{R}$ are such that $f(\lambda) = g(\lambda)h(\lambda)$ and $h(\lambda)$ is nonincreasing for all $\lambda > 0$, then*

$$\frac{\mathbb{E}_{\mu_1}[qf(\lambda)]}{\mathbb{E}_{\mu_2}[qf(\lambda)]} \leq \frac{\mathbb{E}_{\mu_1}[qg(\lambda)]}{\mathbb{E}_{\mu_2}[qg(\lambda)]}. \tag{S10}$$

*Proof.* By the law of iterated expectation, we have

$$\mathbb{E}_{\mu_1}[qf(\lambda)] = \mathbb{E}_{\mu_2}[qg(\lambda)]\mathbb{E}_\lambda\left[\frac{\mathbb{E}_{\mu_2}[qg(\lambda)|\lambda]}{\mathbb{E}_{\mu_2}[qg(\lambda)]}\frac{\mathbb{E}_{\mu_1}[q|\lambda]}{\mathbb{E}_{\mu_2}[q|\lambda]}h(\lambda)\right]. \tag{S11}$$

Note that the expectation $\mathbb{E}_\lambda$ in (S11) over $\lambda$ is the same under $\mu_1$ and $\mu_2$ by assumption. Moreover, the function $h(\lambda)$ is nonincreasing in $\lambda$ by assumption. Finally, observe that the factor $\mathbb{E}_{\mu_2}[rg(\lambda)|\lambda]/\mathbb{E}_{\mu_2}[rg(\lambda)]$ defines a change in distribution for the random variable $\lambda$, since taking its expectation over $\lambda$ yields 1. Denote a new random with this distribution by $\tilde{\lambda}$. Then, we may apply the Harris inequality to see

$$\mathbb{E}_{\mu_1}[qf(\lambda)] = \mathbb{E}_{\mu_2}[qg(\lambda)]\mathbb{E}_{\tilde{\lambda}}\left[\frac{\mathbb{E}_{\mu_1}[q|\tilde{\lambda}]}{\mathbb{E}_{\mu_2}[q|\tilde{\lambda}]}h(\tilde{\lambda})\right] \tag{S12}$$

$$\leq \mathbb{E}_{\mu_2}[qg(\lambda)]\mathbb{E}_{\tilde{\lambda}}\left[\frac{\mathbb{E}_{\mu_1}[q|\tilde{\lambda}]}{\mathbb{E}_{\mu_2}[q|\tilde{\lambda}]}\right]\mathbb{E}_{\tilde{\lambda}}\left[h(\tilde{\lambda})\right] \tag{S13}$$

$$\leq \mathbb{E}_{\mu_2}[qg(\lambda)]\mathbb{E}_\lambda\left[\frac{\mathbb{E}_{\mu_2}[qg(\lambda)|\lambda]}{\mathbb{E}_{\mu_2}[qg(\lambda)]}\frac{\mathbb{E}_{\mu_1}[q|\lambda]}{\mathbb{E}_{\mu_2}[q|\lambda]}\right]\mathbb{E}_\lambda\left[\frac{\mathbb{E}_{\mu_2}[qg(\lambda)|\lambda]}{\mathbb{E}_{\mu_2}[qg(\lambda)]}h(\lambda)\right] \tag{S14}$$

$$= \frac{\mathbb{E}_{\mu_1}[qg(\lambda)]}{\mathbb{E}_{\mu_2}[qg(\lambda)]}\mathbb{E}_{\mu_2}[qf(\lambda)]. \tag{S15}$$

$\square$

**Corollary B.1.** *Let $\mu_1 \leq \mu_2$ and $(\mathcal{I}^\beta_{a,b})_i := \phi\,\mathbb{E}_{\mu_i}\left(q\lambda^a\,(\phi + x\lambda)^{-b}\right)$. Then, for $a \leq 1$ and $b \geq 0$,*

$$\frac{1}{\mathbb{E}_{\mu_2}[\lambda q]}\left(\mathcal{I}^\beta_{a,b}\right)_2 - \frac{1}{\mathbb{E}_{\mu_1}[\lambda q]}\left(\mathcal{I}^\beta_{a,b}\right)_1 \geq 0. \tag{S16}$$

*Proof.* Note that $h : \lambda \mapsto \phi\lambda^{a-1}(\phi + x\lambda)^{-b}$ is a nonincreasing function of $\lambda \geq 0$. Then, setting $g = \lambda$ and $f = gh$ in Lemma B.3 gives the desired result. $\square$

**Lemma B.4.** *Suppose the functions $f, g, h : \mathbb{R} \to \mathbb{R}$ are such that $f(\lambda) = \lambda g(\lambda)h(\lambda)$ and $h(\lambda)$ is nonincreasing for all $\lambda > 0$. Then, if the LJSD $\mu$ is* aligned *(see Definition 2.1), then $\mathbb{E}_\lambda[g]\mathbb{E}_\mu[qf] \leq \mathbb{E}_\mu[qg]\mathbb{E}_\lambda[f]$.*

*Proof.*

$$\mathbb{E}_\mu[qf] = \mathbb{E}_\mu[q\lambda gh] \tag{S17}$$

$$= \mathbb{E}_\lambda[\mathbb{E}_\mu[q\lambda|\lambda]g(\lambda)h(\lambda)] \tag{S18}$$

$$= \mathbb{E}_\mu[g]\mathbb{E}_\lambda\left[\mathbb{E}_\mu[q\lambda|\lambda]h(\lambda)\frac{g(\lambda)}{\mathbb{E}_\mu[g]}\right] \tag{S19}$$

$$\leq \mathbb{E}_\mu[g]\mathbb{E}_\lambda\left[\mathbb{E}_\mu[q\lambda|\lambda]\frac{g(\lambda)}{\mathbb{E}_\mu[g]}\right]\mathbb{E}_\lambda\left[h(\lambda)\frac{g(\lambda)}{\mathbb{E}_\mu[g]}\right] \tag{S20}$$

$$= \frac{1}{\mathbb{E}_\lambda[g]}\mathbb{E}_\mu[q\lambda g]\mathbb{E}_\lambda[f], \tag{S21}$$

where $\mathbb{E}_\mu[q\lambda|\lambda]$ is nondecreasing in $\lambda$ because $\mu$ is aligned, so the inequality again follows from the Harris inequality. $\square$

**Corollary B.2.** *If $\mu$ is aligned, $\mathcal{I}_{a,b}\mathcal{I}_{a,b}^\beta \leq \mathcal{I}_{a-1,b}\mathcal{I}_{a+1,b}^\beta$.*

*Proof.* Take $g : \lambda \to \phi\lambda^a(\phi + x\lambda)^{-b}$ and $h : \lambda \to 1/\lambda$ in Lemma B.4. $\square$

## C  WEIGHT-DATA ALIGNMENT IS A PARTIAL ORDER

We restate Definition 2.1 for reference, and prove that it defines a partial order. The definition and proof are identical to those of Tripuraneni et al. (2021a;b), but differ in notation so we repeat them here for clarity.

**Definition C.1** (Restatement of Definition 2.1). *Let $\mu_1$ and $\mu_2$ be LJSDs with the same marginal distribution of $\lambda$. If the asymptotic overlap coefficients are such that $\mathbb{E}_{\mu_1}[\lambda q|\lambda]/\mathbb{E}_{\mu_2}[\lambda q|\lambda] = \mathbb{E}_{\mu_1}[q|\lambda]/\mathbb{E}_{\mu_2}[q|\lambda]$ is nondecreasing in $\lambda$, we say that $\mu_1$ is more strongly aligned than $\mu_2$ and write $\mu_1 \leq \mu_2$. Comparing against the case of isotropic weight distribution, $\mu_\emptyset$, we say $\mu_1$ is* aligned *when $\mu_1 \leq \mu_\emptyset$ and* anti-aligned *when $\mu_1 \geq \mu_\emptyset$.*

**Proposition C.1.** *Definition 2.1 is a partial order over over weight-data alignments $\mu$.*

*Proof.* Reflexivity is satisfied as $\mathbb{E}_\mu[q|\lambda]/\mathbb{E}_\mu[q|\lambda] = 1$ is nondecreasing for all $\mu$.

For antisymmetry, we see $\mu_1 \leq \mu_2$ and $\mu_2 \leq \mu_1$ imply $\mathbb{E}_{\mu_1}[q|\lambda]/\mathbb{E}_{\mu_2}[q|\lambda]$ is constant in $\lambda$ as it is nonincreasing and nondecreasing. However, setting $\mathbb{E}_{\mu_1}[q|\lambda] = c\mathbb{E}_{\mu_2}[q|\lambda]$ and taking expectation over $\lambda$ and rearranging yields $1 = \mathbb{E}_{\mu_1}[q]/\mathbb{E}_{\mu_2}[q] = c$, so in fact $\mathbb{E}_{\mu_1}[q|\lambda] = \mathbb{E}_{\mu_2}[q|\lambda]$. Assuming that $\mu_1$ and $\mu_2$ are absolutely continuous (the case where they are a sum of point masses is similar), we can write their densities as $p_i(\lambda, q) = p_i(\lambda)p_i(q|\lambda)$. By assumption $p_1(\lambda) = p_2(\lambda)$, so it suffices to show $p_1(q|\lambda) = p_2(q|\lambda)$ almost everywhere. Next note

$$0 = \mathbb{E}_{\mu_1}[q|\lambda] - \mathbb{E}_{\mu_2}[q|\lambda] = \int_{\mathbb{R}^+} q\left(p_1(q|\lambda) - p_2(q|\lambda)\right)\mathrm{d}q, \tag{S22}$$

we have that $p_1(q|\lambda) - p_2(q|\lambda) = 0$ almost everywhere.

Finally, for transitivity assume $\mu_1 \leq \mu_2$ and $\mu_2 \leq \mu_3$, then

$$\frac{\mathbb{E}_{\mu_1}[q|\lambda]}{\mathbb{E}_{\mu_3}[q|\lambda]} = \frac{\mathbb{E}_{\mu_1}[q|\lambda]}{\mathbb{E}_{\mu_2}[q|\lambda]} \cdot \frac{\mathbb{E}_{\mu_2}[q|\lambda]}{\mathbb{E}_{\mu_3}[q|\lambda]}, \tag{S23}$$

so $\mathbb{E}_{\mu_1}[q|\lambda]/\mathbb{E}_{\mu_3}[q|\lambda]$ is the product of two nondecreasing, positive functions and is thus also nondecreasing. $\square$

## D  PROOFS OF PROPOSITIONS

### D.1  PROPOSITION 3.1

**Proposition D.1** (Restatement of Proposition 3.1). *In the setting of Theorem 3.1, the bias $B_\mu$ is a nonincreasing function of overparameterization ratio $\phi/\psi$.*

*Proof.* Recall from Theorem 3.1 that the bias is given by

$$B_\mu = \phi \mathcal{I}_{1,2}^\beta \,, \tag{S24}$$

where $x$ is the unique positive real root of the self-consistent equation,

$$x = \frac{1 - \gamma \tau_1}{\omega + \mathcal{I}_{1,1}} \,. \tag{S25}$$

Differentiating (S24) with respect to $\phi/\psi$ gives,

$$\frac{\partial B_\mu}{\partial (\phi/\psi)} = -\frac{\psi^2}{\phi} \frac{\partial B_\mu}{\partial \psi} = 2\psi^2 \frac{\partial x}{\partial \psi} \mathcal{I}_{1,3}^\beta \,. \tag{S26}$$

Since Lemma B.1 gives $\mathcal{I}_{a,b}^\beta \geq 0$, it suffices to show $\frac{\partial x}{\partial \psi} \leq 0$, which immediately follows by implicitly differentiating (S25)) and simplifying the expression,

$$\frac{\partial x}{\partial \psi} = -\frac{\rho x \tau_1 (\omega + \mathcal{I}_{1,1})}{\phi \big( 1 + \rho(\bar{\tau}_1 + \frac{\psi}{\phi} \tau_1)(\omega + \phi \mathcal{I}_{1,2}) \big)} \leq 0 \,, \tag{S27}$$

where the inequality also follows from Lemma B.1. Therefore we conclude that $\frac{\partial B_\mu}{\partial (\phi/\psi)} \leq 0$. $\qquad\square$

## D.2  PROPOSITION 3.2

**Proposition D.2** (Restatement of Proposition 3.2). *In the setting of Corollary G.1 and in the overparameterized regime ($\psi < \phi$), the variance $V_\mu$ is a nonincreasing function of overparameterization ratio $\phi/\psi$.*

*Proof.* In the overparameterized regime, Corollary G.1 gives the expression for the variance as,

$$V_\mu = \frac{\psi}{\phi - \psi}(\sigma_\varepsilon^2 + \mathcal{I}_{1,1}^\beta) + \frac{x \mathcal{I}_{2,2}}{\omega + \phi \mathcal{I}_{1,2}}(\sigma_\varepsilon^2 + \mathcal{I}_{1,2}^\beta) \,, \tag{S28}$$

and, since the self-consistent equation $x = \frac{1}{\omega + \mathcal{I}_{1,1}}$ is independent of $\psi$, we have $\frac{\partial x}{\partial \psi} = 0$ and,

$$\frac{\partial V_\mu}{\partial \psi} = \frac{\phi}{(\phi - \psi)^2}(\sigma_\varepsilon^2 + \mathcal{I}_{1,1}^\beta) \geq 0 \,, \tag{S29}$$

which implies that the variance is nonincreasing in the overparameterized regime. $\qquad\square$

## D.3  PROPOSITION 3.3

**Proposition D.3** (Restatement of Proposition 3.3). *Let $\mu_1, \mu_2$ be two LJSDs such that $\mu_1 \leq \mu_2$ (see Definition 2.1). Then $B_{\mu_1} \leq B_{\mu_2}$, $E_{\mu_1} \leq E_{\mu_2}$, and $B_{\mu_1}/V_{\mu_1} \leq B_{\mu_2}/V_{\mu_2}$.*

*Proof.* For the bias, Corollary B.1 implies $(\mathcal{I}_{1,2}^\beta)_1 \leq (\mathcal{I}_{1,2}^\beta)_2$ and therefore $B_{\mu_1} \leq B_{\mu_2}$.

For the test error, we use the explicit expression for the variance from Eq. (S378) and the identity $\mathcal{I}_{2,2}^\beta = \frac{1}{x} \mathcal{I}_{1,1}^\beta - \frac{\phi}{x} \mathcal{I}_{1,2}^\beta$ which follows from Eq. (S9) to write,

$$E_\mu = C_0 + C_1 \mathcal{I}_{1,1}^\beta + C_2 \mathcal{I}_{1,2}^\beta \,, \tag{S30}$$

where the $C_i \geq 0$ and depend on $\mu$ only through the marginal $\lambda$ (i.e. they are independent of the weight distribution):

$$C_0 = -\rho \frac{\psi}{\phi} \frac{\partial x}{\partial \gamma} \sigma_\varepsilon^2 \left( (\omega + \phi \mathcal{I}_{1,2})(\omega + \mathcal{I}_{1,1}) + \frac{\phi}{\psi} \gamma \bar{\tau}_1 \mathcal{I}_{2,2} \right) \geq 0 \tag{S31}$$

$$C_1 = -\rho \frac{\psi}{\phi} \frac{\partial x}{\partial \gamma} \left( (\omega + \phi \mathcal{I}_{1,2})(\omega + \mathcal{I}_{1,1}) + \frac{\gamma \tau_1}{x} (\omega + \phi \mathcal{I}_{1,2}) \right) \geq 0 \tag{S32}$$

$$C_2 = \phi - \rho \frac{\psi}{\phi} \frac{\partial x}{\partial \gamma} \left( \frac{\phi^2}{\psi} \gamma \bar{\tau}_1 \mathcal{I}_{2,2} - \frac{\phi \gamma \tau_1}{x} (\omega + \phi \mathcal{I}_{1,2}) \right) \tag{S33}$$

$$= -\rho \frac{\psi}{\phi} \frac{\partial x}{\partial \gamma} \left( \frac{\phi^2}{\psi} \gamma \bar{\tau}_1 \mathcal{I}_{2,2} - \frac{\phi \gamma \tau_1}{x} (\omega + \phi \mathcal{I}_{1,2}) - \frac{\phi^2}{\rho \psi \frac{\partial x}{\partial \gamma}} \right) \tag{S34}$$

$$= -\rho \gamma \frac{\partial x}{\partial \gamma} \left( \phi \bar{\tau}_1 \mathcal{I}_{2,2} - \frac{\psi \tau_1}{x} (\omega + \phi \mathcal{I}_{1,2}) + \frac{\phi}{\rho x} (1 + \rho(\tau_1 \psi / \phi + \bar{\tau}_1)(\omega + \phi \mathcal{I}_{1,2})) \right) \tag{S35}$$

$$= -\rho \gamma \frac{\partial x}{\partial \gamma} \left( \phi \bar{\tau}_1 \mathcal{I}_{2,2} + \frac{\phi}{\rho x} (1 + \rho \bar{\tau}_1 (\omega + \phi \mathcal{I}_{1,2})) \right) \tag{S36}$$

$$\geq 0 \,. \tag{S37}$$

It is now straightforward to write,

$$E_{\mu_2} - E_{\mu_1} = C_1 (\mathcal{I}_{1,1}^\beta)_2 + C_2 (\mathcal{I}_{1,2}^\beta)_2 - C_1 (\mathcal{I}_{1,1}^\beta)_1 + C_2 (\mathcal{I}_{1,2}^\beta)_1 \tag{S38}$$

$$= C_1 \left( (\mathcal{I}_{1,1}^\beta)_2 - (\mathcal{I}_{1,1}^\beta)_1 \right) + C_2 \left( (\mathcal{I}_{1,2}^\beta)_2 - (\mathcal{I}_{1,2}^\beta)_1 \right) \tag{S39}$$

$$\geq 0 \,, \tag{S40}$$

where the inequality follows from Corollary B.1. Similarly, we can write,

$$\frac{B_{\mu_1}}{B_{\mu_2}} E_{\mu_2} - E_{\mu_1} = C_0 \frac{(\mathcal{I}_{1,2}^\beta)_1}{(\mathcal{I}_{1,2}^\beta)_2} + C_1 \frac{(\mathcal{I}_{1,2}^\beta)_1}{(\mathcal{I}_{1,2}^\beta)_2} (\mathcal{I}_{1,1}^\beta)_2 + C_2 (\mathcal{I}_{1,2}^\beta)_1 - C_0 - C_1 (\mathcal{I}_{1,1}^\beta)_1 - C_2 (\mathcal{I}_{1,2}^\beta)_1 \tag{S41}$$

$$= C_0 \left( \frac{(\mathcal{I}_{1,2}^\beta)_1}{(\mathcal{I}_{1,2}^\beta)_2} - 1 \right) + C_1 \left( \frac{(\mathcal{I}_{1,2}^\beta)_1}{(\mathcal{I}_{1,2}^\beta)_2} (\mathcal{I}_{1,1}^\beta)_2 - (\mathcal{I}_{1,1}^\beta)_1 \right) \tag{S42}$$

$$\leq 0 \,, \tag{S43}$$

where the inequality follows from Corollary B.1 and from Lemma B.3 with $g : \lambda \to \phi \lambda (\phi + \lambda x)^{-1}$ and $h : \lambda \to (\phi + \lambda x)^{-1}$. Finally, using $E_{\mu_i} = B_{\mu_i} + V_{\mu_i}$, the above implies $B_{\mu_1}/V_{\mu_1} \leq B_{\mu_2}/V_{\mu_2}$. $\square$

### D.4 PROPOSITION 3.4

**Proposition D.4** (Restatement of Proposition 3.4). *If the LJSD is* aligned *(see Definition 2.1), then, in the setting of Corollary G.1, the test error has at most two interior critical points as a function of the overparameterization ratio $\phi/\psi$.*

*Proof.* From Corollary G.1, there is a critical point at the interpolation threshold $\phi/\psi = 1$. Therefore it suffices to show that there is at most one additional interior critical point. Focusing first on the overparameterized regime $\phi > \psi$, the test error reads,

$$E_\mu = \phi \mathcal{I}_{1,2}^\beta + \frac{\psi}{\phi - \psi} (\sigma_\varepsilon^2 + \mathcal{I}_{1,1}^\beta) + \frac{x \mathcal{I}_{2,2}}{\omega + \phi \mathcal{I}_{1,2}} (\sigma_\varepsilon^2 + \mathcal{I}_{1,2}^\beta) \,, \tag{S44}$$

and, since $\frac{\partial x}{\partial \psi} = 0$,

$$\frac{\partial E}{\partial \psi} = \frac{\phi}{(\phi - \psi)^2} (\sigma_\varepsilon^2 + \mathcal{I}_{1,1}^\beta) > 0 \,, \tag{S45}$$

which implies that the test error is monotone decreasing in the overparameterized regime.

Next, let us consider the case $\phi < \psi$. In this case,

$$E_\mu = \phi \mathcal{I}_{1,2}^\beta + \frac{\phi}{\psi - \phi}(\sigma_\varepsilon^2 + \mathcal{I}_{1,1}^\beta) + x \mathcal{I}_{2,2}^\beta \,, \tag{S46}$$

so that,

$$\frac{\partial E_\mu}{\partial \psi} = \phi \frac{\partial x}{\partial \psi} \frac{\partial}{\partial x} \mathcal{I}_{1,2}^\beta - \frac{\phi}{(\psi - \phi)^2}(\sigma_\varepsilon^2 + \mathcal{I}_{1,1}^\beta) + \frac{\phi}{\psi - \phi} \frac{\partial x}{\partial \psi} \frac{\partial}{\partial x} \mathcal{I}_{1,1}^\beta + \frac{\partial x}{\partial \psi} \frac{\partial}{\partial x}(x \mathcal{I}_{2,2}^\beta) \tag{S47}$$

$$= -\frac{\phi}{(\psi - \phi)^2}(\sigma_\varepsilon^2 + \mathcal{I}_{1,1}^\beta) + \frac{\partial x}{\partial \psi}\left(\phi \frac{\partial}{\partial x} \mathcal{I}_{1,2}^\beta + \frac{\phi}{\psi - \phi} \frac{\partial}{\partial x} \mathcal{I}_{1,1}^\beta + \frac{\partial}{\partial x}(x \mathcal{I}_{2,2}^\beta)\right) \tag{S48}$$

$$= -\frac{\phi}{(\psi - \phi)^2}(\sigma_\varepsilon^2 + \mathcal{I}_{1,1}^\beta) + \frac{\partial x}{\partial \psi}\left(-2\phi \mathcal{I}_{2,3}^\beta - \frac{\phi}{\psi - \phi} \mathcal{I}_{2,2}^\beta + \mathcal{I}_{2,2}^\beta - 2x \mathcal{I}_{3,3}^\beta\right) \tag{S49}$$

$$= -\frac{\phi}{(\psi - \phi)^2}(\sigma_\varepsilon^2 + \mathcal{I}_{1,1}^\beta) - \frac{\partial x}{\partial \psi} \frac{\psi}{\psi - \phi} \mathcal{I}_{2,2}^\beta \tag{S50}$$

$$= -\frac{\phi}{(\psi - \phi)^2}(\sigma_\varepsilon^2 + \mathcal{I}_{1,1}^\beta) + \frac{\phi}{\psi(\psi - \phi)} \frac{\mathcal{I}_{2,2}^\beta}{\omega + \phi \mathcal{I}_{1,2}} \,. \tag{S51}$$

Therefore we see that $\frac{\partial E}{\partial \psi} = 0$ implies

$$\frac{\phi}{\psi} = x(\omega + \mathcal{I}_{1,1}) = 1 - (\omega + \phi \mathcal{I}_{1,2})\frac{\sigma_\varepsilon^2 + \mathcal{I}_{1,1}^\beta}{\mathcal{I}_{2,2}^\beta} \,, \tag{S52}$$

or, equivalently, $g(x) = 0$ for

$$g(x) = 1 - (\omega + \phi \mathcal{I}_{1,2})\frac{\sigma_\varepsilon^2 + \mathcal{I}_{1,1}^\beta}{\mathcal{I}_{2,2}^\beta} - x(\omega + \mathcal{I}_{1,1}) \,. \tag{S53}$$

First we note that $g$ has at most one real root since its derivative is never positive,

$$g'(x) = 2\phi \mathcal{I}_{2,3} \frac{\sigma_\varepsilon^2 + \mathcal{I}_{1,1}^\beta}{\mathcal{I}_{2,2}^\beta} + (\omega + \phi \mathcal{I}_{1,2})\left(1 - 2\frac{\sigma_\varepsilon^2 + \mathcal{I}_{1,1}^\beta}{\mathcal{I}_{3,3}^\beta}\right) - (\omega + \phi \mathcal{I}_{1,1}) + x \mathcal{I}_{2,2} \tag{S54}$$

$$= 2\phi \mathcal{I}_{2,3} \frac{\sigma_\varepsilon^2 + \mathcal{I}_{1,1}^\beta}{\mathcal{I}_{2,2}^\beta} - 2(\omega + \phi \mathcal{I}_{1,2})\frac{\sigma_\varepsilon^2 + \mathcal{I}_{1,1}^\beta}{\mathcal{I}_{3,3}^\beta} \tag{S55}$$

$$= 2\frac{\sigma_\varepsilon^2 + \mathcal{I}_{1,1}^\beta}{(\mathcal{I}_{2,2}^\beta)^2}\left(\phi \mathcal{I}_{2,3} \mathcal{I}_{2,2}^\beta - (\omega + \phi \mathcal{I}_{1,2})\mathcal{I}_{3,3}^\beta\right) \tag{S56}$$

$$= 2\frac{\sigma_\varepsilon^2 + \mathcal{I}_{1,1}^\beta}{(\mathcal{I}_{2,2}^\beta)^2}\left(\phi^2 \mathcal{I}_{2,3} \mathcal{I}_{2,3}^\beta - (\omega + \phi \mathcal{I}_{1,2} - x\phi \mathcal{I}_{2,3})\mathcal{I}_{3,3}^\beta\right) \tag{S57}$$

$$= 2\frac{\sigma_\varepsilon^2 + \mathcal{I}_{1,1}^\beta}{(\mathcal{I}_{2,2}^\beta)^2}\left(\phi^2 \mathcal{I}_{2,3} \mathcal{I}_{2,3}^\beta - (\omega + \phi^2 \mathcal{I}_{1,3})\mathcal{I}_{3,3}^\beta\right) \tag{S58}$$

$$\leq 2\phi^2 \frac{\sigma_\varepsilon^2 + \mathcal{I}_{1,1}^\beta}{(\mathcal{I}_{2,2}^\beta)^2}\left(\mathcal{I}_{2,3} \mathcal{I}_{2,3}^\beta - \mathcal{I}_{1,3} \mathcal{I}_{3,3}^\beta\right) \tag{S59}$$

$$\leq 0 \,, \tag{S60}$$

where the last line follows from Corollary B.2 since we are assuming $\mu$ is aligned.

Next, regarding $x$ as a function of $\phi/\psi$, we consider the interval $(x_-, x_+)$ for $x_- = x(\phi/\psi = 0)$ and $x_+ = x(\phi/\psi = 1)$. From the self-consistent equation for $x$, we immediately see that $x_+(\omega + \mathcal{I}_{1,1}(x_+)) = 1$ and $x_- = 0$ so that

$$g(x_+) = -(\omega + \phi \mathcal{I}_{1,2})\frac{\sigma_\varepsilon^2 + \mathcal{I}_{1,1}^\beta}{\mathcal{I}_{2,2}^\beta} \tag{S61}$$

$$< 0 \,. \tag{S62}$$

and

$$g(x_-) = 1 - (\omega + \phi^2 \mathbb{E}_\mu[\lambda]) \frac{\sigma_\varepsilon^2 + \mathbb{E}_\mu[q\lambda])}{\mathbb{E}_\mu[q\lambda^2])} .$$ (S63)

Observe that,

$$g(x_-) > 0 \quad \Leftrightarrow \quad \sigma_\varepsilon^2 < \sigma_c^2 \equiv \frac{\mathbb{E}_\mu[q\lambda^2]}{\omega + \phi \mathbb{E}_\mu[\lambda]} - \mathbb{E}_\mu[q\lambda] .$$ (S64)

Therefore, from the intermediate value theorem, we conclude that $g$ has no real roots in $(x_-, x_+)$ for $\sigma_\varepsilon^2 > \sigma_c^2$, and exactly one real root if $\sigma_\varepsilon^2 < \sigma_c^2$. $\qquad\square$

## E    LINEAR REGRESSION LIMIT

To reduce to the linear case, we need to take $\psi \to 0$ and $\sigma(x) \to x$, in which case we have that $\eta = \zeta = \rho \to 1$ and $\omega \to 0$, so that

$$\tau_1 \to x \quad \text{and} \quad \bar{\tau}_1 \to \frac{1}{\gamma} ,$$ (S65)

so that

$$\gamma = \frac{1}{x} - \mathcal{I}_{1,1}$$ (S66)

$$= \frac{1}{x} - \phi \mathbb{E}_{s^2 \sim \mu_{\text{data}}} \frac{s^2}{\phi + xs^2} .$$ (S67)

### E.1    COMPARISON TO MEL & GANGULI (2021)

To compare with (Mel & Ganguli, 2021), note that $\phi = 1/\alpha$, $\gamma = 1/\phi\lambda$, $x = \tau_1 = \phi/\tilde{\lambda}$, so we have

$$\lambda = \phi \left( \frac{\tilde{\lambda}}{\phi} - \phi \mathbb{E}_{s^2 \sim \mu_{\text{data}}} \frac{s^2 \tilde{\lambda}/\phi}{\tilde{\lambda} + s^2} \right)$$ (S68)

$$= \tilde{\lambda} - \phi \mathbb{E}_{s^2 \sim \mu_{\text{data}}} \frac{s^2 \tilde{\lambda}}{\tilde{\lambda} + s^2} ,$$ (S69)

which is the expression appearing in Eq. (8) in (Mel & Ganguli, 2021). To compare expressions for the test error, note that

$$\frac{\partial x}{\partial \gamma} \to -\frac{x}{\gamma + \phi \mathcal{I}_{1,2}} ,$$ (S70)

and so,

$$\rho_f = \frac{\partial \tilde{\lambda}}{\partial \lambda}$$ (S71)

$$= \frac{\partial \phi/x}{\partial \phi \gamma}$$ (S72)

$$= -\frac{1}{x^2} \frac{\partial x}{\partial \gamma}$$ (S73)

$$= \frac{1}{x(\gamma + \phi \mathcal{I}_{1,2})} ,$$ (S74)

so that,

$$E = \phi\mathcal{I}_{1,2}^{\beta} + \frac{1}{\rho_f}\left(\phi\mathcal{I}_{1,2}^{\beta} + \sigma_\varepsilon^2\right)x^2\mathcal{I}_{2,2} \tag{S75}$$

$$= \phi\mathcal{I}_{1,2}^{\beta} + \frac{1}{\rho_f}\phi\mathcal{I}_{1,2}^{\beta}(xI_{1,1} - x\phi\mathcal{I}_{1,2}) + \frac{\sigma_\varepsilon^2}{\rho_f}x^2\mathcal{I}_{2,2} \tag{S76}$$

$$= \phi\mathcal{I}_{1,2}^{\beta} + \frac{1}{\rho_f}\phi\mathcal{I}_{1,2}^{\beta}(1 - x(\gamma + \phi\mathcal{I}_{1,2})) + \frac{\sigma_\varepsilon^2}{\rho_f}x^2\mathcal{I}_{2,2} \tag{S77}$$

$$= \phi\mathcal{I}_{1,2}^{\beta} + \frac{1}{\rho_f}\phi\mathcal{I}_{1,2}^{\beta}(1 - \rho_f) + \frac{\sigma_\varepsilon^2}{\rho_f}x^2\mathcal{I}_{2,2} \tag{S78}$$

$$= \frac{1}{\rho_f}\left(\phi\mathcal{I}_{1,2}^{\beta} + \sigma_\varepsilon^2 x^2\mathcal{I}_{2,2}\right). \tag{S79}$$

In contrast to our conventions, the error $\mathcal{F}$ in (Mel & Ganguli, 2021) does include an additive constant induced by the label noise, and also normalizes by the total output variance i.e. $\mathcal{F} = \frac{E+\sigma_\varepsilon^2}{\text{Var}[y]}$. Taking this relation into account and using the definitions of $\mathcal{I}$ and $\mathcal{I}^\beta$, and finally translating the notation via the substitutions $\phi \to 1/\alpha$, $\lambda q \to \mathbf{v} = (\mathbf{S}\mathbf{U}^T\mathbf{w})^2$, $\frac{\sigma_\varepsilon^2}{\text{Var}[y]} \to f_n$, $\frac{|\mathbf{v}|^2}{\text{Var}[y]} \to f_s$, we find

$$\mathcal{F} = \frac{E + \sigma_\varepsilon^2}{\text{Var}[y]} \tag{S80}$$

$$= \frac{\sigma_\varepsilon^2}{\text{Var}[y]} + \frac{1}{\rho_f}\left(\frac{1}{\text{Var}[y]}\mathbb{E}_\mu\left[q\lambda\left(\frac{\tilde{\lambda}}{\tilde{\lambda}+\lambda}\right)^2\right] + \phi\frac{\sigma_\varepsilon^2}{\text{Var}[y]}\mathbb{E}_\mu\left[\left(\frac{\lambda}{\tilde{\lambda}+\lambda}\right)^2\right]\right) \tag{S81}$$

$$= f_n + \frac{1}{\rho_f}\left(f_s\mathbb{E}_\mu\left[\hat{\mathbf{v}}^2\left(\frac{\tilde{\lambda}}{\tilde{\lambda}+\lambda}\right)^2\right] + f_n\frac{1}{\alpha}\mathbb{E}_\mu\left[\left(\frac{\lambda}{\tilde{\lambda}+\lambda}\right)^2\right]\right) \tag{S82}$$

which is Eq. (6) of (Mel & Ganguli, 2021).

### E.2  COMPARISON TO WU & XU (2020)

Wu & Xu (2020) study the case of anisotropic regularizer:

$$\hat{\beta}_\lambda = \left(X^\top X + \lambda\Sigma_w\right)^{-1}X^\top y \tag{S83}$$

with $n$ samples, $p$ features, $X \in \mathbb{R}^{n\times p}$ and $p/n \to \gamma$. After simplifying the error expression they arrive at eq. 3.1:

$$\mathbb{E}\left(\hat{y} - \tilde{x}^\top\hat{\beta}_\lambda\right)^2 = \tilde{\sigma}^2\left(1 + \frac{1}{n}\text{tr}\left(\Sigma_{x/w}\left(X_{/w}^\top X_{/w} + \lambda I\right)^{-1} - \lambda\Sigma_{x/w}\left(X_{/w}^\top X_{/w} + \lambda I\right)^{-2}\right)\right) \tag{S84}$$

$$+ \frac{\lambda^2}{n}\text{tr}\left(\Sigma_{x/w}\left(X_{/w}^\top X_{/w} + \lambda I\right)^{-1}\Sigma_{w\beta}\left(X_{/w}^\top X_{/w} + \lambda I\right)^{-1}\right) \tag{S85}$$

Setting $\Sigma_w \to I$ must give the expression for isotropic regularization, thus the effect of the weighting matrix $\Sigma_w$ can be accounted for by just changing the parameters of the isotropic model. The effective feature covariance is $\Sigma \to \Sigma_{x/w}$ and the effective weight covariance is $\Sigma_\beta \to \Sigma_{w\beta}$.

The error expression given in eqs. 4.1-4.3 is

$$\mathbb{E}\left[\left(\tilde{y} - \tilde{x}^\top\hat{\beta}_\lambda\right)^2\right] \to \frac{m'(-\lambda)}{m^2(-\lambda)}\cdot\left(\gamma\mathbb{E}\frac{gh}{\left(h\cdot m(-\lambda) + 1\right)^2} + \tilde{\sigma}^2\right) \tag{S86}$$

where

$$\lambda = \frac{1}{m(-\lambda)} - \gamma\mathbb{E}\frac{h}{1 + h\cdot m(-\lambda)} \tag{S87}$$

$$1 = \left(\frac{1}{m^2(-\lambda)} - \gamma\mathbb{E}\frac{h^2}{\left(h\cdot m(-\lambda) + 1\right)^2}\right)m'(-\lambda) \tag{S88}$$

In our notation, the predicted output on a new input $x$ is

$$\hat{y} = \left(\frac{1}{\sqrt{n_0}}\beta^\top X + \epsilon_{tr}\right)\left(\frac{1}{n_1}F^\top F + \gamma I_m\right)^{-1}\left(\frac{1}{n_1}F^\top f(x)\right) \tag{S89}$$

$$\rightarrow \left(\frac{1}{\sqrt{n_0}}\beta^\top X + \epsilon_{tr}\right)\left(\frac{1}{n_0}X^\top X + \gamma I_m\right)^{-1}\frac{1}{n_0}X^\top x \tag{S90}$$

$$= \left(\frac{1}{\sqrt{n_0}}\beta^\top X + \epsilon_{tr}\right)X^\top\left(\frac{1}{n_0}XX^\top + \gamma I_{n_0}\right)^{-1}\frac{1}{n_0}x \tag{S91}$$

$$= \hat{y}^\top \tilde{X}^\top\left(\tilde{X}\tilde{X}^\top + \phi\gamma I_{n_0}\right)^{-1}\tilde{x} \tag{S92}$$

where $\tilde{X}$ has $\frac{1}{\sqrt{m}} = \frac{1}{\sqrt{\text{samples}}}$ normalization. Thus translating our notation involves setting $\phi \rightarrow \gamma$, $\gamma \rightarrow \lambda/\gamma$, $\Sigma \rightarrow \Sigma_{x/w}$, $\lambda \rightarrow h$, $\Sigma_\beta \rightarrow \gamma\Sigma_{w\beta}$, and $q \rightarrow \gamma g$. In this new notation, our equation for $x$ reads

$$\lambda = \frac{\gamma}{x} - \gamma\mathbb{E}\frac{h}{1 + h \cdot \left(\frac{x}{\gamma}\right)} \tag{S93}$$

which shows $x \rightarrow \gamma m(-\lambda)$, and therefore $\frac{\partial x}{\partial\gamma} \rightarrow \frac{\partial\gamma m(-\lambda)}{\partial\lambda/\gamma} = -\gamma^2 m'(-\lambda)$. Next, note that

$$-\frac{\partial x}{\partial\gamma}\mathcal{I}_{2,2} = \frac{x}{\gamma + \phi\mathcal{I}_{1,2}}\mathcal{I}_{2,2} \tag{S94}$$

$$= \frac{\mathcal{I}_{1,1} - \phi\mathcal{I}_{1,2}}{\gamma + \phi\mathcal{I}_{1,2}} \tag{S95}$$

$$= \frac{\mathcal{I}_{1,1} + \gamma}{\gamma + \phi\mathcal{I}_{1,2}} - 1 \tag{S96}$$

$$= \frac{1/x}{\gamma + \phi\mathcal{I}_{1,2}} - 1 \tag{S97}$$

$$= -\frac{1}{x^2}\frac{\partial x}{\partial\gamma} - 1 \tag{S98}$$

So the full error is

$$E = \phi\mathcal{I}_{1,2}^\beta - \frac{\partial x}{\partial\gamma}\left(\phi\mathcal{I}_{1,2}^\beta\mathcal{I}_{2,2} + \sigma_e^2\mathcal{I}_{2,2}\right) \tag{S99}$$

$$= \phi\left(1 - \frac{\partial x}{\partial\gamma}\mathcal{I}_{2,2}\right)\mathcal{I}_{1,2}^\beta - \sigma_e^2\frac{\partial x}{\partial\gamma}\mathcal{I}_{2,2} \tag{S100}$$

$$= \left(-\frac{1}{x^2}\frac{\partial x}{\partial\gamma}\right)\phi\mathcal{I}_{1,2}^\beta + \sigma_e^2\left(-\frac{1}{x^2}\frac{\partial x}{\partial\gamma} - 1\right) \tag{S101}$$

$$= \left(-\frac{1}{x^2}\frac{\partial x}{\partial\gamma}\right)\left(\phi\mathcal{I}_{1,2}^\beta + \sigma_e^2\right) - \sigma_e^2 \tag{S102}$$

$$\rightarrow \frac{m'(-\lambda)}{m^2(-\lambda)}\left(\gamma\mathbb{E}\frac{hg}{(1 + m(-\lambda)h)^2} + \tilde{\sigma}^2\right) - \tilde{\sigma}^2 \tag{S103}$$

which, after removing the additive shift, matches the expressions given in (Wu & Xu, 2020) eq. 4.1.

## F  STRUCTURED LEARNING CURVES

### F.1  EFFECT OF SPECTRAL GAP

Here we demonstrate that a large gap in the spectrum of $\Sigma$ can induce steep cliffs in the learning curves as a function of the overparameterization $\phi/\psi$:

Suppose there is a gap in the spectrum of $\Sigma$ of size $g$. That is, there are $\lambda_- < \lambda_+$ such that there is no eigenvalue $\lambda \in (\lambda_-, \lambda_+)$ and $\frac{\lambda_+}{\lambda_-} = g$. Assuming $\phi < 1$ and $\mu$ is aligned, and working in the

noiseless ridgeless limit, we will show the slope of the learning curve $\frac{\partial \log E_\mu}{\partial (\phi/\psi)}$ becomes arbitrarily negative for small $\omega$.

From Theorem (3.1), $x, \tau_1$ satisfy

$$x = \frac{1 - \gamma \tau_1}{\omega + \phi \mathbb{E} \frac{\lambda}{\phi + x\lambda}} \tag{S104}$$

$$\tau_1 = \frac{\sqrt{(\psi - \phi)^2 + 4x\psi\phi\gamma/\rho} + \psi - \phi}{2\psi\gamma} \tag{S105}$$

Since $x \leq \frac{\min\{1, \phi/\psi\}}{\omega}$ (Eq. (S8)), for $\omega > 0$, $x$ stays finite in the ridgeless limit $\gamma \to 0$, so

$$\gamma \tau_1 \to \frac{|\psi - \phi| + \psi - \phi}{2\psi}. \tag{S106}$$

We have the numerator $1 - \gamma \tau_1 \to \min(1, \phi/\psi)$, and

$$x \left( \omega + \phi \mathbb{E} \frac{\lambda}{\phi + x\lambda} \right) = \min \left( 1, \frac{\phi}{\psi} \right). \tag{S107}$$

Since $x = 0$ is not a solution for $0 < \psi, \phi < \infty$, we can change variables to $\tilde{\gamma} = \frac{\phi}{x}$, giving

$$\omega \frac{1}{\tilde{\gamma}} + \mathbb{E} \frac{\lambda}{\tilde{\gamma} + \lambda} = \frac{1}{\phi} \min \left( 1, \frac{\phi}{\psi} \right) \tag{S108}$$

which implies $\tilde{\gamma}$ is a continuous decreasing function of $\phi/\psi$ (keeping $\phi$ fixed). Taking the limit of (S108) directly shows that $\tilde{\gamma}_{\max} := \lim_{\phi/\psi \to 0} \tilde{\gamma} = \infty$, while $\tilde{\gamma}_{\min} := \lim_{\phi/\psi \to \infty} \tilde{\gamma}$ satisfies

$$\omega \frac{1}{\tilde{\gamma}_{\min}} + \mathbb{E} \frac{\lambda}{\tilde{\gamma}_{\min} + \lambda} = \frac{1}{\phi} \tag{S109}$$

By the intermediate value theorem, $\tilde{\gamma}$ takes all values in the interval $(\tilde{\gamma}_{\min}, \infty)$. For $\phi < 1$, using $\mathbb{E} \frac{\lambda}{\tilde{\gamma}_{\min} + \lambda} \leq 1$ we obtain $\tilde{\gamma}_{\min} \leq \omega \frac{\phi}{1 - \phi}$.

We assume that $\omega \frac{\phi}{1 - \phi} \leq \lambda_-$, so the previous bound gives $\tilde{\gamma}_{\min} \leq \lambda_-$ and thus $\tilde{\gamma}$ attains all values in $(\lambda_-, \lambda_+)$. In particular, there is some $0 < \phi/\psi < 1$ such that $\tilde{\gamma}(\phi/\psi) = \sqrt{\lambda_- \lambda_+}$. At this point, differentiating (S108) gives

$$-\tilde{\gamma} \frac{\partial}{\partial \tilde{\gamma}} \frac{1}{\psi} = \omega \frac{1}{\tilde{\gamma}} + \tilde{\gamma} \mathbb{E} \frac{\lambda}{(\tilde{\gamma} + \lambda)^2} \tag{S110}$$

$$\leq \omega \frac{1}{\tilde{\gamma}} + \tilde{\gamma} \left( \frac{\lambda_+}{(\tilde{\gamma} + \lambda_+)^2} p(\lambda \geq \lambda_+) + \frac{\lambda_-}{(\tilde{\gamma} + \lambda_-)^2} p(\lambda \leq \lambda_-) \right) \tag{S111}$$

$$= \omega \frac{1}{\tilde{\gamma}} + \frac{\sqrt{g}}{(\sqrt{g} + 1)^2} \tag{S112}$$

Since $-\tilde{\gamma} \frac{\partial}{\partial \tilde{\gamma}} \frac{1}{\psi} = \frac{1}{\phi} \left( \frac{\partial \log x}{\partial (\phi/\psi)} \right)^{-1}$, we get

$$\frac{1}{\phi} \frac{1}{\omega \frac{1}{\tilde{\gamma}} + \frac{\sqrt{g}}{(\sqrt{g} + 1)^2}} \leq \frac{\partial \log x}{\partial (\phi/\psi)} \tag{S113}$$

For large spectral gap $g$ this tends toward

$$\frac{1}{\phi} \frac{\sqrt{\lambda_+ \lambda_-}}{\omega} \leq \frac{\partial \log x}{\partial (\phi/\psi)} \tag{S114}$$

If the nonlinearity $\omega$ is small compared to the middle of the spectral gap $\sqrt{\lambda_+\lambda_-}$, $x$ undergoes large fractional change as a function of the overparameterization ratio $\phi/\psi$.

To see how this affects the test error, we can use the lowering identity $\mathcal{I}^{\beta}_{a-1,b-1} = \phi\mathcal{I}^{\beta}_{a-1,b} + x\mathcal{I}^{\beta}_{a,b}$ to write the ridgeless error expression from Eq. (S46) as

$$E_\mu = \phi\mathcal{I}^{\beta}_{1,2} + \frac{\phi}{\psi - \phi}\left(\sigma^2_\epsilon + \mathcal{I}^{\beta}_{1,1}\right) + x\mathcal{I}^{\beta}_{2,2} \tag{S115}$$

$$= \frac{\phi}{\psi - \phi}\sigma^2_\epsilon + \frac{\psi}{\psi - \phi}\mathcal{I}^{\beta}_{1,1}. \tag{S116}$$

So we can write

$$\frac{\partial}{\partial\left(\phi/\psi\right)}\log\left(E_\mu - \frac{\phi}{\psi - \phi}\sigma^2_\epsilon\right) = \frac{\partial}{\partial\left(\phi/\psi\right)}\log\frac{\psi}{\psi - \phi}\mathcal{I}^{\beta}_{1,1} \tag{S117}$$

$$= \frac{\psi}{\psi - \phi} + \frac{\partial}{\partial\left(\phi/\psi\right)}\log\mathcal{I}^{\beta}_{1,1} \tag{S118}$$

For general $a, b$, we have

$$\frac{\partial}{\partial\left(\phi/\psi\right)}\log\mathcal{I}^{\beta}_{a,b} = -b\left(\frac{\partial\log x}{\partial\left(\phi/\psi\right)}\right)\frac{\mathbb{E}\left[\frac{\lambda^{a+1}}{(\bar\gamma+\lambda)^{b+1}}q\right]}{\mathbb{E}\left[\frac{\lambda^a}{(\bar\gamma+\lambda)^b}q\right]} \tag{S119}$$

Specializing to $a = b = 1$, and using the fact that $\frac{\lambda}{\bar\gamma+\lambda}\mathbb{E}_q\left[q|\lambda\right]$ is a nondecreasing function of $\lambda$ (guaranteed since $q$ is aligned), we may apply the Harris inequality to obtain

$$-\frac{\partial}{\partial\psi}\log\mathcal{I}^{\beta}_{1,1} = \left(\frac{\partial\log x}{\partial\left(\phi/\psi\right)}\right)\frac{\mathbb{E}_\lambda\left[\frac{\lambda}{\bar\gamma+\lambda}\left(\frac{\lambda}{\bar\gamma+\lambda}\mathbb{E}_q\left[q|\lambda\right]\right)\right]}{\mathbb{E}_\lambda\left[\left(\frac{\lambda}{(\bar\gamma+\lambda)}\mathbb{E}_q\left[q|\lambda\right]\right)\right]} \tag{S120}$$

$$\geq \left(\frac{\partial\log x}{\partial\left(\phi/\psi\right)}\right)\mathbb{E}_\lambda\left[\frac{\lambda}{\bar\gamma+\lambda}\right] \tag{S121}$$

$$\xrightarrow{g\to\infty}\left(\frac{\partial\log x}{\partial\left(\phi/\psi\right)}\right)p\left(\lambda > \lambda_+\right) \tag{S122}$$

$$\geq \frac{1}{\phi}\frac{\sqrt{\lambda_+\lambda_-}}{\omega}p\left(\lambda > \lambda_+\right), \tag{S123}$$

which implies

$$-\frac{\partial}{\partial\left(\phi/\psi\right)}\log\left(E_\mu - \frac{\phi}{\psi - \phi}\sigma^2_\epsilon\right) = -\frac{\psi}{\psi - \phi} - \frac{\partial}{\partial\left(\phi/\psi\right)}\log\mathcal{I}^{\beta}_{1,1} \tag{S124}$$

$$\geq -\frac{\psi}{\psi - \phi} + \frac{1}{\phi}\frac{\sqrt{\lambda_+\lambda_-}}{\omega}p\left(\lambda > \lambda_+\right) \tag{S125}$$

In particular, if $\sigma^2_\varepsilon = 0$, then

$$-\frac{\partial\log E_\mu}{\partial\left(\phi/\psi\right)} \geq -\frac{\psi}{\psi - \phi} + \frac{1}{\phi}\frac{\sqrt{\lambda_+\lambda_-}}{\omega}p\left(\lambda > \lambda_+\right) \tag{S126}$$

Thus as $\omega \to 0$ the learning curve becomes arbitrarily steep at the critical value $x = \phi/\sqrt{\lambda_+\lambda_-}$.

### F.2 ANALYSIS OF THE D-SCALE MODEL IN THE SEPARATED LIMIT

We will consider the $d$-scale covariance model:

$$\lambda_n = C\alpha^n, \quad p_n = \frac{1}{d}, \quad n = 0, 1, \cdots, d - 1 \tag{S127}$$

where $C$ is chosen so that

$$1 = s = \bar{\text{tr}}\left[\Sigma\right] = \frac{1}{d}\sum_{n=0}^{d-1}C\alpha^n = C\frac{1}{d}\frac{1 - \alpha^d}{1 - \alpha} \tag{S128}$$

We will obtain expressions for $x$ in the limit of small $\lambda$. Consider the ridgeless limit of $\tilde{\gamma} := \phi/x$:

$$\frac{1}{\tilde{\gamma}}\omega + \sum_n p_n \frac{\lambda_n}{\tilde{\gamma} + \lambda_n} = \frac{1}{\max(\phi, \psi)} \tag{S129}$$

Suppose, $\omega$ sits between the scales $C\alpha^j, C\alpha^{j+1}$. To enforce this constraint, we will take $\omega = \hat{\omega}\alpha^{j+\frac{1}{2}}$ where $\hat{\omega}$ is a constant independent of $\alpha$.

The $\alpha$ scaling of $\tilde{\gamma}$ will depend on the value of $\max(\phi, \psi)$. Discarding the second term in (S129) we obtain $\max(\phi, \psi)\omega \leq \tilde{\gamma}$, and thus the lowest possible scaling for $\tilde{\gamma}$ is $\tilde{\gamma} = C_{j+\frac{1}{2}}\alpha^{j+\frac{1}{2}}$. Substituting this ansatz into (S129) and taking the limit $\alpha \to 0$, we obtain

$$\frac{1}{\max(\phi, \psi)} = \frac{1}{\tilde{\gamma}}\omega + \frac{1}{d}\sum_n \frac{C\alpha^n}{C_{j+\frac{1}{2}}\alpha^{j+\frac{1}{2}} + C\alpha^n} \tag{S130}$$

$$\xrightarrow{\alpha \to 0} \frac{1}{\tilde{\gamma}}\omega + \frac{j+1}{d} \tag{S131}$$

Solving for $\tilde{\gamma}$ gives $\tilde{\gamma} = \frac{\max(\phi,\psi)}{1-\max(\phi,\psi)\frac{j+1}{d}}\omega$. For other values of $\max(\phi, \psi)$, $\tilde{\gamma}$ may have higher scaling, ie. $\tilde{\gamma} = C_k\alpha^k$ with $k \leq j$. Substituting and solving for $\tilde{\gamma}$ we obtain $\tilde{\gamma} = \frac{\max(\phi,\psi)\frac{k+1}{d}-1}{1-\max(\phi,\psi)\frac{k}{d}}\lambda_k$. Thus we obtain the following self-consistent solutions for $\tilde{\gamma}$:

$$\tilde{\gamma} = \begin{cases} \frac{\max(\phi,\psi)}{1-\max(\phi,\psi)\frac{j+1}{d}}\omega & \max(\phi, \psi) < \frac{d}{j+1} \\ \frac{\max(\phi,\psi)\frac{k+1}{d}-1}{1-\max(\phi,\psi)\frac{k}{d}}\lambda_k & \frac{d}{k+1} < \max(\phi, \psi) < \frac{d}{k} \end{cases} \tag{S132}$$

Thus $\tilde{\gamma}$ takes on the scale of a single eigenvalue $\lambda_k$ for a range of overparameterization ratios corresponding to $\frac{d}{k+1} < \psi\max\left(\frac{\phi}{\psi}, 1\right) < \frac{d}{k}$. To understand what happens at the transitions between these regimes, we can apply the results from the previous subsection F.1 for generic $\Sigma$ with large spectral gap. In the notation of F.1, the $D$-scale model has a spectral gap between each pair of consecutive scales of size $g = \lambda_j/\lambda_{j+1} = C\alpha^j/C\alpha^{j+1} = 1/\alpha$ and as a consequence, $\tilde{\gamma}$ will exhibit near infinite slop as it passes through the middle of a gap $\sqrt{\lambda_{j+1}\lambda_j} = C\alpha^{j+\frac{1}{2}}$. Comparing to the self-consistent solutions (S132) these transitions must happen at the critical values $\max(\phi, \psi) = \frac{d}{k+1}$ for $k \leq j$. At these transition points, the error exhibits steep cliffs in the parameter regime descried in F.1.

## G  PROOF OF THEOREM 3.1

The proof closely follows the methods described in (Adlam et al., 2019; Adlam & Pennington, 2020a;b; Tripuraneni et al., 2021a;b). Indeed, precisely the same techniques from operator-valued free probability used in those works apply here. The main and only difference is the anisotropic weight covariance $\Sigma_\beta$, which changes the details of the computations but not the arguments justifying the linearized Gaussian equivalents and the application of operator-valued free probability. We therefore refer the reader to those previous works for an in-depth discussion of methods and merely focus here on the details of the requisite calculations. Throughout this section, we use $\overline{\text{tr}}$ to denote the dimension-normalized trace, i.e. $\overline{\text{tr}}(A) = \frac{1}{n}\text{tr}(A)$ for a matrix $A \in \mathbb{R}^{n \times n}$.

### G.1  DECOMPOSITION OF THE TEST LOSS

The test loss can be written as,

$$E_{\Sigma^*} = \mathbb{E}_{(\mathbf{x},y)}(y - \hat{y}(\mathbf{x}))^2 = E_1 + E_2 + E_3 \tag{S133}$$

with

$$E_1 = \mathbb{E}_{(\mathbf{x},\varepsilon)}\text{tr}(y(\mathbf{x})y(\mathbf{x})^\top) \tag{S134}$$

$$E_2 = -2\mathbb{E}_{(\mathbf{x},\varepsilon)}\text{tr}(K_{\mathbf{x}}^\top K^{-1}Y^\top y(\mathbf{x})) \tag{S135}$$

$$E_3 = \mathbb{E}_{(\mathbf{x},\varepsilon)}\text{tr}(K_{\mathbf{x}}^\top K^{-1}Y^\top Y K^{-1}K_{\mathbf{x}}). \tag{S136}$$

Recall the kernels $K = K(X, X)$ and $K_{\mathbf{x}} = K(X, \mathbf{x})$ are given by,

$$K = \frac{F^\top F}{n_1} + \gamma I_m \qquad \text{and} \qquad K_{\mathbf{x}} = \frac{1}{n_1} F^\top f\,. \tag{S137}$$

Using the cyclicity and linearity of the trace, the expectation over $\mathbf{x}$ requires the computation of

$$\mathbb{E}_{\mathbf{x}} K_{\mathbf{x}} K_{\mathbf{x}}^\top\,, \qquad \mathbb{E}_{\mathbf{x}} y(\mathbf{x}) K_{\mathbf{x}}^\top\,, \qquad \mathbb{E}_{\mathbf{x}} y(\mathbf{x}) y(\mathbf{x})^\top\,. \tag{S138}$$

As described in detail in (Tripuraneni et al., 2021a;b; Adlam et al., 2019; Adlam & Pennington, 2020a; Mei & Montanari, 2019), asympotically the trace terms $E_1$, $E_2$, and $E_3$ are invariant to a linearization of the random feature vector $f$,

$$f \to f^{\mathrm{lin}} = \frac{\sqrt{\rho}}{\sqrt{n_0}} W \mathbf{x} + \sqrt{\eta - \zeta}\,\theta\,, \tag{S139}$$

where $\theta \in \mathbb{R}^{n_1}$ is a vector of iid standard normal variates. Similarly, we will take the linearization of the training features to be $\frac{\sqrt{\rho}}{\sqrt{n_0}} W X + \sqrt{\eta - \zeta}\Theta$ where $\Theta \in \mathbb{R}^{n_1 \times m}$ has standard normal components. The expectations over $\mathbf{x}$ are now trivial and we readily find,

$$\mathbb{E}_{\mathbf{x}} K_{\mathbf{x}} K_{\mathbf{x}}^\top = \frac{1}{n_1^2} F^\top \Big( \frac{\rho}{n_0} W \Sigma W^\top + (\eta - \zeta) I_{n_1} \Big) F \tag{S140}$$

$$\mathbb{E}_{\mathbf{x}} y(\mathbf{x}) K_{\mathbf{x}}^\top = \frac{\sqrt{\rho}}{n_0 n_1} \beta^\top \Sigma W^\top F \tag{S141}$$

$$\mathbb{E}_{\mathbf{x}} y(\mathbf{x}) y(\mathbf{x})^\top = \frac{1}{n_0} \beta \Sigma \beta^\top \tag{S142}$$

Next, we recall the definition, $Y = \beta^\top X / \sqrt{n_0} + \boldsymbol{\epsilon}$, and, using the above substitution, we find

$$\mathbb{E}_{\boldsymbol{\epsilon}} \left[ Y^\top Y \right] = \frac{1}{n_0} X^\top \Sigma_\beta X + \sigma_\varepsilon^2 I_m \tag{S143}$$

$$\mathbb{E}_{\boldsymbol{\epsilon}} \left[ Y^\top \mathbb{E}_{\mathbf{x}} y(\mathbf{x}) K_{\mathbf{x}}^\top \right] = \frac{\sqrt{\rho}}{n_0^{3/2} n_1} X^\top \Sigma_\beta \Sigma W^\top F\,. \tag{S144}$$

Putting these pieces together, we have

$$E_1 = \frac{\mathrm{tr}(\Sigma_\beta \Sigma)}{n_0} \tag{S145}$$

$$E_2 = E_{21} \tag{S146}$$

$$E_3 = E_{31} + E_{32}\,, \tag{S147}$$

where,

$$E_{21} = -2 \frac{\sqrt{\rho}}{n_0^{3/2} n_1} \mathbb{E} \mathrm{tr} \left( X^\top \Sigma_\beta \Sigma W^\top F K^{-1} \right) \tag{S148}$$

$$E_{31} = \sigma_\varepsilon^2 \mathbb{E} \mathrm{tr} \left( K^{-1} \Sigma_3 K^{-1} \right) \tag{S149}$$

$$E_{32} = \frac{1}{n_0} \mathbb{E} \mathrm{tr} \left( K^{-1} \Sigma_3 K^{-1} X^\top \Sigma_\beta X \right) \tag{S150}$$

and,

$$\Sigma_3 = \frac{\rho}{n_0 n_1^2} F^\top W \Sigma W^\top F + \frac{\eta - \zeta}{n_1^2} F^\top F\,. \tag{S151}$$

## G.2 Decomposition of the bias and total variance

Note that it is sufficient to calculate the bias term given the total test loss, since the total variance can be obtained as $V_\Sigma = E_\Sigma - B_\Sigma$. Following the total multivariate bias-variance decomposition

of (Adlam & Pennington, 2020b), for each random variable in question we introduce an iid copy of it denoted by either the subscript 1 or 2. We can then write,

$$B_\Sigma = \mathbb{E}_{(\mathbf{x},y)}(y - \mathbb{E}_{(W,X,\varepsilon)}\hat{y}(\mathbf{x};W,X,\varepsilon))^2 \tag{S152}$$

$$= \mathbb{E}_{(\mathbf{x},y)}\mathbb{E}_{(W_1,X_1,\varepsilon_1)}\mathbb{E}_{(W_2,X_2,\epsilon_2)}(y - \hat{y}(\mathbf{x};W_1,X_1,\varepsilon_1))(y - \hat{y}(\mathbf{x};W_2,X_2,\epsilon_2)) \tag{S153}$$

$$= \frac{\text{tr}(\Sigma_\beta\Sigma)}{n_0} + E_{21} + H_{000}\,, \tag{S154}$$

where an expression for $E_{21}$ was given previously and $H_{000}$ satisfies

$$H_{000} = \mathbb{E}\hat{y}(\mathbf{x};W_1,X_1,\varepsilon_1)\hat{y}(\mathbf{x};W_2,X_2,\epsilon_2)\,, \tag{S155}$$

where the expectations are over $\mathbf{x}, W_1, X_1, \varepsilon_1, W_2, X_2$, and $\epsilon_2$. Recalling the definition of $\hat{y}$,

$$\hat{y}(\mathbf{x};W,X,\varepsilon) := Y(X,\epsilon)K(X,X;W)^{-1}K(X,\mathbf{x};W) \tag{S156}$$

and the techniques described in the previous section, it is straightforward to analyze the above term. First note we can write,

$$\mathbb{E}_\mathbf{x}K(X_1,\mathbf{x};W_1)K(\mathbf{x},X_2;W_2) = \frac{\rho}{n_0 n_1^2}F_{11}^\top W_1\Sigma W_2^\top F_{22}\,. \tag{S157}$$

Here we have defined $F_{11} \equiv F(W_1,X_1)$ and $F_{22} \equiv F(W_2,X_2)$. Now we proceed to calculate $H_{000}$ as

$$H_{000} = \mathbb{E}\hat{y}(\mathbf{x};W_1,X_1,\varepsilon)\hat{y}(\mathbf{x};W_2,X_2,\epsilon_2) \tag{S158}$$

$$= \mathbb{E}K(\mathbf{x},X_2;W_2)K(X_2,X_2;W_2)^{-1}Y(X_2,\epsilon_2)^\top Y(X_1,\varepsilon_1)K(X_1,X_1;W_1)^{-1}K(X_1,\mathbf{x};W) \tag{S159}$$

$$= \mathbb{E}\text{tr}\left(K(X_2,X_2;W_2)^{-1}X_2^\top X_1 K(X_1,X_1;W_1)^{-1}K(X_1,\mathbf{x};W)K(\mathbf{x},X_2;W_2)\right) \tag{S160}$$

$$= \frac{\rho}{n_0^2 n_1^2}\mathbb{E}\text{tr}\left(K_{22}^{-1}X_2^\top\Sigma_\beta X_1 K_{11}^{-1}F_{11}^\top W_1\Sigma W_2^\top F_{22}\right) \tag{S161}$$

$$\equiv E_4\,, \tag{S162}$$

where in the second-to-last line we have defined $K_{11} \equiv K(X_1,X_1;W_1)$ and $K_{22} \equiv K(X_2,X_2;W_2)$.

### G.3 SUMMARY OF LINEARIZED TRACE TERMS

We now summarize the requisite terms needed to compute the total test error, bias, and variance after using cyclicity of the trace to rearrange several of them. In the following, we slightly change notation in order to make explicit the dependence on the covariance matrix $\Sigma$. To be specific, whereas above we assumed that the columns of $X_1$ and $X_2$ were drawn from multivariate Gaussians with covariance $\Sigma$, below we assume that they are drawn from multivariate Gausssians with identity covariance. This change is equivalent to replacing $X_1 \to \Sigma^{1/2}X_1$ and $X_2 \to \Sigma^{1/2}X_2$ in the above expressions. We utilize this definition so that $X_1, X_2, W_1, W_2$, and $\Theta$ all have iid standard Gaussian entries. From the previous computations, we can now write the requisite terms as,

$$\Sigma_3 = \frac{\rho}{n_0 n_1^2}F_{11}^\top W_1\Sigma W_1^\top F_{11} + \frac{\eta - \zeta}{n_1^2}F_{11}^\top F_{11} \tag{S163}$$

$$E_{21} = -2\frac{\sqrt{\rho}}{n_0^{3/2}n_1}\text{tr}\left(X_1^\top\Sigma^{1/2}\Sigma_\beta\Sigma W_1^\top F_{11}K_{11}^{-1}\right) \tag{S164}$$

$$E_{31} = \sigma_\epsilon^2\text{tr}\left(K_{11}^{-1}\Sigma_3 K_{11}^{-1}\right) \tag{S165}$$

$$E_{32} = \frac{1}{n_0}\text{tr}\left(K_{11}^{-1}\Sigma_3 K_{11}^{-1}X_1^\top\Sigma^{1/2}\Sigma_\beta\Sigma^{1/2}X_1\right) \tag{S166}$$

$$E_4 = \frac{\rho}{n_0^2 n_1^2}\text{tr}\left(F_{22}K_{22}^{-1}X_2^\top\Sigma^{1/2}\Sigma_\beta\Sigma^{1/2}X_1 K_{11}^{-1}F_{11}^\top W_1\Sigma W_2^\top\right) \tag{S167}$$

$$E_\Sigma = \frac{1}{n_0}\text{tr}\left(\Sigma\Sigma_\beta\right) + E_{21} + E_{31} + E_{32} \tag{S168}$$

$$B_\Sigma = \frac{1}{n_0}\text{tr}\left(\Sigma\Sigma_\beta\right) + E_{21} + E_4 \tag{S169}$$

$$V_\Sigma = E_\Sigma - B_\Sigma \tag{S170}$$

### G.4 Calculation of error terms

To compute the test error, bias, and total variance, we need to evaluate the asymptotic trace objects appearing in the expressions for $E_{21}$, $E_{31}$, $E_{32}$, and $E_4$, defined in the previous section. As these expressions are essentially rational functions of the random matrices $X$, $W$, $\Theta$, $\Sigma$, and $\Sigma_\beta$, these computations can be accomplished by representing the rational functions as single blocks of a suitably-defined block matrix inverse - the so-called linear pencil method (see eg . Far et al., 2006) - and then applying the theory of operator-valued free probability (Mingo & Speicher, 2017). These techniques and their application to problems of this type have been well-established elsewhere (Adlam et al., 2019; Adlam & Pennington, 2020a;b), we only lightly sketch the mathematical details, referring the reader to the literature for a more pedagogical overview. Instead, we focus on presenting the details of the requisite calculations.

Relative to prior work, the main challenge in the current setting is generalizing the calculations to include an arbitrary weight covariance matrix $\Sigma_\beta$. This generalization is facilitated by the general theory of operator-valued free probability, and in particular through the subordinated form of the operator-valued self-consistent equations that we first present in eqn. (S201). The form of this equation enables the simple computation of the operator-valued R-transform of the remaining random matrices, $W$, $X$, and $\Theta$, which are all iid Gaussian and can therefore be obtained simply by using the methods of (Far et al., 2006). The remaining complication amounts to performing the trace in eqn. (S201), which asymptotically becomes an integral over the LJSD $\mu$. While this might in general lead to a complicated coupling of many transcendental equations, it turns out that the trascendentality can be entirely factored into a single scalar fixed-point equation, whose solution we denote by $x$ (see eqn. (S237)), and the remaining equations are purely algebraic given $x$. To facilitate this particular simplification, it is necessary to first compute all of the entries in the operator-valued Stieltjes transform of the kernel matrix $K$, which we do in Sec. G.4.1. Using these results, we compute the remaining error terms in the subsequent sections.

As a matter of notation, note that throughout this entire section whenever a matrix $X$, $X_1$, or $X_2$ appears it is composed of iid $\mathcal{N}(0,1)$ entries as in Appendix G.3. This differs from the notation of the main paper, but we follow this prescription to ease the already cumbersome presentation. This definition of $X$ allows us to explicitly extract and represent the training covariance $\Sigma$ in our calculations.

### G.4.1 $K^{-1}$

The NCAlgebra Mathematica package (NCRealization method; algorithm described in Helton et al., 2006) was used to generate the following matrix pencil $Q^{K^{-1}}$:

$$Q^{K^{-1}} = \begin{pmatrix} I_m & \frac{\sqrt{\eta-\zeta}\Theta^\top}{\gamma\sqrt{n_1}} & \frac{\sqrt{\rho}X^\top}{\gamma\sqrt{n_0}} & 0 & 0 & 0 & 0 & 0 & 0 \\ -\frac{\Theta\sqrt{\eta-\zeta}}{\sqrt{n_1}} & I_{n_1} & 0 & 0 & -\frac{\sqrt{\rho}W}{\sqrt{n_1}} & 0 & 0 & 0 & 0 \\ 0 & 0 & I_{n_0} & -\Sigma^{1/2} & 0 & 0 & 0 & 0 & 0 \\ 0 & -\frac{W^\top}{\sqrt{n_1}} & 0 & I_{n_0} & 0 & 0 & \frac{\Sigma_\beta}{\sqrt{\rho}} & 0 & 0 \\ 0 & 0 & 0 & 0 & I_{n_0} & -\Sigma^{1/2} & 0 & 0 & 0 \\ -\frac{X}{\sqrt{n_0}} & 0 & 0 & 0 & 0 & I_{n_0} & 0 & 0 & 0 \\ 0 & 0 & 0 & 0 & 0 & 0 & I_{n_0} & -\Sigma^{1/2} & 0 \\ 0 & 0 & 0 & 0 & 0 & 0 & 0 & I_{n_0} & -\frac{X}{\sqrt{n_0}} \\ 0 & 0 & 0 & 0 & 0 & 0 & 0 & 0 & I_m \end{pmatrix}. \tag{S171}$$

This matrix is specifically chosen so that inverting $[Q^{K^{-1}}]^\top$ and taking the normalized trace of its first block gives exactly $\gamma \bar{\mathrm{tr}} K^{-1}$, the quantity of interest. Computing the full inverse of $[Q^{K^{-1}}]^\top$ via repeated applications of the Schur complement formula and taking block-wise traces shows that

$$G_{1,1}^{K^{-1}} = \gamma\,\bar{\mathrm{tr}}(K^{-1}) \tag{S172}$$

$$G_{9,1}^{K^{-1}} = \frac{\phi\,\bar{\mathrm{tr}}\left(\Sigma_\beta\Sigma^{1/2}XK^{-1}X^\top\Sigma^{1/2}\right)}{n_0} \tag{S173}$$

$$G_{2,2}^{K^{-1}} = \gamma\,\bar{\mathrm{tr}}(\hat{K}^{-1}) \tag{S174}$$

$$G_{3,3}^{K^{-1}} = G_{6,6}^{K^{-1}} = 1 - \frac{\sqrt{\rho}\,\bar{\mathrm{tr}}\left(\Sigma^{1/2}W^\top F K^{-1} X^\top\right)}{\sqrt{n_0} n_1} \tag{S175}$$

$$G_{4,3}^{K^{-1}} = G_{6,5}^{K^{-1}} = \bar{\mathrm{tr}}(\Sigma^{1/2}) - \frac{\sqrt{\rho}\,\bar{\mathrm{tr}}\left(\Sigma W^\top F K^{-1} X^\top\right)}{\sqrt{n_0} n_1} \tag{S176}$$

$$G_{5,3}^{K^{-1}} = G_{6,4}^{K^{-1}} = \frac{\gamma\sqrt{\rho}\,\bar{\mathrm{tr}}\left(\Sigma^{1/2}W^\top \hat{K}^{-1} W\right)}{n_1} \tag{S177}$$

$$G_{6,3}^{K^{-1}} = \frac{\gamma\sqrt{\rho}\,\bar{\mathrm{tr}}\left(\Sigma W^\top \hat{K}^{-1} W\right)}{n_1} \tag{S178}$$

$$G_{7,3}^{K^{-1}} = \frac{\bar{\mathrm{tr}}\left(\Sigma_\beta \Sigma^{1/2} W^\top F K^{-1} X^\top \Sigma^{1/2}\right)}{\sqrt{n_0} n_1} - \frac{\bar{\mathrm{tr}}(\Sigma_\beta \Sigma^{1/2})}{\sqrt{\rho}} \tag{S179}$$

$$G_{8,3}^{K^{-1}} = \frac{\bar{\mathrm{tr}}\left(\Sigma_\beta \Sigma W^\top F K^{-1} X^\top \Sigma^{1/2}\right)}{\sqrt{n_0} n_1} - \frac{\bar{\mathrm{tr}}(\Sigma_\beta \Sigma)}{\sqrt{\rho}} \tag{S180}$$

$$G_{3,4}^{K^{-1}} = G_{5,6}^{K^{-1}} = -\frac{\sqrt{\rho}\,\bar{\mathrm{tr}}\left(F K^{-1} X^\top W^\top\right)}{\sqrt{n_0} n_1 \psi} \tag{S181}$$

$$G_{4,4}^{K^{-1}} = G_{5,5}^{K^{-1}} = 1 - \frac{\sqrt{\rho}\,\bar{\mathrm{tr}}\left(\Sigma^{1/2}W^\top F K^{-1} X^\top\right)}{\sqrt{n_0} n_1} \tag{S182}$$

$$G_{5,4}^{K^{-1}} = \frac{\gamma\sqrt{\rho}\,\bar{\mathrm{tr}}\left(\hat{K}^{-1} W W^\top\right)}{n_1 \psi} \tag{S183}$$

$$G_{7,4}^{K^{-1}} = \frac{\bar{\mathrm{tr}}\left(\Sigma_\beta \Sigma^{1/2} X F^\top \hat{K}^{-1} W\right)}{\sqrt{n_0} n_1} - \frac{\bar{\mathrm{tr}}(\Sigma_\beta)}{\sqrt{\rho}} \tag{S184}$$

$$G_{8,4}^{K^{-1}} = \frac{\bar{\mathrm{tr}}\left(\Sigma_\beta \Sigma^{1/2} W^\top F K^{-1} X^\top \Sigma^{1/2}\right)}{\sqrt{n_0} n_1} - \frac{\bar{\mathrm{tr}}(\Sigma_\beta \Sigma^{1/2})}{\sqrt{\rho}} \tag{S185}$$

$$G_{3,5}^{K^{-1}} = G_{4,6}^{K^{-1}} = -\frac{\sqrt{\rho}\,\bar{\mathrm{tr}}\left(\Sigma^{1/2} X K^{-1} X^\top\right)}{n_0} \tag{S186}$$

$$G_{4,5}^{K^{-1}} = -\frac{\sqrt{\rho}\,\bar{\mathrm{tr}}\left(\Sigma X K^{-1} X^\top\right)}{n_0} \tag{S187}$$

$$G_{7,5}^{K^{-1}} = \frac{\bar{\mathrm{tr}}\left(\Sigma_\beta \Sigma^{1/2} X K^{-1} X^\top \Sigma^{1/2}\right)}{n_0} \tag{S188}$$

$$G_{8,5}^{K^{-1}} = \frac{\bar{\mathrm{tr}}\left(\Sigma_\beta \Sigma X K^{-1} X^\top \Sigma^{1/2}\right)}{n_0} \tag{S189}$$

$$G_{3,6}^{K^{-1}} = -\frac{\sqrt{\rho}\,\bar{\mathrm{tr}}\left(K^{-1} X^\top X\right)}{n_0 \phi} \tag{S190}$$

$$G_{7,6}^{K^{-1}} = \frac{\bar{\mathrm{tr}}\left(\Sigma_\beta \Sigma^{1/2} X K^{-1} X^\top\right)}{n_0} \tag{S191}$$

$$G_{8,6}^{K^{-1}} = \frac{\bar{\mathrm{tr}}\left(\Sigma_\beta \Sigma^{1/2} X K^{-1} X^\top \Sigma^{1/2}\right)}{n_0} \tag{S192}$$

$$G_{7,7}^{K^{-1}} = G_{8,8}^{K^{-1}} = G_{9,9}^{K^{-1}} = 1 \tag{S193}$$

$$G_{8,7}^{K^{-1}} = \bar{\mathrm{tr}}(\Sigma^{1/2}), \tag{S194}$$

where $G^{K^{-1}} := \mathrm{id}_9 \otimes \bar{\mathrm{tr}}\,[(Q^{K^{-1}})^\top]^{-1} \in M_9(\mathbb{C})$ is a scalar $9 \times 9$ matrix whose $i, j$ entry $G_{i,j}^{K^{-1}}$ is the normalized trace of the $(i,j)$-block of the inverse of $[Q^{K^{-1}}]^\top$. We have also defined $\hat{K} = \frac{1}{n_1} F F^\top + \gamma I_{n_1}$ (note that $K$ is $m \times m$ while $\hat{K}$ is $n_1 \times n_1$). It is straightforward to verify that when

the $n_0, n_1, m \to \infty$ limit is eventually taken, each entry of $G^{K^{-1}}$ is properly scaled and will tend toward a finite value.

We aim to compute the limiting values of these trace terms as $n_0, n_1, m \to \infty$, as they will be related to the error terms of interest. To proceed, recall that the asymptotic block-wise traces of the inverse of $Q^{K^{-1}}$ can be determined from its operator-valued Stieltjes transform (Mingo & Speicher, 2017). The simplest way to apply the results of (Far et al., 2006; Mingo & Speicher, 2017) is to augment $Q^{K^{-1}}$ to form the the self-adjoint matrix $\bar{Q}^{K^{-1}}$,

$$\bar{Q}^{K^{-1}} = \begin{pmatrix} 0 & [Q^{K^{-1}}]^\top \\ Q^{K^{-1}} & 0 \end{pmatrix}, \tag{S195}$$

and observe that we can write $\bar{Q}^{K^{-1}}$ as,

$$\begin{aligned} \bar{Q}^{K^{-1}} &= \bar{Z} - \bar{Q}^{K^{-1}}_{W,X,\Theta} - \bar{Q}^{K^{-1}}_\Sigma \\ &= \begin{pmatrix} 0 & I_9 \\ I_9 & 0 \end{pmatrix} - \begin{pmatrix} 0 & [Q^{K^{-1}}_{W,X,\Theta}]^\top \\ Q^{K^{-1}}_{W,X,\Theta} & 0 \end{pmatrix} - \begin{pmatrix} 0 & [Q^{K^{-1}}_\Sigma]^\top \\ Q^{K^{-1}}_\Sigma & 0 \end{pmatrix}, \end{aligned} \tag{S196}$$

where

$$Q^{K^{-1}}_{W,X,\Theta} = - \begin{pmatrix} 0 & \frac{\sqrt{\eta-\zeta}\Theta^\top}{\gamma\sqrt{n_1}} & \frac{\sqrt{\rho}X^\top}{\gamma\sqrt{n_0}} & 0 & 0 & 0 & 0 & 0 & 0 \\ -\frac{\Theta\sqrt{\eta-\zeta}}{\sqrt{n_1}} & 0 & 0 & 0 & -\frac{\sqrt{\rho}W}{\sqrt{n_1}} & 0 & 0 & 0 & 0 \\ 0 & 0 & 0 & 0 & 0 & 0 & 0 & 0 & 0 \\ 0 & -\frac{W^\top}{\sqrt{n_1}} & 0 & 0 & 0 & 0 & 0 & 0 & 0 \\ 0 & 0 & 0 & 0 & 0 & 0 & 0 & 0 & 0 \\ -\frac{X}{\sqrt{n_0}} & 0 & 0 & 0 & 0 & 0 & 0 & 0 & 0 \\ 0 & 0 & 0 & 0 & 0 & 0 & 0 & 0 & 0 \\ 0 & 0 & 0 & 0 & 0 & 0 & 0 & 0 & -\frac{X}{\sqrt{n_0}} \\ 0 & 0 & 0 & 0 & 0 & 0 & 0 & 0 & 0 \end{pmatrix} \tag{S197}$$

$$Q^{K^{-1}}_\Sigma = - \begin{pmatrix} 0 & 0 & 0 & 0 & 0 & 0 & 0 & 0 & 0 \\ 0 & 0 & 0 & 0 & 0 & 0 & 0 & 0 & 0 \\ 0 & 0 & 0 & -\Sigma^{1/2} & 0 & 0 & 0 & 0 & 0 \\ 0 & 0 & 0 & 0 & 0 & 0 & \frac{\Sigma_\beta}{\sqrt{\rho}} & 0 & 0 \\ 0 & 0 & 0 & 0 & 0 & -\Sigma^{1/2} & 0 & 0 & 0 \\ 0 & 0 & 0 & 0 & 0 & 0 & 0 & 0 & 0 \\ 0 & 0 & 0 & 0 & 0 & 0 & 0 & -\Sigma^{1/2} & 0 \\ 0 & 0 & 0 & 0 & 0 & 0 & 0 & 0 & 0 \\ 0 & 0 & 0 & 0 & 0 & 0 & 0 & 0 & 0 \end{pmatrix}, \tag{S198}$$

and the addition in (S196) is performed block-wise. Note that we have separated the iid Gaussian matrices $W, X, \Theta$ from the constant terms and from the $\Sigma$-dependent terms. Denote by $\bar{G}^{K^{-1}} \in M_{18}(\mathbb{C})$ the block matrix

$$\bar{G}^{K^{-1}} = \begin{pmatrix} 0 & [G^{K^{-1}}]^\top \\ G^{K^{-1}} & 0 \end{pmatrix} = \mathrm{id}_{18} \otimes \bar{\mathrm{tr}}\left(\bar{Q}^{K^{-1}}\right)^{-1}, \tag{S199}$$

and by $\bar{G}^{K^{-1}}_\Sigma \in M_{18}(\mathbb{C})$ the operator-valued Stieltjes transform of $\bar{Q}^{K^{-1}}_\Sigma$. Using (S196) and the definition of the operator-valued Stieltjes transform $G_{\bar{Q}^{K^{-1}}_{W,X,\Theta}+\bar{Q}^{K^{-1}}_\Sigma}$, we can write

$$\bar{G}^{K^{-1}} = \mathrm{id}_{18} \otimes \bar{\mathrm{tr}}\left(\bar{Z} - \bar{Q}^{K^{-1}}_{W,X,\Theta} - \bar{Q}^{K^{-1}}_\Sigma\right)^{-1} = G_{\bar{Q}^{K^{-1}}_{W,X,\Theta}+\bar{Q}^{K^{-1}}_\Sigma}(\bar{Z}) \tag{S200}$$

Thus using the subordinated form of the equations for addition of free variables (Mingo & Speicher, 2017; section 9.2 Thm. 11), and the defining equation for $\bar{G}^{K^{-1}}_\Sigma$, the operator-valued theory of free probability shows that in the limit $n_0, n_1, m \to \infty$, the Stieltjes transform $\bar{G}^{K^{-1}}$ satisfies the

following $18 \times 18$ matrix equation:

$$
\begin{aligned}
\bar{G}^{K^{-1}} &= \bar{G}_\Sigma^{K^{-1}}(\bar{Z} - \bar{R}_{W,X,\Theta}^{K^{-1}}(\bar{G}^{K^{-1}})) \\
&= \mathrm{id} \otimes \bar{\mathrm{tr}}\left(\bar{Z} - \bar{R}_{W,X,\Theta}^{K^{-1}}(\bar{G}^{K^{-1}}) - \bar{Q}_\Sigma^{K^{-1}}\right)^{-1},
\end{aligned}
\tag{S201}
$$

where $\bar{R}_{W,X,\Theta}^{K^{-1}}(\bar{G}^{K^{-1}}) \in M_{18}(\mathbb{C})$ is the operator-valued R-transform of $\bar{Q}_{W,X,\Theta}^{K^{-1}}$. Note that (S201) is a coupled set of $18 \times 18$ *scalar* equations and thus eliminates all reference to large random matrices. To see this, note that $\bar{Z}, \bar{G}^{K^{-1}}, \bar{R}_{W,X,\Theta}^{K^{-1}}(\bar{G}^{K^{-1}})$ are all scalar-entried $18 \times 18$ matrices. The right-hand side of (S201) is defined by expanding the inverse to obtain an $18 \times 18$ block matrix whose blocks involve various rational functions of $\Sigma, \Sigma_\beta$ and the scalar entries of $\bar{Z}, \bar{G}^{K^{-1}}, \bar{R}_{W,X,\Theta}^{K^{-1}}(\bar{G}^{K^{-1}})$. Finally one computes the normalized traces of these blocks, giving scalar values and eliminating all reference to random matrices. Below, when writing out these equations explicitly, we will use the fact that traces of rational functions of $\Sigma, \Sigma_\beta$ tend toward expectations of the corresponding rational functions over the LJSD $\mu$. Both here and in the sequel, to ease the already cumbersome presentation, we use $G^{K^{-1}}$ to also denote the limiting value satisfying (S201).

As described in (Adlam & Pennington, 2020a;b), since $\bar{Q}_{W,X,\Theta}^{K^{-1}}$ is a block matrix whose blocks are iid Gaussian matrices (and their transposes), an explicit expression for $\bar{R}_{W,X,\Theta}^{K^{-1}}(\bar{G}^{K^{-1}})$ can be obtained through a covariance map, denoted by $\eta$ (Far et al., 2006). In particular, $\eta : M_d(\mathbb{C}) \to M_d(\mathbb{C})$ is defined by,

$$
[\eta(D)]_{ij} = \sum_{kl} \sigma(i,k;l,j)\alpha_k D_{kl},
\tag{S202}
$$

where $\alpha_k$ is dimensionality of the $k$th block and $\sigma(i,k;l,k)$ denotes the covariance between the entries of the blocks $ij$ block of $\bar{Q}_{W,X,\Theta}^{K^{-1}}$ and entries of the $kl$ block of $\bar{Q}_{W,X,\Theta}^{K^{-1}}$. Here $d = 18$ is the number of blocks. When the constituent blocks are iid Gaussian matrices and their transposes, as is the case here, then $\bar{R}_{W,X,\Theta}^{K^{-1}} = \eta$ (Mingo & Speicher, 2017; section 9.1 and 9.2 Thm. 11), and therefore the entries of $\bar{R}_{W,X,\Theta}^{K^{-1}}$ can be read off from eqn. (S195). To simplify the presentation, we only report the entries of $\bar{R}_{W,X,\Theta}^{K^{-1}}(G^{K^{-1}})$ that are nonzero, given the specific sparsity pattern of $G^{K^{-1}}$. The latter follows from eqn. (S201) in the manner described in (Mingo & Speicher, 2017; Far et al., 2006). Practically speaking, the sparsity pattern can be obtained by iterating an eqn. (S201), starting with an ansatz sparsity pattern determined by $\bar{Z}$, and stopping when the iteration converges to a fixed sparsity pattern. In this case (and all cases that follow in the subsequent sections), the number of necessary iterations is small and can be done explicitly. We omit the details and instead simply report the following results for the nonzero entries:

$$
\bar{R}_{W,X,\Theta}^{K^{-1}}(\bar{G}^{K^{-1}}) = \begin{pmatrix} 0 & R_{W,X,\Theta}^{K^{-1}}(G^{K^{-1}})^\top \\ R_{W,X,\Theta}^{K^{-1}}(G^{K^{-1}}) & 0 \end{pmatrix},
\tag{S203}
$$

where,

$$
[R_{W,X,\Theta}^{K^{-1}}(G^{K^{-1}})]_{1,1} = \frac{G_{2,2}^{K^{-1}}(\zeta - \eta) - \sqrt{\rho}G_{6,3}^{K^{-1}}}{\gamma}
\tag{S204}
$$

$$
[R_{W,X,\Theta}^{K^{-1}}(G^{K^{-1}})]_{1,9} = -\frac{\sqrt{\rho}G_{8,3}^{K^{-1}}}{\gamma}
\tag{S205}
$$

$$
[R_{W,X,\Theta}^{K^{-1}}(G^{K^{-1}})]_{2,2} = \frac{\psi G_{1,1}^{K^{-1}}(\zeta - \eta)}{\gamma\phi} + \sqrt{\rho}\psi G_{4,5}^{K^{-1}}
\tag{S206}
$$

$$
[R_{W,X,\Theta}^{K^{-1}}(G^{K^{-1}})]_{4,5} = \sqrt{\rho}G_{2,2}^{K^{-1}}
\tag{S207}
$$

$$
[R_{W,X,\Theta}^{K^{-1}}(G^{K^{-1}})]_{6,3} = -\frac{\sqrt{\rho}G_{1,1}^{K^{-1}}}{\gamma\phi}
\tag{S208}
$$

$$
[R_{W,X,\Theta}^{K^{-1}}(G^{K^{-1}})]_{8,3} = -\frac{\sqrt{\rho}G_{1,9}^{K^{-1}}}{\gamma\phi},
\tag{S209}
$$

and the remaining entries of $R_{W,X,\Theta}^{K^{-1}}(G^{K^{-1}})$ are zero. Owing to the large degree of sparsity, the matrix inverse in (S201) can be performed explicitly and yields relatively simple expressions that depend on the entries of $G^{K^{-1}}$ and the matrices $\Sigma$ and $\Sigma_\beta$. For example, the $(16, 4)$ entry of the self-consistent equation reads,

$$G_{7,4}^{K^{-1}} = \left[\mathrm{id} \otimes \bar{\mathrm{tr}}\left(\bar{Z} - \bar{R}_{W,X,\Theta}^{K^{-1}}(\bar{G}^{K^{-1}}) - \bar{Q}_\Sigma^{K^{-1}}\right)^{-1}\right]_{16,4} \tag{S210}$$

$$= \bar{\mathrm{tr}}\left[-\frac{1}{\sqrt{\rho}}\Sigma_\beta\left(I_{n_0} + \frac{\rho}{\phi\gamma}G_{1,1}^{K^{-1}}G_{2,2}^{K^{-1}}\Sigma\right)^{-1}\right] \tag{S211}$$

$$\overset{n_0 \to \infty}{=} -\mathbb{E}_\mu\left[\frac{q/\sqrt{\rho}}{1 + \frac{x}{\phi}\lambda}\right] \tag{S212}$$

$$= -\frac{\mathcal{I}_{0,1}^\beta}{\sqrt{\rho}}, \tag{S213}$$

where to compute the asymptotic normalized trace we moved to an eigenbasis of $\Sigma$ and recalled the definition of the LJSD $\mu$ and the definition of $\mathcal{I}^\beta$ in Eq. (12). The remaining entries of the (S201) can be obtained in a similar manner and together yield the following set of coupled equations for the entries of $G^{K^{-1}}$,

$$G_{1,1}^{K^{-1}} = -\frac{\gamma}{-G_{2,2}^{K^{-1}}(-\zeta + \eta + \rho) + \rho G_{2,2}^{K^{-1}} - \sqrt{\rho}G_{6,3}^{K^{-1}} - \gamma} \tag{S214}$$

$$G_{2,2}^{K^{-1}} = \frac{\gamma\phi}{\psi G_{1,1}^{K^{-1}}(\eta - \zeta) - \gamma\phi\left(\sqrt{\rho}\psi G_{4,5}^{K^{-1}} - 1\right)} \tag{S215}$$

$$G_{3,6}^{K^{-1}} = \mathbb{E}_\mu\left[\frac{\sqrt{\rho}G_{1,1}^{K^{-1}}}{-\lambda\rho G_{1,1}^{K^{-1}}G_{2,2}^{K^{-1}} - \gamma\phi}\right] \tag{S216}$$

$$G_{4,5}^{K^{-1}} = \mathbb{E}_\mu\left[\frac{\lambda\sqrt{\rho}G_{1,1}^{K^{-1}}}{-\lambda\rho G_{1,1}^{K^{-1}}G_{2,2}^{K^{-1}} - \gamma\phi}\right] \tag{S217}$$

$$G_{5,4}^{K^{-1}} = \mathbb{E}_\mu\left[-\frac{\gamma\sqrt{\rho}\phi G_{2,2}^{K^{-1}}}{-\lambda\rho G_{1,1}^{K^{-1}}G_{2,2}^{K^{-1}} - \gamma\phi}\right] \tag{S218}$$

$$G_{6,3}^{K^{-1}} = \mathbb{E}_\mu\left[-\frac{\gamma\lambda\sqrt{\rho}\phi G_{2,2}^{K^{-1}}}{-\lambda\rho G_{1,1}^{K^{-1}}G_{2,2}^{K^{-1}} - \gamma\phi}\right] \tag{S219}$$

$$G_{7,4}^{K^{-1}} = \mathbb{E}_\mu\left[-\frac{q\gamma\phi}{\sqrt{\rho}\left(\lambda\rho G_{1,1}^{K^{-1}}G_{2,2}^{K^{-1}} + \gamma\phi\right)}\right] \tag{S220}$$

$$G_{7,6}^{K^{-1}} = \mathbb{E}_\mu\left[\frac{q\sqrt{\lambda}G_{1,1}^{K^{-1}}}{\lambda\rho G_{1,1}^{K^{-1}}G_{2,2}^{K^{-1}} + \gamma\phi}\right] \tag{S221}$$

$$G_{8,3}^{K^{-1}} = \mathbb{E}_\mu\left[-\frac{q\gamma\lambda\phi}{\sqrt{\rho}\left(\lambda\rho G_{1,1}^{K^{-1}}G_{2,2}^{K^{-1}} + \gamma\phi\right)}\right] \tag{S222}$$

$$G_{8,5}^{K^{-1}} = \mathbb{E}_\mu\left[\frac{q\lambda^{3/2}G_{1,1}^{K^{-1}}}{\lambda\rho G_{1,1}^{K^{-1}}G_{2,2}^{K^{-1}} + \gamma\phi}\right] \tag{S223}$$

$$G_{8,7}^{K^{-1}} = \mathbb{E}_\mu\left[\sqrt{\lambda}\right] \tag{S224}$$

$$G_{9,1}^{K^{-1}} = \frac{\sqrt{\rho}G_{8,3}^{K^{-1}}}{-G_{2,2}^{K^{-1}}(-\zeta + \eta + \rho) + \rho G_{2,2}^{K^{-1}} - \sqrt{\rho}G_{6,3}^{K^{-1}} - \gamma} \tag{S225}$$

$$G_{3,4}^{K^{-1}} = G_{5,6}^{K^{-1}} = \mathbb{E}_\mu\left[\frac{\sqrt{\lambda}\rho G_{1,1}^{K^{-1}}G_{2,2}^{K^{-1}}}{-\lambda\rho G_{1,1}^{K^{-1}}G_{2,2}^{K^{-1}} - \gamma\phi}\right] \tag{S226}$$

$$G_{3,5}^{K^{-1}} = G_{4,6}^{K^{-1}} = \mathbb{E}_\mu\left[-\frac{\sqrt{\lambda}\sqrt{\rho}G_{1,1}^{K^{-1}}}{\lambda\rho G_{1,1}^{K^{-1}}G_{2,2}^{K^{-1}} + \gamma\phi}\right] \tag{S227}$$

$$G_{4,3}^{K^{-1}} = G_{6,5}^{K^{-1}} = \mathbb{E}_\mu\left[ - \frac{\gamma\sqrt{\lambda}\phi}{-\lambda\rho G_{1,1}^{K^{-1}} G_{2,2}^{K^{-1}} - \gamma\phi} \right] \tag{S228}$$

$$G_{5,3}^{K^{-1}} = G_{6,4}^{K^{-1}} = \mathbb{E}_\mu\left[ \frac{\gamma\sqrt{\lambda}\sqrt{\rho}\phi G_{2,2}^{K^{-1}}}{\lambda\rho G_{1,1}^{K^{-1}} G_{2,2}^{K^{-1}} + \gamma\phi} \right] \tag{S229}$$

$$G_{7,3}^{K^{-1}} = G_{8,4}^{K^{-1}} = \mathbb{E}_\mu\left[ - \frac{q\gamma\sqrt{\lambda}\phi}{\sqrt{\rho}\left(\lambda\rho G_{1,1}^{K^{-1}} G_{2,2}^{K^{-1}} + \gamma\phi\right)} \right] \tag{S230}$$

$$G_{7,5}^{K^{-1}} = G_{8,6}^{K^{-1}} = \mathbb{E}_\mu\left[ \frac{q\lambda G_{1,1}^{K^{-1}}}{\lambda\rho G_{1,1}^{K^{-1}} G_{2,2}^{K^{-1}} + \gamma\phi} \right] \tag{S231}$$

$$G_{7,7}^{K^{-1}} = G_{8,8}^{K^{-1}} = G_{9,9}^{K^{-1}} = 1 \tag{S232}$$

$$G_{3,3}^{K^{-1}} = G_{4,4}^{K^{-1}} = G_{5,5}^{K^{-1}} = G_{6,6}^{K^{-1}} = \mathbb{E}_\mu\left[ - \frac{\gamma\phi}{-\lambda\rho G_{1,1}^{K^{-1}} G_{2,2}^{K^{-1}} - \gamma\phi} \right], \tag{S233}$$

where we have used the fact that, asymptotically, the normalized trace becomes equivalent to an expectation over $\mu$. After eliminating $G_{6,3}^{K^{-1}}$ and $G_{4,5}^{K^{-1}}$ from the first two equations, it is straightforward to show that

$$\tau_1 \equiv \bar{\mathrm{tr}}(K^{-1}) = \frac{1}{\gamma}G_{1,1}^{K^{-1}} = \frac{\sqrt{(\psi-\phi)^2 + 4x\psi\phi\gamma/\rho} + \psi - \phi}{2\psi\gamma} \tag{S234}$$

$$\bar{\tau}_1 \equiv \bar{\mathrm{tr}}(\hat{K}^{-1}) = \frac{1}{\gamma}G_{2,2}^{K^{-1}} = \frac{1}{\gamma} + \frac{\psi}{\phi}\left(\tau_1 - \frac{1}{\gamma}\right) \tag{S235}$$

$$\tau_2 = \bar{\mathrm{tr}}(\frac{1}{n_0}X^\top\Sigma^{1/2}\Sigma_\beta\Sigma^{1/2}XK^{-1}) = \tau_1\mathcal{I}_{1,1}^\beta \tag{S236}$$

where we have used the notation $\tau_1$ and $\tau_2$ from (Adlam & Pennington, 2020a;b), and $\bar{\tau}_1$ is the companion transform of $\tau_1$, and where $x$ satisfies the self-consistent equation,

$$x = \frac{1 - \gamma\tau_1}{\omega + \mathcal{I}_{1,1}} = \frac{1 - \frac{\sqrt{(\psi-\phi)^2 + 4x\psi\phi\gamma/\rho} + \psi - \phi}{2\psi}}{\omega + \mathcal{I}_{1,1}}. \tag{S237}$$

Here we utilized the two-index set of functionals of $\mu$, $\mathcal{I}_{a,b}$ defined in Eq. (12).

Note that the product $\tau_1\bar{\tau}_1$ is simply related to $x$,

$$x = \gamma\rho\tau_1\bar{\tau}_1, \tag{S238}$$

so that, given $x$, the equations for the remaining entries of $G^{K^{-1}}$ completely decouple. In particular,

$$G_{3,6}^{K^{-1}} = -\frac{\sqrt{\rho}G_{1,1}^{K^{-1}}\mathcal{I}_{0,1}}{\gamma\phi} \tag{S239}$$

$$G_{4,5}^{K^{-1}} = -\frac{\sqrt{\rho}G_{1,1}^{K^{-1}}\mathcal{I}_{1,1}}{\gamma\phi} \tag{S240}$$

$$G_{5,4}^{K^{-1}} = \sqrt{\rho}G_{2,2}^{K^{-1}}\mathcal{I}_{0,1} \tag{S241}$$

$$G_{6,3}^{K^{-1}} = \sqrt{\rho}G_{2,2}^{K^{-1}}\mathcal{I}_{1,1} \tag{S242}$$

$$G_{7,4}^{K^{-1}} = -\frac{\mathcal{I}_{0,1}^\beta}{\sqrt{\rho}} \tag{S243}$$

$$G_{7,6}^{K^{-1}} = \frac{\mathcal{I}_{\frac{1}{2},1}^\beta G_{1,1}^{K^{-1}}}{\gamma\phi} \tag{S244}$$

$$G_{8,3}^{K^{-1}} = -\frac{\mathcal{I}_{1,1}^\beta}{\sqrt{\rho}} \tag{S245}$$

$$G_{8,5}^{K^{-1}} = \frac{\mathcal{I}_{\frac{3}{2},1}^{\beta} G_{1,1}^{K^{-1}}}{\gamma\phi} \tag{S246}$$

$$G_{8,7}^{K^{-1}} = \frac{\mathcal{I}_{\frac{1}{2},0}}{\phi} \tag{S247}$$

$$G_{9,1}^{K^{-1}} = -\frac{\sqrt{\rho}G_{1,1}^{K^{-1}}G_{8,3}^{K^{-1}}}{\gamma} \tag{S248}$$

$$G_{3,4}^{K^{-1}} = G_{5,6}^{K^{-1}} = -\frac{x\mathcal{I}_{\frac{1}{2},1}}{\phi} \tag{S249}$$

$$G_{3,5}^{K^{-1}} = G_{4,6}^{K^{-1}} = -\frac{\sqrt{\rho}G_{1,1}^{K^{-1}}\mathcal{I}_{\frac{1}{2},1}}{\gamma\phi} \tag{S250}$$

$$G_{4,3}^{K^{-1}} = G_{6,5}^{K^{-1}} = \mathcal{I}_{\frac{1}{2},1} \tag{S251}$$

$$G_{5,3}^{K^{-1}} = G_{6,4}^{K^{-1}} = \sqrt{\rho}G_{2,2}^{K^{-1}}\mathcal{I}_{\frac{1}{2},1} \tag{S252}$$

$$G_{7,3}^{K^{-1}} = G_{8,4}^{K^{-1}} = -\frac{\mathcal{I}_{\frac{1}{2},1}^{\beta}}{\sqrt{\rho}} \tag{S253}$$

$$G_{7,5}^{K^{-1}} = G_{8,6}^{K^{-1}} = \frac{\mathcal{I}_{1,1}^{\beta} G_{1,1}^{K^{-1}}}{\gamma\phi} \tag{S254}$$

$$G_{7,7}^{K^{-1}} = G_{8,8}^{K^{-1}} = G_{9,9}^{K^{-1}} = 1 \tag{S255}$$

$$G_{3,3}^{K^{-1}} = G_{4,4}^{K^{-1}} = G_{5,5}^{K^{-1}} = G_{6,6}^{K^{-1}} = \mathcal{I}_{0,1}, \tag{S256}$$

which will be important intermediate results for the subsequent sections.

Finally, we note that these results are sufficient to compute the training error. The expected training loss can be written as,

$$E_{\text{train}} = \frac{1}{m}\mathbb{E}\text{tr}\big((Y - \hat{y}(X))(Y - \hat{y}(X))^{\top}\big) \tag{S257}$$

$$= \frac{\gamma^2}{m}\mathbb{E}\text{tr}\big(Y^{\top}YK^{-2}\big) \tag{S258}$$

$$= \frac{\gamma^2}{m}\mathbb{E}\text{tr}\big(\frac{1}{n_0}(X^{\top}\Sigma^{1/2}\Sigma_\beta\Sigma^{1/2}X + \sigma_\varepsilon^2 I_m)K^{-2}\big) \tag{S259}$$

$$= -\gamma^2\left(\partial_\gamma\tau_2 + \sigma_\varepsilon^2\partial_\gamma\tau_1\right) \tag{S260}$$

$$= -\gamma^2\left(\partial_\gamma(\tau_1\mathcal{I}_{1,1}^{\beta}) + \sigma_\varepsilon^2\partial_\gamma\tau_1\right). \tag{S261}$$

### G.4.2   $E_{21}$

The calculation of $E_{21}$ proceeds exactly as in (Tripuraneni et al., 2021a;b) with the simple modification of including an additional factor $\Sigma_\beta$ inside the final trace term, yielding

$$E_{21} = -2\frac{x}{\phi}\mathcal{I}_{2,1}^{\beta}. \tag{S262}$$

### G.4.3   $E_{31}$

The calculation of $E_{31}$ proceeds exactly as in (Tripuraneni et al., 2021a;b) with no modifications since there is no dependence on $\Sigma_\beta$. The result is,

$$E_{31} = -\rho\frac{\psi}{\phi}\frac{\partial x}{\partial\gamma}\left(\sigma_\varepsilon^2\big((\omega + \phi\mathcal{I}_{1,2})(\omega + \mathcal{I}_{1,1}) + \frac{\phi}{\psi}\gamma\bar{\tau}_1\mathcal{I}_{2,2}\big)\right), \tag{S263}$$

### G.4.4 $E_{32}$

Define the block matrix $Q^{E_{32}} \equiv [Q_1^{E_{32}} \ Q_2^{E_{32}}]$ by,

$$Q_1^{E_{32}} = \begin{pmatrix}
I_m & \frac{\sqrt{\eta-\zeta}\Theta^\top}{\gamma\sqrt{n_1}} & \frac{\sqrt{\rho}X^\top}{\gamma\sqrt{n_0}} & 0 & 0 & 0 & \frac{\sqrt{\eta-\zeta}\Theta^\top(\zeta-\eta)}{\gamma\sqrt{n_1}} & 0 \\
-\frac{\Theta\sqrt{\eta-\zeta}}{\sqrt{n_1}} & I_{n_1} & 0 & 0 & -\frac{\sqrt{\rho}W}{\sqrt{n_1}} & 0 & 0 & 0 \\
0 & 0 & I_{n_0} & -\Sigma^{1/2} & 0 & 0 & 0 & \Sigma^{1/2}(\eta-\zeta) \\
0 & -\frac{W^\top}{\sqrt{n_1}} & 0 & I_{n_0} & 0 & 0 & 0 & 0 \\
0 & 0 & 0 & 0 & I_{n_0} & -\Sigma^{1/2} & 0 & \frac{n_1\Sigma\rho}{n_0\sqrt{\rho}} \\
-\frac{X}{\sqrt{n_0}} & 0 & 0 & 0 & 0 & I_{n_0} & 0 & 0 \\
0 & 0 & 0 & 0 & 0 & 0 & I_{n_1} & 0 \\
0 & 0 & 0 & 0 & 0 & 0 & -\frac{W^\top}{\sqrt{n_1}} & I_{n_0} \\
0 & 0 & 0 & 0 & 0 & 0 & \frac{\sqrt{\eta-\zeta}\Theta^\top}{\gamma\sqrt{n_1}} & 0 \\
0 & 0 & 0 & 0 & 0 & 0 & 0 & 0 \\
0 & 0 & 0 & 0 & 0 & 0 & 0 & 0 \\
0 & 0 & 0 & 0 & 0 & 0 & 0 & 0 \\
0 & 0 & 0 & 0 & 0 & 0 & -\frac{W^\top}{\sqrt{n_1}} & 0 \\
0 & 0 & 0 & 0 & 0 & 0 & 0 & 0 \\
0 & 0 & 0 & 0 & 0 & 0 & 0 & 0 \\
0 & 0 & 0 & 0 & 0 & 0 & 0 & 0
\end{pmatrix}, \tag{S264}$$

and,

$$Q_2^{E_{32}} = \begin{pmatrix}
0 & 0 & 0 & 0 & 0 & 0 & 0 & 0 \\
0 & 0 & 0 & 0 & 0 & 0 & 0 & 0 \\
0 & 0 & 0 & 0 & 0 & 0 & 0 & 0 \\
0 & 0 & 0 & 0 & 0 & 0 & 0 & 0 \\
0 & 0 & 0 & 0 & 0 & 0 & 0 & 0 \\
0 & 0 & 0 & 0 & 0 & 0 & 0 & 0 \\
-\frac{\Theta\sqrt{\eta-\zeta}}{\sqrt{n_1}} & -\frac{\sqrt{\rho}W}{\sqrt{n_1}} & 0 & 0 & 0 & 0 & 0 & 0 \\
0 & 0 & 0 & 0 & 0 & 0 & 0 & 0 \\
I_m & 0 & 0 & \frac{\sqrt{\rho}X^\top}{\gamma\sqrt{n_0}} & 0 & 0 & 0 & 0 \\
0 & I_{n_0} & -\Sigma^{1/2} & 0 & 0 & 0 & 0 & 0 \\
-\frac{X}{\sqrt{n_0}} & 0 & I_{n_0} & 0 & 0 & 0 & 0 & 0 \\
0 & 0 & 0 & I_{n_0} & -\Sigma^{1/2} & 0 & 0 & 0 \\
0 & 0 & 0 & 0 & I_{n_0} & \frac{\Sigma_\beta}{\sqrt{\rho}} & 0 & 0 \\
0 & 0 & 0 & 0 & 0 & I_{n_0} & -\Sigma^{1/2} & 0 \\
0 & 0 & 0 & 0 & 0 & 0 & I_{n_0} & -\frac{X}{\sqrt{n_0}} \\
0 & 0 & 0 & 0 & 0 & 0 & 0 & I_m
\end{pmatrix}. \tag{S265}$$

Then block matrix inversion (i.e. repeated applications of the Schur complement formula) shows that,

$$G_{8,8}^{E_{32}} = G_{14,14}^{E_{32}} = G_{15,15}^{E_{32}} = G_{16,16}^{E_{32}} = 1 \tag{S266}$$

$$G_{1,1}^{E_{32}} = G_{9,9}^{E_{32}} = G_{1,1}^{K^{-1}} \tag{S267}$$

$$G_{2,2}^{E_{32}} = G_{7,7}^{E_{32}} = G_{2,2}^{K^{-1}} \tag{S268}$$

$$G_{13,8}^{E_{32}} = G_{3,3}^{K^{-1}} - 1 \tag{S269}$$

$$G_{3,3}^{E_{32}} = G_{6,6}^{E_{32}} = G_{11,11}^{E_{32}} = G_{12,12}^{E_{32}} = G_{4,4}^{E_{32}} = G_{5,5}^{E_{32}} = G_{10,10}^{E_{32}} = G_{13,13}^{E_{32}} = G_{3,3}^{K^{-1}} \tag{S270}$$

$$G_{3,4}^{E_{32}} = G_{5,6}^{E_{32}} = G_{10,11}^{E_{32}} = G_{12,8}^{E_{32}} = G_{12,13}^{E_{32}} = G_{3,4}^{K^{-1}} \tag{S271}$$

$$G_{3,5}^{E_{32}} = G_{4,6}^{E_{32}} = G_{12,10}^{E_{32}} = G_{13,11}^{E_{32}} = G_{3,5}^{K^{-1}} \tag{S272}$$

$$G_{3,6}^{E_{32}} = G_{12,11}^{E_{32}} = G_{3,6}^{K^{-1}} \tag{S273}$$

$$G_{4,3}^{E_{32}} = G_{6,5}^{E_{32}} = G_{11,10}^{E_{32}} = G_{13,12}^{E_{32}} = G_{4,3}^{K^{-1}} \tag{S274}$$

$$G_{4,5}^{E_{32}} = G_{13,10}^{E_{32}} = G_{4,5}^{K^{-1}} \tag{S275}$$

$$G_{5,3}^{E_{32}} = G_{6,4}^{E_{32}} = G_{10,12}^{E_{32}} = G_{11,8}^{E_{32}} = G_{11,13}^{E_{32}} = G_{5,3}^{K^{-1}} \tag{S276}$$

$$G_{5,4}^{E_{32}} = G_{10,8}^{E_{32}} = G_{10,13}^{E_{32}} = G_{5,4}^{K^{-1}} \tag{S277}$$

$$G_{6,3}^{E_{32}} = G_{11,12}^{E_{32}} = G_{6,3}^{K^{-1}} \tag{S278}$$

$$G_{14,12}^{E_{32}} = G_{15,13}^{E_{32}} = G_{7,3}^{K^{-1}} \tag{S279}$$

$$G_{14,13}^{E_{32}} = G_{7,4}^{K^{-1}} \tag{S280}$$

$$G_{14,11}^{E_{32}} = G_{7,6}^{K^{-1}} \tag{S281}$$

$$G_{15,12}^{E_{32}} = G_{8,3}^{K^{-1}} \tag{S282}$$

$$G_{15,10}^{E_{32}} = G_{8,5}^{K^{-1}} \tag{S283}$$

$$G_{15,14}^{E_{32}} = G_{8,7}^{K^{-1}} \tag{S284}$$

$$G_{16,9}^{E_{32}} = G_{9,1}^{K^{-1}} \tag{S285}$$

$$G_{14,10}^{E_{32}} = G_{15,11}^{E_{32}} = \frac{G_{9,1}^{K^{-1}}}{\phi} \tag{S286}$$

$$G_{16,1}^{E_{32}} = \frac{\phi}{\psi} E_{32}, \tag{S287}$$

where $G_{i,j}^{E_{32}}$ denotes the normalized trace of the $(i,j)$-block of the inverse of $\left(Q^{E_{32}}\right)^\top$. For brevity, we have suppressed the expressions for the other non-zero blocks.

To compute the limiting values of these traces, we require the asymptotic block-wise traces of $Q^{E_{32}}$, which may be determined from the operator-valued Stieltjes transform. To proceed, we first augment $Q^{E_{32}}$ to form the the self-adjoint matrix $\bar{Q}^{E_{32}}$,

$$\bar{Q}^{E_{32}} = \begin{pmatrix} 0 & [Q^{E_{32}}]^\top \\ Q^{E_{32}} & 0 \end{pmatrix}. \tag{S288}$$

and observe that we can write $\bar{Q}^{E_{32}}$ as,

$$\begin{aligned} \bar{Q}^{E_{32}} &= \bar{Z} - \bar{Q}_{W,X,\Theta}^{E_{32}} - \bar{Q}_\Sigma^{E_{32}} \\ &= \begin{pmatrix} 0 & I_{16} \\ I_{16} & 0 \end{pmatrix} - \begin{pmatrix} 0 & [Q_{W,X,\Theta}^{E_{32}}]^\top \\ Q_{W,X,\Theta}^{E_{32}} & 0 \end{pmatrix} - \begin{pmatrix} 0 & [Q_\Sigma^{E_{32}}]^\top \\ Q_\Sigma^{E_{32}} & 0 \end{pmatrix}, \end{aligned} \tag{S289}$$

where $Q_{W,X,\theta}^{E_{32}} \equiv [[Q_{W,X,\theta}^{E_{32}}]_1 \ [Q_{W,X,\theta}^{E_{32}}]_2]$ and,

$$[Q_{W,X,\theta}^{E_{32}}]_1 = - \begin{pmatrix} 0 & \frac{\sqrt{\eta-\zeta}\Theta^\top}{\gamma\sqrt{n_1}} & \frac{\sqrt{\rho}X^\top}{\gamma\sqrt{n_0}} & 0 & 0 & 0 & \frac{\sqrt{\eta-\zeta}\Theta^\top(\zeta-\eta)}{\gamma\sqrt{n_1}} & 0 \\ -\frac{\Theta\sqrt{\eta-\zeta}}{\sqrt{n_1}} & 0 & 0 & 0 & -\frac{\sqrt{\rho}W}{\sqrt{n_1}} & 0 & 0 & 0 \\ 0 & 0 & 0 & 0 & 0 & 0 & 0 & 0 \\ 0 & -\frac{W^\top}{\sqrt{n_1}} & 0 & 0 & 0 & 0 & 0 & 0 \\ 0 & 0 & 0 & 0 & 0 & 0 & 0 & 0 \\ -\frac{X}{\sqrt{n_0}} & 0 & 0 & 0 & 0 & 0 & 0 & 0 \\ 0 & 0 & 0 & 0 & 0 & 0 & 0 & 0 \\ 0 & 0 & 0 & 0 & 0 & 0 & -\frac{W^\top}{\sqrt{n_1}} & 0 \\ 0 & 0 & 0 & 0 & 0 & 0 & \frac{\sqrt{\eta-\zeta}\Theta^\top}{\gamma\sqrt{n_1}} & 0 \\ 0 & 0 & 0 & 0 & 0 & 0 & 0 & 0 \\ 0 & 0 & 0 & 0 & 0 & 0 & 0 & 0 \\ 0 & 0 & 0 & 0 & 0 & 0 & 0 & 0 \\ 0 & 0 & 0 & 0 & 0 & 0 & -\frac{W^\top}{\sqrt{n_1}} & 0 \\ 0 & 0 & 0 & 0 & 0 & 0 & 0 & 0 \\ 0 & 0 & 0 & 0 & 0 & 0 & 0 & 0 \\ 0 & 0 & 0 & 0 & 0 & 0 & 0 & 0 \end{pmatrix} \tag{S290}$$

$$[Q_{W,X,\theta}^{E_{32}}]_2 = - \begin{pmatrix} 0 & 0 & 0 & 0 & 0 & 0 & 0 & 0 \\ 0 & 0 & 0 & 0 & 0 & 0 & 0 & 0 \\ 0 & 0 & 0 & 0 & 0 & 0 & 0 & 0 \\ 0 & 0 & 0 & 0 & 0 & 0 & 0 & 0 \\ 0 & 0 & 0 & 0 & 0 & 0 & 0 & 0 \\ 0 & 0 & 0 & 0 & 0 & 0 & 0 & 0 \\ -\frac{\Theta\sqrt{\eta-\zeta}}{\sqrt{n_1}} & -\frac{\sqrt{\rho}W}{\sqrt{n_1}} & 0 & 0 & 0 & 0 & 0 & 0 \\ 0 & 0 & 0 & 0 & 0 & 0 & 0 & 0 \\ 0 & 0 & 0 & \frac{\sqrt{\rho}X^\top}{\gamma\sqrt{n_0}} & 0 & 0 & 0 & 0 \\ 0 & 0 & 0 & 0 & 0 & 0 & 0 & 0 \\ -\frac{X}{\sqrt{n_0}} & 0 & 0 & 0 & 0 & 0 & 0 & 0 \\ 0 & 0 & 0 & 0 & 0 & 0 & 0 & 0 \\ 0 & 0 & 0 & 0 & 0 & 0 & 0 & 0 \\ 0 & 0 & 0 & 0 & 0 & 0 & 0 & 0 \\ 0 & 0 & 0 & 0 & 0 & 0 & 0 & -\frac{X}{\sqrt{n_0}} \\ 0 & 0 & 0 & 0 & 0 & 0 & 0 & 0 \end{pmatrix} \tag{S291}$$

$$Q_\Sigma^{E_{32}} = - \begin{pmatrix} 0 & 0 & 0 & 0 & 0 & 0 & 0 & 0 & 0 & 0 & 0 & 0 & 0 & 0 & 0 & 0 \\ 0 & 0 & 0 & 0 & 0 & 0 & 0 & 0 & 0 & 0 & 0 & 0 & 0 & 0 & 0 & 0 \\ 0 & 0 & 0 & -\Sigma^{1/2} & 0 & 0 & 0 & \Sigma^{1/2}(\eta-\zeta) & 0 & 0 & 0 & 0 & 0 & 0 & 0 & 0 \\ 0 & 0 & 0 & 0 & 0 & 0 & 0 & 0 & 0 & 0 & 0 & 0 & 0 & 0 & 0 & 0 \\ 0 & 0 & 0 & 0 & 0 & -\Sigma^{1/2} & 0 & \frac{n_1\Sigma\rho}{n_0\sqrt{\rho}} & 0 & 0 & 0 & 0 & 0 & 0 & 0 & 0 \\ 0 & 0 & 0 & 0 & 0 & 0 & 0 & 0 & 0 & 0 & 0 & 0 & 0 & 0 & 0 & 0 \\ 0 & 0 & 0 & 0 & 0 & 0 & 0 & 0 & 0 & 0 & 0 & 0 & 0 & 0 & 0 & 0 \\ 0 & 0 & 0 & 0 & 0 & 0 & 0 & 0 & 0 & 0 & 0 & 0 & 0 & 0 & 0 & 0 \\ 0 & 0 & 0 & 0 & 0 & 0 & 0 & 0 & 0 & 0 & -\Sigma^{1/2} & 0 & 0 & 0 & 0 & 0 \\ 0 & 0 & 0 & 0 & 0 & 0 & 0 & 0 & 0 & 0 & 0 & 0 & 0 & 0 & 0 & 0 \\ 0 & 0 & 0 & 0 & 0 & 0 & 0 & 0 & 0 & 0 & 0 & 0 & -\Sigma^{1/2} & 0 & 0 & 0 \\ 0 & 0 & 0 & 0 & 0 & 0 & 0 & 0 & 0 & 0 & 0 & 0 & 0 & \frac{\Sigma_\beta}{\sqrt{\rho}} & 0 & 0 \\ 0 & 0 & 0 & 0 & 0 & 0 & 0 & 0 & 0 & 0 & 0 & 0 & 0 & 0 & -\Sigma^{1/2} & 0 \\ 0 & 0 & 0 & 0 & 0 & 0 & 0 & 0 & 0 & 0 & 0 & 0 & 0 & 0 & 0 & 0 \\ 0 & 0 & 0 & 0 & 0 & 0 & 0 & 0 & 0 & 0 & 0 & 0 & 0 & 0 & 0 & 0 \end{pmatrix}.$$
(S292)

The operator-valued Stieltjes transforms satisfy,

$$\bar{G}^{E_{32}} = \bar{G}_\Sigma^{E_{32}}(\bar{Z} - \bar{R}_{W,X,\Theta}^{E_{32}}(\bar{G}^{E_{32}}))$$
$$= \mathrm{id} \otimes \bar{\mathrm{tr}}\left(\bar{Z} - \bar{R}_{W,X,\Theta}^{E_{32}}(\bar{G}^{E_{32}}) - \bar{Q}_\Sigma^{E_{32}}\right)^{-1},$$
(S293)

where $\bar{R}_{W,X,\Theta}^{E_{32}}(\bar{G}^{E_{32}})$ is the operator-valued R-transform of $\bar{Q}_{W,X,\Theta}^{E_{32}}$. As discussed above, since $\bar{Q}_{W,X,\Theta}^{E_{32}}$ is a block matrix whose blocks are iid Gaussian matrices (and their transposes), an explicit expression for $\bar{R}_{W,X,\Theta}^{E_{32}}(\bar{G}^{E_{32}})$ can be obtained from the covariance map $\eta$, which can be read off from eqn. (S288). As above, we utilize the specific sparsity pattern for $G^{E_{32}}$ that is induced by Eq. (S293), to obtain,

$$\bar{R}_{W,X,\Theta}^{E_{32}}(\bar{G}^{E_{32}}) = \begin{pmatrix} 0 & R_{W,X,\Theta}^{E_{32}}(G^{E_{32}})^\top \\ R_{W,X,\Theta}^{E_{32}}(G^{E_{32}}) & 0 \end{pmatrix},$$
(S294)

where,

$$[R_{W,X,\theta}^{E_{32}}(G^{E_{32}})]_{1,1} = \frac{G_{2,2}^{E_{32}}(\zeta-\eta)}{\gamma} - \frac{\sqrt{\rho}G_{6,3}^{E_{32}}}{\gamma} + \frac{G_{2,7}^{E_{32}}(\zeta-\eta)(\zeta-\eta)}{\gamma}$$
(S295)

$$[R_{W,X,\theta}^{E_{32}}(G^{E_{32}})]_{1,9} = \frac{G_{7,2}^{E_{32}}(\zeta-\eta)}{\gamma} - \frac{\sqrt{\rho}G_{11,3}^{E_{32}}}{\gamma} + \frac{G_{7,7}^{E_{32}}(\zeta-\eta)(\zeta-\eta)}{\gamma}$$
(S296)

$$[R_{W,X,\theta}^{E_{32}}(G^{E_{32}})]_{1,16} = -\frac{\sqrt{\rho}G_{15,3}^{E_{32}}}{\gamma}$$
(S297)

$$[R_{W,X,\theta}^{E_{32}}(G^{E_{32}})]_{2,2} = \frac{\psi G_{1,1}^{E_{32}}(\zeta-\eta)}{\gamma\phi} + \sqrt{\rho}\psi G_{4,5}^{E_{32}}$$
(S298)

$$[R_{W,X,\theta}^{E_{32}}(G^{E_{32}})]_{2,7} = \frac{\psi G_{9,1}^{E_{32}}(\zeta-\eta)}{\gamma\phi} + \sqrt{\rho}\psi G_{8,5}^{E_{32}} + \sqrt{\rho}\psi G_{13,5}^{E_{32}} + \frac{\psi G_{1,1}^{E_{32}}(\zeta-\eta)(\zeta-\eta)}{\gamma\phi}$$
(S299)

$$[R_{W,X,\theta}^{E_{32}}(G^{E_{32}})]_{4,5} = \sqrt{\rho}G_{2,2}^{E_{32}}$$
(S300)

$$[R_{W,X,\theta}^{E_{32}}(G^{E_{32}})]_{4,10} = \sqrt{\rho}G_{7,2}^{E_{32}}$$
(S301)

$$[R_{W,X,\theta}^{E_{32}}(G^{E_{32}})]_{6,3} = -\frac{\sqrt{\rho}G_{1,1}^{E_{32}}}{\gamma\phi}$$
(S302)

$$[R_{W,X,\theta}^{E_{32}}(G^{E_{32}})]_{6,12} = -\frac{\sqrt{\rho}G_{9,1}^{E_{32}}}{\gamma\phi}$$
(S303)

$$[R_{W,X,\theta}^{E_{32}}(G^{E_{32}})]_{7,2} = \frac{\psi G_{1,9}^{E_{32}}(\zeta-\eta)}{\gamma\phi} + \sqrt{\rho}\psi G_{4,10}^{E_{32}}$$
(S304)

$$[R_{W,X,\theta}^{E_{32}}(G^{E_{32}})]_{7,7} = \frac{\psi G_{9,9}^{E_{32}}(\zeta-\eta)}{\gamma\phi} + \sqrt{\rho}\psi G_{8,10}^{E_{32}} + \sqrt{\rho}\psi G_{13,10}^{E_{32}} + \frac{\psi G_{1,9}^{E_{32}}(\zeta-\eta)(\zeta-\eta)}{\gamma\phi}$$
(S305)

$$[R_{W,X,\theta}^{E_{32}}(G^{E_{32}})]_{8,5} = \sqrt{\rho}G_{2,7}^{E_{32}} \tag{S306}$$

$$[R_{W,X,\theta}^{E_{32}}(G^{E_{32}})]_{8,10} = \sqrt{\rho}G_{7,7}^{E_{32}} \tag{S307}$$

$$[R_{W,X,\theta}^{E_{32}}(G^{E_{32}})]_{9,1} = \frac{G_{2,7}^{E_{32}}(\zeta - \eta)}{\gamma} - \frac{\sqrt{\rho}G_{6,12}^{E_{32}}}{\gamma} \tag{S308}$$

$$[R_{W,X,\theta}^{E_{32}}(G^{E_{32}})]_{9,9} = \frac{G_{7,7}^{E_{32}}(\zeta - \eta)}{\gamma} - \frac{\sqrt{\rho}G_{11,12}^{E_{32}}}{\gamma} \tag{S309}$$

$$[R_{W,X,\theta}^{E_{32}}(G^{E_{32}})]_{9,16} = -\frac{\sqrt{\rho}G_{15,12}^{E_{32}}}{\gamma} \tag{S310}$$

$$[R_{W,X,\theta}^{E_{32}}(G^{E_{32}})]_{11,3} = -\frac{\sqrt{\rho}G_{1,9}^{E_{32}}}{\gamma\phi} \tag{S311}$$

$$[R_{W,X,\theta}^{E_{32}}(G^{E_{32}})]_{11,12} = -\frac{\sqrt{\rho}G_{9,9}^{E_{32}}}{\gamma\phi} \tag{S312}$$

$$[R_{W,X,\theta}^{E_{32}}(G^{E_{32}})]_{13,5} = \sqrt{\rho}G_{2,7}^{E_{32}} \tag{S313}$$

$$[R_{W,X,\theta}^{E_{32}}(G^{E_{32}})]_{13,10} = \sqrt{\rho}G_{7,7}^{E_{32}} \tag{S314}$$

$$[R_{W,X,\theta}^{E_{32}}(G^{E_{32}})]_{15,3} = -\frac{\sqrt{\rho}G_{1,16}^{E_{32}}}{\gamma\phi} \tag{S315}$$

$$[R_{W,X,\theta}^{E_{32}}(G^{E_{32}})]_{15,12} = -\frac{\sqrt{\rho}G_{9,16}^{E_{32}}}{\gamma\phi}, \tag{S316}$$

and the remaining entries of $R_{W,X,\theta}^{E_{32}}(G^{E_{32}})$ are zero. As above, plugging these expressions into eqn. (S293) and explicitly performing the block-matrix inverse yields the following set of coupled equations,

$$G_{7,2}^{E_{32}} = \gamma^2\sqrt{\rho}\bar{\tau}_1^2\psi G_{8,5}^{E_{32}} + \gamma^2\sqrt{\rho}\bar{\tau}_1^2\psi G_{13,5}^{E_{32}} + \frac{\gamma\bar{\tau}_1^2\psi G_{9,1}^{E_{32}}(\zeta - \eta)}{\phi} + \frac{\gamma^2\tau_1\bar{\tau}_1^2\psi(\zeta - \eta)(\zeta - \eta)}{\phi} \tag{S317}$$

$$G_{8,3}^{E_{32}} = \mathcal{I}_{\frac{1}{2},1}\zeta - \mathcal{I}_{\frac{1}{2},1}\eta - \frac{\gamma\bar{\tau}_1\mathcal{I}_{\frac{3}{2},1}\rho}{\psi} \tag{S318}$$

$$G_{8,4}^{E_{32}} = -\frac{\gamma\bar{\tau}_1\mathcal{I}_{1,1}\left(\rho\tau_1\psi(\zeta - \eta) + \phi\rho\right)}{\psi\phi} \tag{S319}$$

$$G_{8,5}^{E_{32}} = -\frac{\mathcal{I}_{1,1}\left(\rho\tau_1\psi(\zeta - \eta) + \phi\rho\right)}{\sqrt{\rho}\psi\phi} \tag{S320}$$

$$G_{8,6}^{E_{32}} = -\frac{\sqrt{\rho}\tau_1\left(\psi\mathcal{I}_{\frac{1}{2},1}\zeta - \psi\mathcal{I}_{\frac{1}{2},1}\eta - \gamma\bar{\tau}_1\mathcal{I}_{\frac{3}{2},1}\rho\right)}{\psi\phi} \tag{S321}$$

$$G_{9,1}^{E_{32}} = \gamma\tau_1^2 G_{7,2}^{E_{32}}(\zeta - \eta) - \gamma\sqrt{\rho}\tau_1^2 G_{11,3}^{E_{32}} + \gamma^2\tau_1^2\bar{\tau}_1(\zeta - \eta)(\zeta - \eta) \tag{S322}$$

$$G_{10,3}^{E_{32}} = \sqrt{\rho}\phi G_{7,2}^{E_{32}}\mathcal{I}_{\frac{1}{2},2} - \gamma\rho^{3/2}\bar{\tau}_1^2 G_{9,1}^{E_{32}}\mathcal{I}_{\frac{3}{2},2} - \frac{\gamma\sqrt{\rho}\bar{\tau}_1\phi\left(-\psi\mathcal{I}_{\frac{1}{2},2}\zeta + \psi\mathcal{I}_{\frac{1}{2},2}\eta + \gamma\bar{\tau}_1\mathcal{I}_{\frac{3}{2},2}\rho\right)}{\psi} \tag{S323}$$

$$G_{10,4}^{E_{32}} = \sqrt{\rho}\phi G_{7,2}^{E_{32}}\mathcal{I}_{0,2} - \gamma\rho^{3/2}\bar{\tau}_1^2 G_{9,1}^{E_{32}}\mathcal{I}_{1,2} - \frac{\gamma^2\sqrt{\rho}\bar{\tau}_1^2\mathcal{I}_{1,2}\left(\rho\tau_1\psi(\zeta - \eta) + \phi\rho\right)}{\psi} \tag{S324}$$

$$G_{10,5}^{E_{32}} = -\rho\tau_1 G_{7,2}^{E_{32}}\mathcal{I}_{1,2} - \rho\bar{\tau}_1 G_{9,1}^{E_{32}}\mathcal{I}_{1,2} - \frac{\mathcal{I}_{1,2}\left(\gamma\bar{\tau}_1\phi\rho + x\psi\zeta - x\psi\eta\right)}{\psi} \tag{S325}$$

$$G_{10,6}^{E_{32}} = -\rho\tau_1 G_{7,2}^{E_{32}}\mathcal{I}_{\frac{1}{2},2} - \rho\bar{\tau}_1 G_{9,1}^{E_{32}}\mathcal{I}_{\frac{1}{2},2} + \frac{\gamma^2\rho\tau_1\bar{\tau}_1^2\mathcal{I}_{\frac{3}{2},2}\rho}{\psi} + x\mathcal{I}_{\frac{1}{2},2}(\eta - \zeta) \tag{S326}$$

$$G_{11,3}^{E_{32}} = \sqrt{\rho}\phi G_{7,2}^{E_{32}}\mathcal{I}_{1,2} - \gamma\rho^{3/2}\bar{\tau}_1^2 G_{9,1}^{E_{32}}\mathcal{I}_{2,2} - \frac{\gamma\sqrt{\rho}\bar{\tau}_1\phi\left(-\psi\mathcal{I}_{1,2}\zeta + \psi\mathcal{I}_{1,2}\eta + \gamma\bar{\tau}_1\mathcal{I}_{2,2}\rho\right)}{\psi}$$

(S327)

$$G_{11,4}^{E_{32}} = \sqrt{\rho}\phi G_{7,2}^{E_{32}}\mathcal{I}_{\frac{1}{2},2} - \gamma\rho^{3/2}\bar{\tau}_1^2 G_{9,1}^{E_{32}}\mathcal{I}_{\frac{3}{2},2} - \frac{\gamma^2\sqrt{\rho}\bar{\tau}_1^2\mathcal{I}_{\frac{3}{2},2}\left(\rho\tau_1\psi\left(\zeta - \eta\right) + \phi\rho\right)}{\psi} \tag{S328}$$

$$G_{11,5}^{E_{32}} = -\rho\tau_1 G_{7,2}^{E_{32}}\mathcal{I}_{\frac{3}{2},2} - \rho\bar{\tau}_1 G_{9,1}^{E_{32}}\mathcal{I}_{\frac{3}{2},2} - \frac{\mathcal{I}_{\frac{3}{2},2}\left(\gamma\bar{\tau}_1\phi\rho + x\psi\zeta - x\psi\eta\right)}{\psi} \tag{S329}$$

$$G_{12,4}^{E_{32}} = -\rho\tau_1 G_{7,2}^{E_{32}}\mathcal{I}_{\frac{1}{2},2} - \rho\bar{\tau}_1 G_{9,1}^{E_{32}}\mathcal{I}_{\frac{1}{2},2} + \mathcal{I}_{\frac{3}{2},2}\left(\frac{\gamma^2\rho\tau_1\bar{\tau}_1^2\rho}{\psi} + \frac{x^2\left(\zeta - \eta\right)}{\phi}\right) \tag{S330}$$

$$G_{12,5}^{E_{32}} = -\frac{\sqrt{\rho}G_{9,1}^{E_{32}}\mathcal{I}_{\frac{1}{2},2}}{\gamma} + \frac{\rho^{3/2}\tau_1^2 G_{7,2}^{E_{32}}\mathcal{I}_{\frac{3}{2},2}}{\phi} + \frac{\mathcal{I}_{\frac{3}{2},2}\left(\gamma\rho^2\tau_1^2\bar{\tau}_1\psi\left(\zeta - \eta\right) + x\phi\rho\right)}{\sqrt{\rho}\psi\phi} \tag{S331}$$

$$G_{12,6}^{E_{32}} = -\frac{\sqrt{\rho}G_{9,1}^{E_{32}}\mathcal{I}_{0,2}}{\gamma} + \frac{\rho^{3/2}\tau_1^2 G_{7,2}^{E_{32}}\mathcal{I}_{1,2}}{\phi} + \frac{\gamma\rho^2\tau_1^2\bar{\tau}_1\mathcal{I}_{1,2}\left(\zeta - \eta\right) - \frac{x^2\mathcal{I}_{2,2}\rho}{\psi}}{\sqrt{\rho}\phi} \tag{S332}$$

$$G_{13,3}^{E_{32}} = -\rho\tau_1 G_{7,2}^{E_{32}}\mathcal{I}_{\frac{3}{2},2} - \rho\bar{\tau}_1 G_{9,1}^{E_{32}}\mathcal{I}_{\frac{3}{2},2} + \frac{\gamma^2\rho\tau_1\bar{\tau}_1^2\mathcal{I}_{\frac{5}{2},2}\rho}{\psi} + x\mathcal{I}_{\frac{3}{2},2}\left(\eta - \zeta\right) \tag{S333}$$

$$G_{13,4}^{E_{32}} = -\rho\tau_1 G_{7,2}^{E_{32}}\mathcal{I}_{1,2} - \rho\bar{\tau}_1 G_{9,1}^{E_{32}}\mathcal{I}_{1,2} + \mathcal{I}_{2,2}\left(\frac{\gamma^2\rho\tau_1\bar{\tau}_1^2\rho}{\psi} + \frac{x^2\left(\zeta - \eta\right)}{\phi}\right) \tag{S334}$$

$$G_{13,5}^{E_{32}} = -\frac{\sqrt{\rho}G_{9,1}^{E_{32}}\mathcal{I}_{1,2}}{\gamma} + \frac{\rho^{3/2}\tau_1^2 G_{7,2}^{E_{32}}\mathcal{I}_{2,2}}{\phi} + \frac{\mathcal{I}_{2,2}\left(\gamma\rho^2\tau_1^2\bar{\tau}_1\psi\left(\zeta - \eta\right) + x\phi\rho\right)}{\sqrt{\rho}\psi\phi} \tag{S335}$$

$$G_{13,6}^{E_{32}} = -\frac{\sqrt{\rho}G_{9,1}^{E_{32}}\mathcal{I}_{\frac{1}{2},2}}{\gamma} + \frac{\rho^{3/2}\tau_1^2 G_{7,2}^{E_{32}}\mathcal{I}_{\frac{3}{2},2}}{\phi} + \frac{\gamma\rho^2\tau_1^2\bar{\tau}_1\mathcal{I}_{\frac{3}{2},2}\left(\zeta - \eta\right) - \frac{x^2\mathcal{I}_{\frac{5}{2},2}\rho}{\psi}}{\sqrt{\rho}\phi} \tag{S336}$$

$$G_{13,8}^{E_{32}} = -\frac{x\mathcal{I}_{1,1}}{\phi} \tag{S337}$$

$$G_{14,3}^{E_{32}} = \sqrt{\rho}\tau_1\mathcal{I}_{\frac{3}{2},2}^{\beta}G_{7,2}^{E_{32}} + \sqrt{\rho}\bar{\tau}_1\mathcal{I}_{\frac{3}{2},2}^{\beta}G_{9,1}^{E_{32}} + \frac{x\psi\mathcal{I}_{\frac{3}{2},2}^{\beta}\left(\zeta - \eta\right) - \gamma^2\rho\tau_1\bar{\tau}_1^2\mathcal{I}_{\frac{5}{2},2}^{\beta}\rho}{\sqrt{\rho}\psi} \tag{S338}$$

$$G_{14,4}^{E_{32}} = \sqrt{\rho}\tau_1\mathcal{I}_{1,2}^{\beta}G_{7,2}^{E_{32}} + \sqrt{\rho}\bar{\tau}_1\mathcal{I}_{1,2}^{\beta}G_{9,1}^{E_{32}} - \frac{\mathcal{I}_{2,2}^{\beta}\left(\gamma^2\rho\tau_1\bar{\tau}_1^2\phi\rho + \psi x^2\zeta - \psi x^2\eta\right)}{\sqrt{\rho}\psi\phi} \tag{S339}$$

$$G_{14,5}^{E_{32}} = \frac{\mathcal{I}_{1,2}^{\beta}G_{9,1}^{E_{32}}}{\gamma} - \frac{\rho\tau_1^2\mathcal{I}_{2,2}^{\beta}G_{7,2}^{E_{32}}}{\phi} + \frac{\mathcal{I}_{2,2}^{\beta}\left(\frac{\gamma\rho^2\tau_1^2\bar{\tau}_1(\eta - \zeta)}{\phi} - \frac{x\rho}{\psi}\right)}{\rho} \tag{S340}$$

$$G_{14,6}^{E_{32}} = \frac{\mathcal{I}_{\frac{1}{2},2}^{\beta}G_{9,1}^{E_{32}}}{\gamma} - \frac{\rho\tau_1^2\mathcal{I}_{\frac{3}{2},2}^{\beta}G_{7,2}^{E_{32}}}{\phi} + \frac{\gamma\rho^2\tau_1^2\bar{\tau}_1\mathcal{I}_{\frac{3}{2},2}^{\beta}\left(\eta - \zeta\right) + \frac{x^2\mathcal{I}_{\frac{5}{2},2}^{\beta}\rho}{\psi}}{\rho\phi} \tag{S341}$$

$$G_{14,8}^{E_{32}} = \frac{x\mathcal{I}_{1,1}^{\beta}}{\sqrt{\rho}\phi} \tag{S342}$$

$$G_{14,11}^{E_{32}} = \frac{\tau_1\mathcal{I}_{\frac{1}{2},1}^{\beta}}{\phi} \tag{S343}$$

$$G_{14,13}^{E_{32}} = -\frac{\mathcal{I}_{0,1}^{\beta}}{\sqrt{\rho}} \tag{S344}$$

$$G_{15,3}^{E_{32}} = \sqrt{\rho}\tau_1\mathcal{I}_{2,2}^{\beta}G_{7,2}^{E_{32}} + \sqrt{\rho}\bar{\tau}_1\mathcal{I}_{2,2}^{\beta}G_{9,1}^{E_{32}} + \frac{x\psi\mathcal{I}_{2,2}^{\beta}\left(\zeta - \eta\right) - \gamma^2\rho\tau_1\bar{\tau}_1^2\mathcal{I}_{3,2}^{\beta}\rho}{\sqrt{\rho}\psi} \tag{S345}$$

$$G_{15,4}^{E_{32}} = \sqrt{\rho}\tau_1\mathcal{I}_{\frac{3}{2},2}^{\beta}G_{7,2}^{E_{32}} + \sqrt{\rho}\bar{\tau}_1\mathcal{I}_{\frac{3}{2},2}^{\beta}G_{9,1}^{E_{32}} - \frac{\mathcal{I}_{\frac{5}{2},2}^{\beta}\left(\gamma^2\rho\tau_1\bar{\tau}_1^2\phi\rho + \psi x^2\zeta - \psi x^2\eta\right)}{\sqrt{\rho}\psi\phi} \tag{S346}$$

$$G_{15,5}^{E_{32}} = \frac{\mathcal{I}_{\frac{3}{2},2}^{\beta} G_{9,1}^{E_{32}}}{\gamma} - \frac{\rho\tau_1^2 \mathcal{I}_{\frac{5}{2},2}^{\beta} G_{7,2}^{E_{32}}}{\phi} - \frac{\mathcal{I}_{\frac{5}{2},2}^{\beta}\left(\gamma\rho^2\tau_1^2\bar{\tau}_1\psi\left(\zeta-\eta\right)+x\phi\rho\right)}{\rho\psi\phi} \tag{S347}$$

$$G_{15,6}^{E_{32}} = \frac{\mathcal{I}_{1,2}^{\beta} G_{9,1}^{E_{32}}}{\gamma} - \frac{\rho\tau_1^2 \mathcal{I}_{2,2}^{\beta} G_{7,2}^{E_{32}}}{\phi} + \frac{\gamma\rho^2\tau_1^2\bar{\tau}_1\mathcal{I}_{2,2}^{\beta}\left(\eta-\zeta\right)+\frac{x^2\mathcal{I}_{3,2}^{\beta}\rho}{\psi}}{\rho\phi} \tag{S348}$$

$$G_{15,8}^{E_{32}} = \frac{x\mathcal{I}_{\frac{3}{2},1}^{\beta}}{\sqrt{\rho}\phi} \tag{S349}$$

$$G_{15,10}^{E_{32}} = \frac{\tau_1\mathcal{I}_{\frac{3}{2},1}^{\beta}}{\phi} \tag{S350}$$

$$G_{15,12}^{E_{32}} = -\frac{\mathcal{I}_{1,1}^{\beta}}{\sqrt{\rho}} \tag{S351}$$

$$G_{15,14}^{E_{32}} = \frac{\mathcal{I}_{\frac{1}{2},0}}{\phi} \tag{S352}$$

$$G_{16,1}^{E_{32}} = \tau_1^2\mathcal{I}_{1,1}^{\beta} G_{7,2}^{E_{32}}(\zeta-\eta) - \sqrt{\rho}\tau_1^2\mathcal{I}_{1,1}^{\beta} G_{11,3}^{E_{32}} + \gamma\tau_1^2\bar{\tau}_1\mathcal{I}_{1,1}^{\beta}(\zeta-\eta)(\zeta-\eta) - \sqrt{\rho}\tau_1 G_{15,3}^{E_{32}} \tag{S353}$$

$$G_{16,9}^{E_{32}} = \tau_1\mathcal{I}_{1,1}^{\beta} \tag{S354}$$

$$G_{1,1}^{E_{32}} = G_{9,9}^{E_{32}} = \gamma\tau_1 \tag{S355}$$

$$G_{2,2}^{E_{32}} = G_{7,7}^{E_{32}} = \gamma\bar{\tau}_1 \tag{S356}$$

$$G_{3,6}^{E_{32}} = G_{12,11}^{E_{32}} = -\frac{\sqrt{\rho}\tau_1\mathcal{I}_{0,1}}{\phi} \tag{S357}$$

$$G_{4,5}^{E_{32}} = G_{13,10}^{E_{32}} = -\frac{\sqrt{\rho}\tau_1\mathcal{I}_{1,1}}{\phi} \tag{S358}$$

$$G_{6,3}^{E_{32}} = G_{11,12}^{E_{32}} = \gamma\sqrt{\rho}\bar{\tau}_1\mathcal{I}_{1,1} \tag{S359}$$

$$G_{11,6}^{E_{32}} = G_{12,3}^{E_{32}} = -\rho\tau_1 G_{7,2}^{E_{32}}\mathcal{I}_{1,2} - \rho\bar{\tau}_1 G_{9,1}^{E_{32}}\mathcal{I}_{1,2} + \frac{\gamma^2\rho\tau_1\bar{\tau}_1^2\mathcal{I}_{2,2}\rho}{\psi} + x\mathcal{I}_{1,2}\left(\eta-\zeta\right) \tag{S360}$$

$$G_{14,10}^{E_{32}} = G_{15,11}^{E_{32}} = \frac{\tau_1\mathcal{I}_{1,1}^{\beta}}{\phi} \tag{S361}$$

$$G_{14,12}^{E_{32}} = G_{15,13}^{E_{32}} = -\frac{\mathcal{I}_{\frac{1}{2},1}^{\beta}}{\sqrt{\rho}} \tag{S362}$$

$$G_{5,4}^{E_{32}} = G_{10,8}^{E_{32}} = G_{10,13}^{E_{32}} = \gamma\sqrt{\rho}\bar{\tau}_1\mathcal{I}_{0,1} \tag{S363}$$

$$G_{3,5}^{E_{32}} = G_{4,6}^{E_{32}} = G_{12,10}^{E_{32}} = G_{13,11}^{E_{32}} = -\frac{\sqrt{\rho}\tau_1\mathcal{I}_{\frac{1}{2},1}}{\phi} \tag{S364}$$

$$G_{4,3}^{E_{32}} = G_{6,5}^{E_{32}} = G_{11,10}^{E_{32}} = G_{13,12}^{E_{32}} = \mathcal{I}_{\frac{1}{2},1} \tag{S365}$$

$$G_{8,8}^{E_{32}} = G_{14,14}^{E_{32}} = G_{15,15}^{E_{32}} = G_{16,16}^{E_{32}} = 1 \tag{S366}$$

$$G_{3,4}^{E_{32}} = G_{5,6}^{E_{32}} = G_{10,11}^{E_{32}} = G_{12,8}^{E_{32}} = G_{12,13}^{E_{32}} = -\frac{x\mathcal{I}_{\frac{1}{2},1}}{\phi} \tag{S367}$$

$$G_{5,3}^{E_{32}} = G_{6,4}^{E_{32}} = G_{10,12}^{E_{32}} = G_{11,8}^{E_{32}} = G_{11,13}^{E_{32}} = \gamma\sqrt{\rho}\bar{\tau}_1\mathcal{I}_{\frac{1}{2},1} \tag{S368}$$

$$G_{3,3}^{E_{32}} = G_{4,4}^{E_{32}} = G_{5,5}^{E_{32}} = G_{6,6}^{E_{32}} = G_{10,10}^{E_{32}} = G_{11,11}^{E_{32}} = G_{12,12}^{E_{32}} = G_{13,13}^{E_{32}} = \mathcal{I}_{0,1}\,, \tag{S369}$$

Here we have used the relations in eqns. (S266)-(S287), the definition of $\mathcal{I}_{a,b}^{\beta}$, as well as the results in Sec. G.4.1 to simplify the expressions. It is straightforward algebra to solve these equations for the undetermined entries of $G^{E_{32}}$ and thereby obtain the following expression for $E_{32}$,

$$E_{32} = \frac{(\eta-\zeta)A_{32} + \rho B_{32}}{D_{32}}\,, \tag{S370}$$

where,

$$
\begin{aligned}
A_{32} = &-\rho^3 \tau_1 \psi^2 x^4 \mathcal{I}_{1,1} \mathcal{I}_{2,2} \mathcal{I}_{2,2}^\beta + \rho^2 \tau_1 \psi x^3 \mathcal{I}_{2,2} \mathcal{I}_{2,2}^\beta (\rho\phi + x\psi(\zeta - \eta)) \\
&- \rho^3 \tau_1 \psi^2 x^3 \phi \mathcal{I}_{1,1} \mathcal{I}_{1,2} \mathcal{I}_{2,2}^\beta + \rho^2 \tau_1 \psi^2 x^2 \mathcal{I}_{1,1} \mathcal{I}_{1,1}^\beta (\eta - \zeta) \\
&+ \rho^2 \tau_1 \psi^2 x^2 \mathcal{I}_{1,1} \mathcal{I}_{2,2}^\beta (\rho + x(\zeta - \eta)) + \rho^2 \tau_1 \psi x^2 \phi \mathcal{I}_{1,2} \mathcal{I}_{2,2}^\beta (\rho\phi + x\psi(\zeta - \eta)) \\
&+ \rho^3 \tau_1 \psi^2 x^2 \phi \mathcal{I}_{1,1} \mathcal{I}_{1,2} \mathcal{I}_{1,1}^\beta - \rho^2 \tau_1 \psi x \phi \mathcal{I}_{1,2} \mathcal{I}_{1,1}^\beta (\rho\phi + x\psi(\zeta - \eta)) \\
&+ \rho \tau_1 \psi x \mathcal{I}_{1,1}^\beta (\zeta - \eta)(\rho\phi + x\psi(\zeta - \eta)) \\
&- \rho \tau_1 \psi x \mathcal{I}_{2,2}^\beta (\rho + x(\zeta - \eta))(\rho\phi + x\psi(\zeta - \eta)) \\
\end{aligned}
\tag{S371}
$$

$$
\begin{aligned}
B_{32} = &-\rho^2 \psi x^6 \mathcal{I}_{2,2}^2 \mathcal{I}_{3,2}^\beta - 2\rho^2 \psi x^5 \phi \mathcal{I}_{2,2}^2 \mathcal{I}_{2,2}^\beta + 2\rho \psi x^4 \phi \mathcal{I}_{1,2} \mathcal{I}_{3,2}^\beta (\eta - \zeta) \\
&- 2\rho^2 \psi x^4 \phi^2 \mathcal{I}_{1,2} \mathcal{I}_{2,2} \mathcal{I}_{2,2}^\beta + \rho^2 \psi x^4 \phi^2 \mathcal{I}_{1,2}^2 \mathcal{I}_{3,2}^\beta + \rho^2 \psi x^4 \phi \mathcal{I}_{2,2}^2 \mathcal{I}_{1,1}^\beta \\
&+ \rho^2 \psi x^4 \phi \mathcal{I}_{1,1} \mathcal{I}_{2,2} \mathcal{I}_{2,2}^\beta + \rho^2 x^4 \mathcal{I}_{2,2} \mathcal{I}_{3,2}^\beta (\psi + \phi) \\
&+ \rho x^3 \phi \mathcal{I}_{2,2} \mathcal{I}_{2,2}^\beta (\rho(\psi + \phi) + 2x\psi(\zeta - \eta)) \\
&+ \rho^2 \psi x^3 \phi^2 \mathcal{I}_{1,2} \mathcal{I}_{2,2} \mathcal{I}_{1,1}^\beta + \rho^2 \psi x^3 \phi^2 \mathcal{I}_{1,1} \mathcal{I}_{1,2} \mathcal{I}_{2,2}^\beta + \rho \psi x^2 \phi \mathcal{I}_{1,1} \mathcal{I}_{1,1}^\beta (\zeta - \eta) \\
&- \rho x^2 \phi \mathcal{I}_{2,2} \mathcal{I}_{1,1}^\beta (\rho\phi + x\psi(\zeta - \eta)) - \rho \psi x^2 \phi \mathcal{I}_{1,1} \mathcal{I}_{2,2}^\beta (\rho + x(\zeta - \eta)) \\
&- \rho^2 \psi x^2 \phi^2 \mathcal{I}_{1,1} \mathcal{I}_{1,2} \mathcal{I}_{1,1}^\beta + \mathcal{I}_{3,2}^\beta \left( x^4 \psi(\zeta - \eta)^2 - \rho^2 x^2 \phi \right) \\
\end{aligned}
\tag{S372}
$$

$$
\begin{aligned}
D_{32} = &-\rho^3 \psi x^4 \phi \mathcal{I}_{2,2}^2 + 2\rho^2 \psi x^2 \phi^2 \mathcal{I}_{1,2} (\eta - \zeta) \\
&+ \rho^3 \psi x^2 \phi^3 \mathcal{I}_{1,2}^2 + \rho^3 x^2 \phi \mathcal{I}_{2,2} (\psi + \phi) + \rho\phi \left( x^2 \psi(\zeta - \eta)^2 - \rho^2 \phi \right) .
\end{aligned}
\tag{S373}
$$

Further simplifications are possible using the raising and lowering identities in eqn. (S9), as well as the results in Sec. G.4.1, to obtain,

$$
E_{32} = \frac{x^2}{\phi} \mathcal{I}_{3,2}^\beta - \rho \frac{\psi}{\phi} \frac{\partial x}{\partial \gamma} \left( \mathcal{I}_{1,1}^\beta (\omega + \phi \mathcal{I}_{1,2})(\omega + \mathcal{I}_{1,1}) + \frac{\phi^2}{\psi} \gamma \bar{\tau}_1 \mathcal{I}_{1,2}^\beta \mathcal{I}_{2,2} + \gamma \tau_1 \mathcal{I}_{2,2}^\beta (\omega + \phi \mathcal{I}_{1,2}) \right),
\tag{S374}
$$

where

$$
\frac{\partial x}{\partial \gamma} = -\frac{x}{\gamma + \rho \gamma (\tau_1 \psi / \phi + \bar{\tau}_1)(\omega + \phi \mathcal{I}_{1,2})} .
\tag{S375}
$$

### G.4.5   $E_4$

The calculation of $E_4$ proceeds exactly as in (Tripuraneni et al., 2021a;b) with the simple modification of including an additional factor $\Sigma_\beta$ inside the final trace term, yielding

$$
E_4 = \frac{x^2}{\phi} \mathcal{I}_{3,2}^\beta .
\tag{S376}
$$

### G.5   FINAL RESULT FOR BIAS, VARIANCE, AND TEST ERROR

Putting the above pieces together, we have,

$$
B_\mu = \phi \mathcal{I}_{1,2}^\beta
\tag{S377}
$$

$$
V_\mu = -\rho \frac{\psi}{\phi} \frac{\partial x}{\partial \gamma} \left( \mathcal{I}_{1,1}^\beta (\omega + \phi \mathcal{I}_{1,2})(\omega + \mathcal{I}_{1,1}) + \frac{\phi^2}{\psi} \gamma \bar{\tau}_1 \mathcal{I}_{1,2}^\beta \mathcal{I}_{2,2} + \gamma \tau_1 \mathcal{I}_{2,2}^\beta (\omega + \phi \mathcal{I}_{1,2}) \right.
$$

$$
\left. + \sigma_\varepsilon^2 \left( (\omega + \phi \mathcal{I}_{1,2})(\omega + \mathcal{I}_{1,1}) + \frac{\phi}{\psi} \gamma \bar{\tau}_1 \mathcal{I}_{2,2} \right) \right).
\tag{S378}
$$

$$
\tag{S379}
$$

Some algebra shows that

$$
E_\mu = B_\mu + V_\mu
\tag{S380}
$$

$$= -\frac{\partial_\gamma(\tau_1(\sigma_\varepsilon^2 + \mathcal{I}_{1,1}^\beta))}{\tau_1^2} - \sigma_\varepsilon^2 \tag{S381}$$

$$= \frac{E_{\text{train}}}{\gamma^2 \tau_1^2} - \sigma_\varepsilon^2 . \tag{S382}$$

**Corollary G.1.** *In the setting of Theorem 3.1, as the ridge regularization constant $\gamma \to 0$, $E_\mu = B_\mu + V_\mu$ with $B_\mu = \phi \mathcal{I}_{1,2}^\beta$ and $V_\mu$ given by*

$$V_\mu \xrightarrow{\gamma \to 0} \frac{\min(\phi, \psi)}{|\phi - \psi|}(\sigma_\varepsilon^2 + \mathcal{I}_{1,1}^\beta) + \begin{cases} x \mathcal{I}_{2,2}^\beta & \text{if } \phi < \psi \\ \frac{x \mathcal{I}_{2,2}}{\omega + \phi \mathcal{I}_{1,2}}(\sigma_\varepsilon^2 + \mathcal{I}_{1,2}^\beta) & \text{otherwise} \end{cases}, \tag{S383}$$

*where $x$ is the unique positive real root of $x = \frac{\min(1, \phi/\psi)}{\omega + \mathcal{I}_{1,1}}$.*

