# OpenReview forum: "Anisotropic Random Feature Regression in High Dimensions"
_ICLR.cc/2022/Conference — ICLR 2022 Poster_

### Official Review · Reviewer_3RfG · 2021-10-19

**Correctness:** 3
**Technical Novelty And Significance:** 2
**Empirical Novelty And Significance:** 2
**Recommendation:** 6
**Confidence:** 4

**Main Review:**

My main concern with this work is the total disconnection with the relevant literature in the topic. Different flavours of anisotropic random features have been previously studied in the literature: on the first layer weights [1], on the input features $x_{i}$ [2, 3], on both the target function weights $\beta$ and on the input features $x_{i}$ [4, 5], with **NONE** of these works cited in this manuscript.

Moreover, in the isotropic case it has been proven that the asymptotic generalisation and training errors of RF in the proportional regime studied in this manuscript are equivalent to the error of an equivalent Gaussian model with correlated inputs [3, 6, 7, 8, 9]. This Gaussian equivalence was conjectured to hold also in the the anisotropic case of correlated inputs and target function weights, with extensive numerical evidence supporting it [3, 5]. Under this conjecture, the asymptotic formulas in this manuscript follow from previous works studying anisotropic ridge regression [10], and were generalised to generic convex losses and penalties in [5]. Therefore, it is no surprise that the effects of anisotropy in this work are similar to the ones reported in [11].

I don't think this completely voids the interest of the discussion in this manuscript: even though the formulas might not be new, it unveils some interesting phenomenology. However, I do believe it would be much more interesting if it was put in perspective with the aforementioned related literature. For instance, how does the phenomenology induced by anisotropy compare with the results reported in [1,2,4]? What insights the RMT proof provide on the extent the Gaussian equivalence holds? etc.

[1] Behrooz Ghorbani, Song Mei, Theodor Misiakiewicz, Andrea Montanari, *Limitations of Lazy Training of Two-layers Neural Networks*, arXiv: arXiv:1906.08899 [stat.ML].

[2] Behrooz Ghorbani, Song Mei, Theodor Misiakiewicz, Andrea Montanari, *When do neural networks outperform kernel methods?*, arXiv:2006.13409 [stat.ML].

[3] Sebastian Goldt, Bruno Loureiro, Galen Reeves, Florent Krzakala, Marc Mézard, Lenka Zdeborová, *The Gaussian equivalence of generative models for learning with shallow neural networks*, arXiv:2006.14709 [stat.ML].

[4] Stéphane d'Ascoli, Marylou Gabrié, Levent Sagun, Giulio Biroli, *On the interplay between data structure and loss function in classification problems*, arXiv:2103.05524 [cs.LG]

[5] Bruno Loureiro, Cédric Gerbelot, Hugo Cui, Sebastian Goldt, Florent Krzakala, Marc Mézard, Lenka Zdeborová, *Learning curves of generic features maps for realistic datasets with a teacher-student model*, arXiv:2102.08127 [stat.ML]

[6] Song Mei, Andrea Montanari, *The generalization error of random features regression: Precise asymptotics and double descent curve*, arXiv:1908.05355 [math.ST]

[7] Federica Gerace, Bruno Loureiro, Florent Krzakala, Marc Mézard, Lenka Zdeborová, *Generalisation error in learning with random features and the hidden manifold model*, arXiv:2002.09339 [math.ST].

[8] Hong Hu, Yue M. Lu, *Universality Laws for High-Dimensional Learning with Random Features*, arXiv:2009.07669 [cs.IT].

[9] Pennington and P. Worah, *Nonlinear random matrix theory for deep learning*, in Advances in Neural Information Processing Systems, 2017, pp. 2637-2646.

[10] Denny Wu and Ji Xu. *On the optimal weighted $\ell_2$ regularization in overparameterized linear regression*. arXiv:2006.05800 [stat.ML]

[11] Gabriel Mel, Surya Ganguli, *A theory of high dimensional regression with arbitrary correlations between input features and target functions: sample complexity, multiple descent curves and a hierarchy of phase transitions*, Proceedings of the 38th International Conference on Machine Learning, PMLR 139:7578-7587, 2021.

**Summary Of The Paper:**

This work studies random features regression on a setting where data $(x_{i}, y_{i})\in\mathbb{R}^{n_{0}+1}$, $i=1, \cdots,n$ is generated from a noisy linear model $y_{i}=\beta^{\top}x_{i}+\epsilon_{i}$ with correlated covariates $x_{i}\sim\mathcal{N}(0, \Sigma)$, planted weights $\beta\sim\mathcal{N}(0,\Sigma_{\beta})$ and independent Gaussian noise $\epsilon_{i}\sim\mathcal{N}(0,\sigma_{\epsilon}^2)$. Estimation is given by $\hat{y}(x) = \hat{\beta}^{\top}\sigma\left(Wx\right)$ with $\sigma$ a real-valued activation function and $W\in\mathbb{R}^{n_{1}\times n_{0}}$ a Gaussian matrix with iid $\mathcal{N}(0, 1/n_{0})$ entries, and $\hat{\beta}\in\mathbb{R}^{n_{1}}$ minimising the ridge regression risk on the features $f = \sigma\left(Wx\right)\in\mathbb{R}^{n_{1}}$.

Its main theoretical contribution is to provide an asymptotic characterisation of the test MSE, the bias and the variance on the proportional regime $m,n_{0}, n_{1}\to\infty$ with fixed ratios $\phi=n_{1}/m, \psi=n_{0}/n_{1}$, as a function of the asymptotic spectral statistics of $(\Sigma, \Sigma_{\beta})$, the ratios $(\phi,\psi)$, the $\ell_2$ penalty strength $\gamma>0$, the noise variance $\sigma_{\epsilon}^2$ and the constants $\rho = \mathbb{E}\left[z\sigma(z)\right]$, $\eta = \rm{Var}\left[\sigma(z)\right]$, $z\sim\mathcal{N}(0,1)$, characterising the activation $\sigma$.

It provides numerical experiments confirming the validity of their formulas for finite but large dimension, and discusses the implications of the anisotropies $(\Sigma, \Sigma_{\beta})$  on the bias, variance and test error in different regimes. In particular, it shows that alignment between $\beta$ and the leading eigenvectors of $\Sigma$ reduces the test MSE, and that anisotropy can induce double-descent like behaviour in the learning curves.


**Summary Of The Review:**

This work provides some interesting insights on the effects of anisotropy in the RF ridge regression setting, but would strongly benefit from a major rewriting to account for a discussion of the relevant literature in this topic.

---------------------------------------------
**Update after the discussion period**

My main concern with this work was the disconnection with the existing literature dealing with anisotropic features on the asymptotic proportional regime. At first, I was surprised the authors had missed a long line of works based on a different approach than RMT to treat the same regime (Gordon inequalities), and the corresponding Gaussian equivalence discussion in a subset of these works. However, given their engament during the discussion period and their willingness to acknowledge this gap, I believe they have acted in good faith.

The revised manuscript from 23.11.2021 (including the revised appendix A) goes in the right direction. However, I join reviewer *ot31* in the opinion that it could be further improved by:

- Including the explicit connection with [5] in the case of the square loss with $\ell_2$ penalty (which the authors said they are already working on). This would be an interesting result in itself, bridging a gap between the two approaches in the existing literature and opening a possible direction of future research consisting of exploring whether the alignment discussion in this work hold for other losses.
- Being more explicit about the linearisation assumption (c.f. *ot31* updated review).

Assuming these suggestions will be incorporated in the final manuscript, I am changing my recommendation to an accept.

---

> ### Author Response · Authors · 2021-11-23
> **Response 1 to Reviewer 3RfG [1/3]**
>
> *Different flavours of anisotropic random features have been previously studied in the literature: on the first layer weights [1], on the input features $x_i$ [2, 3], on both the target function weights $\beta$ and on the input features $x_i$ [4, 5], with NONE of these works cited in this manuscript.*
>
> We thank the reviewer for pointing us to these interesting related works, many of which were unknown to us. We have updated the manuscript with a detailed discussion of these and other works and our paper’s relation to them. In particular, we have inserted short references throughout the text where appropriate, revised Section 1.2, and added a longer more detailed discussion in Sec. A.
>
> Among these related works, [4] is by far the closest to the current manuscript, and the only one that overlaps with the specific focus of this work, namely the phenomenology of weight-data alignment and anisotropy in the random feature model. However, we would like to highlight that the current version (arXiv v2) of [4] appeared on Oct 12, after the submission deadline for ICLR.  Additionally, it seems that [4] will appear at NeurIPS, but of course the proceedings were also unavailable when this manuscript was submitted. The version of this paper that was available prior to the deadline (arXiv v1) has a substantially different focus and narrative, with in fact no mention of “alignment” whatsoever (the discussion is in terms of “relevance” and “salience”). We discuss the connections to both versions of this paper in detail below, but we would kindly ask the reviewer to bear these facts in mind when reassessing the relative contributions and novelty of our work.
>
> Regarding the other references, we were actually aware of [1] and [2], but we felt their connection was too distant to merit in-depth discussion in our short manuscript. However, upon rereading these works, we did find a few connections worth highlighting, which are now incorporated into Sec. A.
>
> We are grateful to learn about [3] and [5], which we were previously unaware of. We are sure the reviewer appreciates the difficulty in tracking every paper that lands on the arxiv, and these two important references missed our radar. We have added a discussion of these papers to Sec 1.2 and Sec A. (Though we would like to emphasize that neither the linearization nor its proof are a focus of the current manuscript).

---

> ### Author Response · Authors · 2021-11-23
> **Response to Reviewer 3RfG [2/3]**
>
> *Moreover, in the isotropic case it has been proven that the asymptotic generalisation and training errors of RF in the proportional regime studied in this manuscript are equivalent to the error of an equivalent Gaussian model with correlated inputs [3, 6, 7, 8, 9]. This Gaussian equivalence was conjectured to hold also in the the anisotropic case of correlated inputs and target function weights, with extensive numerical evidence supporting it [3, 5].*
>
> While we were previously unaware of this conjecture, we are pleased to know that it is consistent with our results and with the results of (Anonymous, 2021), which established the invariance of the test error under a replacement of the random feature matrix with a linear-signal-plus-noise Gaussian equivalent, in the setting of anisotropic covariates.
> &nbsp;
> &nbsp;
> *Under this conjecture, the asymptotic formulas in this manuscript follow from previous works studying anisotropic ridge regression [10], and were generalized to generic convex losses and penalties in [5].*
>
> As far as we can tell, it is not possible to obtain our formulas from the results for ridge regression presented in [10]. On the contrary, in Sec D.2 (E.2 in the revised manuscript), we show that a simple (and seemingly non-invertible) degeneration of our formulas reproduces the results of that paper.
>
> On the other hand, after some initial inspection, it does seem that our expression for the total error may be equivalent to that of [4], at least when the LJSD $\mu$ is a finite sum of delta masses. To this point we make several remarks:
>
> 1. Our development of these results was completely independent and we were previously unaware of this work on arXiv. Moreover, our characterization of the LJSD $\mu$ is slightly more general than the case studied in [4] (though our results are less general in another sense, as they apply only to squared loss).
> &nbsp;
> 2. Given the highly detailed calculations involved in these efforts, and the fact that the techniques are completely different, it is valuable to publish both derivations.
> &nbsp;
> 3. The ultimate expression of our result has a much simpler form, involving only a single scalar self-consistent equation, which lends itself to more straightforward downstream calculations.
> &nbsp;
> 4. Our calculation does not rely on the conjectured extension of the Gaussian equivalence theorem or the replica method; as such, it can be made completely rigorous (though of course, we are relying on a proof of the linearization for anisotropic covariates from (Anonymous, 2021), as highlighted by reviewer ot31).
> &nbsp;
> 5. We also derive the bias and variance, which aids significantly in the interpretation of the phenomenology, and is a novel result.
> &nbsp;
> 6. Theorem 3.1 is not the only contribution of this paper, nor is it the main one. It only constitutes one of the five main contributions that we list in Sec 1.1. We strongly believe that these other contributions are valuable to the community, irrespective of the provenance or novelty of the main theorem, or the techniques used to derive it.
> &nbsp;
> &nbsp;
> *Therefore, it is no surprise that the effects of anisotropy in this work are similar to the ones reported in [11].*
>
> Only some of the effects are similar to the linear case studied in [11] -- we also uncover novel phenomena that are unique to the random feature model, and novel phenomena that were not identified in [4]. See the response below discussing our detailed phenomenological observations and contributions in the context of previous work.

---

> ### Author Response · Authors · 2021-11-23
> **Response to Reviewer 3RfG [3/3]**
>
> *I don't think this completely voids the interest of the discussion in this manuscript: even though the formulas might not be new, it unveils some interesting phenomenology. However, I do believe it would be much more interesting if it was put in perspective with the aforementioned related literature.*
>
> We agree that it is important to cast our results in the broader context as much as possible. Having said that, many of the most salient comparison points come from [4v2], which were absent from the version [4v1] available at the time of submission. Nevertheless, we have added some discussion throughout the manuscript that positions our results in the context of this and additional related literature. Owing to space constraints, it was not possible to include all relevant discussion in the main text, so we also added an extended discussion in Sec. A. For a summary, see the comments below.
> &nbsp;
> &nbsp;
> *For instance, how does the phenomenology induced by anisotropy compare with the results reported in [1,2,4]?*
>
> At a high level, we demonstrate several novel phenomena related to input data anisotropy, some with analogs in the linear setting (eg. sample-wise multiple descent), and others which are totally novel (eg. learning curve cliffs). We also introduce a novel definition of weight-data alignment and a corresponding partial order and show how it affects the total error, bias, and variance. Finally, unlike many previous results, we include proofs for many of the phenomena we observe, which are made possible by the relatively simple expressions given in Theorem 3.1.
>
> At a more detailed level:
>
> 1. We define a partial order on the space of weight-data alignments and prove that the total error and the bias decrease in response to stronger alignment (Proposition 3.3). While [4] introduces a notion of alignment, it does not define the concept, and the conclusions about it derive from numerical evaluation of the formulas rather than from rigorous proofs.
> &nbsp;
> As a result, some of the underlying phenomena are obfuscated. To give one example, it is not clear why the “isotropic” and “misaligned” curves cross each other in Fig 2c of that paper -- shouldn’t the misaligned model always perform worse? Our results provide the explanation: owing to the differing covariate distributions, the two forms of alignment are incomparable under the partial order.
> &nbsp;
> Our results also connect with several recent observations in related but distinct learning contexts: [1] studies the random feature model with isotropic inputs and anisotropic weights in the case of a fixed quadratic target function and derives an asymptotic formula for the test error in the population limit (ie. $m\gg n_0, n_1$). For wide networks, $n_1\gg n_0$, the error simplifies and is exactly proportional to a simple measure of alignment between the random feature weights and the target that is loosely related to the measure we propose in Definition 2.1. [2] studies a related random feature model in the population limit, and shows that increasing the power of the input data in the target subspace generally decreases test error and the number of random features required to learn a function of fixed complexity, paralleling our general results on alignment (see also Fig. 3b for illustration in the context of the d-scale model).
> &nbsp;
> 2. We observe steep cliffs in the parameter-wise multiple descent curves when input data-scales are highly separated. We show that cliffs manifest very generally when the spectrum of the input data has a large gap, and we observe that these cliffs persist under optimal regularization. This is a novel phenomenon that is not observed in the setting of linear regression, or indeed in any previous work on random feature regression.
> &nbsp;
> 3. We prove that parameter-wise multiple descent is limited to double descent in this model, which is a novel result even in the isotropic setting. We also demonstrate sample-wise multiple descent, generalizing the observation of triple descent in [4v1], and provide an explanation for its origin in terms of large gaps in the input data spectrum (note that similar behavior occurs in the linear case [10,11]).
> &nbsp;
> 4. Unlike previous work on the random feature model, we provide detailed observations connecting learning curve phenomenology with the spectrum of the kernel matrix.

---

> > ### Comment · Reviewer_3RfG · 2021-11-23
> > **Response to authors**
> >
> > Thank you for your detailed reply and for the revised manuscript. I appreaciate that the authors acknowledge the missing literature and that they have updated the discussion.
> >
> > While I understand that the focus of the manuscript is rather on the phenomenology, and not in the formula itself, Theorem 3.1 is the starting point for the discussion, and a key ingredient supporting it is the linearization / Gaussian equivalence. Therefore, I do believe that the added discussion stating this clearly and situating the work within this literature is important.
> >
> > A small comment: unless I am missing something, I believe that up to the linearization the formula of [4] for convex losses derived using the non-rigorous replica method is a specific case of the one appearing in [5], which is rigorous (uses Gordon min-max inequalities).
> >
> > I will read the revised manuscript during the remaining discussion period, and update my review accordingly.

---

> > > ### Author Response · Authors · 2021-11-23
> > > **Response to Reviewer 3RfG**
> > >
> > > Thank you for the clarification! This line of work is new to us and the chronology seems to be somewhat complicated, so we hope the reviewer can forgive small inaccuracies as we continue to read these interesting papers. As the expressions and proofs in [4,5] are quite involved, it is not trivial to assess their equivalence to our formulas, but we are still working through this (particularly the case of [5]) and will update the final version accordingly.
> > >
> > > If the reviewer believes any additional clarifications of the related work are warranted, we are happy to accommodate them in a final version.

---

### Official Review · Reviewer_bozT · 2021-10-29

**Correctness:** 4
**Technical Novelty And Significance:** 4
**Empirical Novelty And Significance:** Not applicable
**Recommendation:** 8
**Confidence:** 4

**Main Review:**

The major contribution of this work is to extend the high-dimensional characterization of random feature (kernel) regression model to the case of anisotropic data. This improves previous efforts of [Mel & Ganguli, 2021] by considering nonlinear random features models, and of [Mei & Montanari, 2021] by considering anisotropic data.

I think this paper makes a solid contribution to the understanding of high-dimensional random feature models.  Many insights (multiple descents due to data and model structure, steep cliffs, etc.) offered by the proposed analysis are valuable in the future (and improved) design of large-scale machine learning models more generally.



**Summary Of The Paper:**

This paper evaluates the training and generalization errors of the random feature (ridge) regression model, in the regime where the number of samples $m$, the input feature dimension $n_0$, and the hidden layer size $n_1$ tend to infinity at the same rate. More precisely, the authors consider
* Gaussian data with zero mean and general covariance $x_i \sim \mathcal{N}(0, \Sigma)$;
* label generated by $y(x_i) = \beta^\top x_i/\sqrt{n_0} + \epsilon_i$, for some additive noise $\epsilon_i \sim \mathcal{N}(0, \sigma_\epsilon^2 ) $ and coefficient vector $\beta \sim \mathcal{N} (0, \Sigma_\beta)$;
* (random feature) kernel regression on $K(x_i, x_j) = \frac1{n_1} \sigma(W x_i/\sqrt{n_0})^\top \sigma(W x_j/\sqrt{n_0})$;

and evaluate the training and test performance as a function of the dimensionality, the nonlinear function (via a few scalar parameters), as well as the spectral behavior of $\Sigma$ and $\Sigma_\beta$.

As a consequence of this theoretical evaluation, the authors characterize sample-wise multiple descents, steep cliffs, and how they relate to, e.g., the spectrum of the (random feature) kernel matrices and the structure of coefficient vector $\beta$.

**Summary Of The Review:**

Some detailed typos and remarks:
* page 3 on the model: I do not understand why one needs to consider the setting of random coefficient vector $\beta \sim \mathcal{N}(0, \Sigma_\beta)$ instead of, e.g., considering deterministic $\beta$, as in previous efforts. I find it somewhat confusing since the authors say after Equation (6) that "the outer expectation over $\beta$ has been suppressed since ..." Could the authors comment on this?
* page 3 "namely $\beta \sim \mathcal{N} (0, \Sigma_\beta)$.."
* page 5: after (11) there should be more discussions and intuitions to help the readers understand the definition of $d$-scale LJSDs introduced here, which turns out to be of crucial significance in this paper.
* page 5 Equation (14): perhaps add a few words to define the (partial) derivative.
* page 7 Figure 2 covariance model with $\alpha = 10^3$: a typo here? Also, we see the sample-wise multiple descents appear for relatively small values of $\omega$, which, I imagine, implies a close-to-linear regime, could the authors comment on this?

---

> ### Author Response · Authors · 2021-11-17
> **Response 1 to Reviewer bozT [1/1]**
>
> **Some detailed typos and remarks:**
> 1. *page 3 on the model: I do not understand why one needs to consider the setting of random coefficient vector $\beta∼\mathcal{N}(0,\Sigma_\beta)$ instead of, e.g., considering deterministic $\beta$, as in previous efforts. I find it somewhat confusing since the authors say after Equation (6) that "the outer expectation over $\beta$ has been suppressed since ..." Could the authors comment on this? page 3 "namely $\beta∼\mathcal{N}(0,\Sigma_\beta)$.."*
> &nbsp;
> See response above to Reviewer 6qzE’s comment 1.
> &nbsp;
> 2. *page 5: after (11) there should be more discussions and intuitions to help the readers understand the definition of d-scale LJSDs introduced here, which turns out to be of crucial significance in this paper.*
> &nbsp;
> We thank Reviewer bozT for bringing this to our attention. We have added a key sentence aiding the interpretation of the d-scale model that was missing from the original manuscript.
> &nbsp;
> 3. *page 5 Equation (14): perhaps add a few words to define the (partial) derivative.*
> &nbsp;
> Done.
> &nbsp;
> 4. *page 7 Figure 2 covariance model with $\alpha=10^3$: a typo here? Also, we see the sample-wise multiple descents appear for relatively small values of $\omega$, which, I imagine, implies a close-to-linear regime, could the authors comment on this?*
> &nbsp;
> $\alpha=10^3$ is correct. We chose this large value for $\alpha$ (corresponding to data scale standard deviations separated by factors of $\approx$ 30) to cleanly illustrate the effects of strong anisotropy in Figure 2. As $\alpha$ is reduced and the scales are brought closer together, the peaks gradually merge. Regarding $\omega$ see our response to Reviewer 6qzE’s comment 4.

---

### Official Review · Reviewer_ot31 · 2021-11-03

**Correctness:** 3
**Technical Novelty And Significance:** 3
**Empirical Novelty And Significance:** 2
**Recommendation:** 6
**Confidence:** 4

**Details Of Ethics Concerns:**

No concerns.

**Main Review:**

**Strengths**:

- To my knowledge this is a novel contribution that broadens the understanding of overparameterization, a topic of interest to the ICLR readership

- The asymptotic formulas are exact and relatively simple.

- This paper introduces an interesting notion of alignment between the covariance matrix of the covariates and that of the parameter, and it is shown how this alignment affects the test error

- The main paper is well-organized, clear, and has interesting, illustrative figures.

**Concerns**:

- I am mainly concerned with the rigor and clarity of the derivation of the main theorem, Theorem 3.1. Please see the section "Proof of Theorem 3.1" below.

- To my understanding, Figures 1--3 are generated by plotting the asymptotic formulae from Theorem 3.1. It would strengthen the paper to confirm these plots with simulations on synthetic data.


**Proof of Theorem 3.1**

My points below mostly concern Section F to the end of Section F.4.1 in the Supplement. The proof happens in two steps: (i) linearization of random features and (ii) application of free-probability theory to compute the trace of some complicated matrices.

(i) *Linearization*: This paper claims without proof that the test error is unchanged by an appropriate linearization of the non-linear random features. Of the given references on pg.13, I could only find a proof of the invariance of test error under linearization in Mei--Montanari '19, which only studies the case of random spherical (that is, isotropic) covariates. This requires justification in the anisotropic case.

Also a minor point: after linearization, the definition of $F$ changes I think. Please make this clear in the text

(ii) *Free probability*:  Most of Section F relies on computations of various traces of matrix products involving covariance matrices and iid Gaussian matrices.

*Major issues with (ii)*

-  pg 13: There is no explanation of the construction of the $Q^{K^{-1}}$, which I believe is the *linear pencil*. A short, precise description of the linear pencil methodology would improve the paper since it is crucial to the proof.

-  pg 17: "We aim to compute the asymptotic values..." I am concerned about the looseness in taking limits described here. How can the limit in $n_0, n_1, m$ be taken in two stages? Once these go to infinity, there should be no dependence of any expressions on $n_0, n_1, m$. It is also not clear at what point limits are taken in the computation in Section F.4.1.

-  pg 18: Formula (S200) is crucial. Please give a reference to the precise section and theorem in Mingo--Speicher '17.

-  pg 19: The R-transforms in (S202)--(S208) depend on indices of $G^{K^{-1}}$ that go out of range. For example there is a $(3, 15)$ entry in (S203).

-  pg 19: The details of R-transform computations are omitted. I find this acceptable, but please cite the specific chapter in Mingo--Speicher that describes the technique for computing them

-  pg 19: How is (S200) used to derive equations (S209)--(S228)? I would appreciate if in the discussion period the authors can explain how to derive (S209) from (S200) and the various R-transforms. In particular, I would like to understand the details of why the expectation over $\mu$ shows up and what happened to all of the random matrices.

-  pg 19: The appearance of $\mu$ in (S209)--(S228) suggests that limits with respect to $n_0, n_1, m$ have been taken. So how is it that the terms $n_1, n_0, m$ still appear in these expressions?

-  pg 22: The calculation in Section F.4.4 involves the system (S312)--(S364) consisting of ~50 equations that spans 3 pages. It is hard to verify the end result. Are all of these equations necessary for the derivation? If a simpler explanation isn't possible, I recommend giving a computer-assisted proof citing Mathematica or other algebra software.

*Minor issues with (ii)*

-  pg 13: $K$ appears to be regularized in (S137), but later $\hat K$ is introduced that has the same formula. Should $K$ be non-regularized in Section F?

-  pg 13: $\theta$ is not defined and seems to sometimes be written as $\Theta$ or $\Theta_F$.

-  pg 18: "Here the normalized trace... acts on the space of 18x18 matrices." Should it be 9x9 matrices?

-  For readability, I suggest re-organizing lists of equations with indices listed in lexicographic order ( for example in (S172)--(S194) )

-  pg 19: The terms in (S209, S210) have no expectation over $\mu$, but the remaining ones do. Is this a typo?

**Minor issues/typos in the main text**

- For consistency, use either "total error" throughout or "test error" throughout

- pg 2: "... in the relatively trivial case of isotropic covariates..." Please replace this with "... in the case of isotropic covariates..." It is clear that isotropy is a simpler case, but I think it's good to avoid referring to prior works as relatively trivial.

- pg 3, Eq (5, 6): The expectation over $\beta$ appears and then disappears; for consistency please make a choice whether or not to include the expectation over $\beta$ in the error metric.

- pg 3: "...the test error of linear regression depends on the geometry of $(\Sigma, \beta)$." This is confusing since $\beta$ is random. Perhaps it should be the geometry of $(\Sigma, \Sigma_\beta)$

- pg 4, Eq (9): Usually the term "empirical spectral distribution" is reserved for the random measure given by the eigenvalues of a *random* matrix. Since $\Sigma, \Sigma_\beta, \mu_{n_0}$ are deterministic, perhaps just saying "joint spectral distribution" suffices?

- pg 5, Definition 2.1: "If the asymptotic coefficients are such that $\mathbb{E}^{\mu_1}[\lambda q| \lambda]/\mathbb{E}^{\mu_2}[\lambda q| \lambda]$..." I think it should be $\mathbb{E}^{\mu_1}[ q| \lambda]/\mathbb{E}^{\mu_2}[ q| \lambda]$ instead (otherwise the value is independent of $\lambda$)

- pg 5, Theorem 3.1: The main result is hard to parse because the definitions of parameters are scattered throughout pgs 1--5. A short recap or pointer to prior definitions could help with this.

- pg 5, Theorem 3.1: It took me a while to understand the meaning of  "the training error $E_{train}^{\Sigma} \to E_{train}^{\mu}$ where *insert Eq (14)*".  A clearer rephrasing is "the training error $E_{train}^{\Sigma}$ tends to the value $E_{train}^{\mu}$, where *insert Eq (14)*".

- pg 6, Figure 1: (f) is missing a legend

- pg 6, Figure 1: In (d, e), it took me a while to understand what the black solid and dashed lines correspond to in the plots. I recommend just making a slightly bigger legend of the form " Total error ($\theta$ = 1), Bias ($\theta$ = 1), Variance ($\theta$ = 1), Total error ($\theta$ = -1), Bias ($\theta$ = -1), Variance ($\theta$ = -1)"

- pg 6, Figure 1: What do the blue dashed/solid lines indicate in (a)? What do the gold dashed/solid lines indicate in (b, c)? Are these necessary?

- pg 7, Figure 2: There is not much color variation in (b). Is this intentional?

- pg 7, Figure 2: Please add to the caption that (b) plots the limiting spectral density of the kernel matrix

- pg 8, Figure 3: The tick marks in (a) are very small and hard to read

- pg 8, Figure 3: In (b, c), it is hard to see the color gradient. Different textures for each curve would improve this.



**Summary Of The Paper:**

This paper studies nonlinear random feature regression of a linear response in the *anisotropic* setting where both the covariates and parameter have arbitrary covariance matrix. The training error and test error are computed explicitly as the dimension, sample size, and number of features jointly tend to infinity. The proof method relies on linearization and a technique from free probability involving the construction of linear pencils. The asymptotic formulas are used to theoretically describe (i) the beneficial effect of overparameterization on the bias and variance components of the test error and (ii) the number of critical points of the test error as a function of overparameterization. This paper graphs the asymptotic formulas for some specific models to demonstrate parameter-wise and sample-wise multiple descent phenomena.

**Summary Of The Review:**

The topic of this theoretical paper is of interest to the machine learning community, and the contributions given would extend understanding of the generalization behavior of high-dimensional, overparameterized models. My current recommendation to reject is because of the concerns mentioned above about the correctness, rigor, and clarity of the proof of the main theorem, Theorem 3.1.

======================

Update on review (11/29)

Following the discussion and revision, my confidence has improved on the correctness of the results. Further the surrounding literature that was overlooked is now addressed, especially in Appendix A.

This work does have some lack of novelty since versions of Theorem 3.1 were known and multiple descent phenomena and some effects of alignment had been recognized in anisotropic random feature regression. Still the main contribution of this paper is the new phenomenology that it identifies, which to my knowledge include the behavior of the bias and variance under overparameterization and (mis)alignment, steep cliffs in the learning curves and their relation to spectral gaps in the covariance matrix, and that parameter-wise descent is limited to double descent. This in itself is a noteworthy contribution.

This paper would be further strengthened by establishing the equivalence of the formulas here with those of d'Ascoli et al '21 and Loureiro et al '21. This would help compensate for the loss in novelty. Also, while the relationship with d'Ascoli et al '21 seems well explained, not as many details are given about the connections with Loureiro et al '21. I also think that this paper needs to be careful regarding claims that it sets these prior works on rigorous footing because (i) the proof of the linearization invariance in Theorem 3.1 is not contained in this paper,  and (ii) to my understanding, the formulas of d'Ascoli et al '21 and Loureiro et al '21 are rigorous modulo linearization invariance. To summarize, the manuscript would benefit from further revision to clarify its contributions and their place in the literature.

All things considered, I am now slightly leaning toward acceptance.

---

> ### Author Response · Authors · 2021-11-17
> **Response 1 to Reviewer ot31 [1/3]**
>
> We thank the reviewer for the time spent providing such a detailed review. The comments have significantly improved the manuscript. We address the reviewer’s main technical feedback point-by-point below.
>
> At a high level though, we believe the reviewer may have misinterpreted the contributions of this paper. Theorem 3.1 is not our main contribution. It only constitutes one of the five main contributions that we list in Sec 1.1. The other four contributions relate to insights derived from the theorem, including: the benefit of overparameterization; a definition of and the effect of weight-data alignment; an analysis of anisotropy and its connection to multiple descent; and the existence of steep cliffs in the learning curves induced by phase transitions in the kernel spectrum. In our view, these insights are nontrivial and valuable to the community, irrespective of the provenance or novelty of the main theorem, or the techniques used to derive it.
>
> Moreover, we believe the reviewer’s concerns regarding the rigor of the proof of the main theorem are misplaced: the main technical elements of the proof (including the linearization for anisotropic covariates) are in fact provided elsewhere (Anonymous, 2021). What is presented here are just the nontrivial components of the calculation, which is a straightforward extension of the calculation provided in (Anonymous, 2021).
> We recognize that the reviewer does not have access to the proof details in (Anonymous, 2021). However, as reviewer 3RfG affirms and as we discuss in more detail below, given prior public work, it is not surprising that the linearization continues to hold for anisotropic covariates. Given this fact, the highly detailed calculations provided (and now improved!) in the supplement, as well as the excellent agreement that we observe between predictions from Thm 3.1 and finite-size simulations (see Fig 1 and the figure linked below), we hope the reviewer does not maintain doubts regarding the correctness of the results. And if the correctness of the Theorem is not in doubt, then we would kindly ask the reviewer to consider reassessing our work based on its main contributions, as highlighted in Sec 1.1.
>
> **Concerns:**
> 1. *I am mainly concerned with the rigor and clarity of the derivation of the main theorem, Theorem 3.1. Please see the section "Proof of Theorem 3.1" below.*
> &nbsp;
> See point by point responses below
> &nbsp;
> 2. *To my understanding, Figures 1--3 are generated by plotting the asymptotic formulae from Theorem 3.1. It would strengthen the paper to confirm these plots with simulations on synthetic data.*
> &nbsp;
> Our revised manuscript includes full simulations in Figure 1 (d,e) which closely match the predictions of our analytical formulas. We are also considering adding a [new figure](https://ibb.co/bLGHKTW) to the supplement with additional simulations showing strong match to theory under a variety of parameter values (traces show theory; crosses show empirical values).
> &nbsp;
> &nbsp;
>
> **Proof of Theorem 3.1**
>
> *My points below mostly concern Section F to the end of Section F.4.1 in the Supplement. The proof happens in two steps: (i) linearization of random features and (ii) application of free-probability theory to compute the trace of some complicated matrices.*
>
> *(i) Linearization: This paper claims without proof that the test error is unchanged by an appropriate linearization of the non-linear random features. Of the given references on pg.13, I could only find a proof of the invariance of test error under linearization in Mei--Montanari '19, which only studies the case of random spherical (that is, isotropic) covariates. This requires justification in the anisotropic case.*
>
> The reviewer is correct that we rely on the invariance of the test error under linearization of the random feature matrix for anisotropic covariates. We do not prove this result because it is proved elsewhere (Anonymous, 2021). Since we acknowledge that the reviewer cannot consult this reference, we give two perspectives that should hopefully persuade the reviewer of the correctness of the linearization and our results.
> As brought to our attention by reviewer 3RfG, a number of recent works have investigated the linearization for nontrivial learning problems, with some even providing compelling numerical evidence supporting its correctness in the situation we study here (Goldt et. al, 2020; Loureiro et al., 2021).
>
> Additionally, in the context of the random feature kernel, the trace of its inverse, and the trace of its inverse multiplied by the sample covariance matrix, Adlam et. al (2019) prove the correctness of the linearization for anisotropic covariates. That this analysis can be extended to cover the terms in the test error should not be too surprising.
>
> *(ii) Free probability: Most of Section F relies on computations of various traces of matrix products involving covariance matrices and iid Gaussian matrices.*

---

> ### Author Response · Authors · 2021-11-17
> **Response 1 to Reviewer ot31 [2/3]**
>
> *Major issues with (ii)*
> 1. *pg 13: There is no explanation of the construction of the $Q^{K^{-1}}$, which I believe is the linear pencil. A short, precise description of the linear pencil methodology would improve the paper since it is crucial to the proof.*
> &nbsp;
> Agreed. A sentence explaining the linear pencil method and the construction of this particular block matrix has been added to the supplement.
> &nbsp;
> 2. *pg 17: "We aim to compute the asymptotic values..." I am concerned about the looseness in taking limits described here. How can the limit in $n_0,n_1,m$ be taken in two stages? Once these go to infinity, there should be no dependence of any expressions on $n_0,n_1,m$. It is also not clear at what point limits are taken in the computation in Section F.4.1.*
> &nbsp;
> We thank Reviewer ot31 for bringing this point to our attention. We have updated the manuscript with a revised linear pencil that streamlines the presentation regarding the limits. In particular, by choosing the linearization matrix $Q^{K^{-1}}$ such that each block of its inverse has an $O(1)$ normalized trace in the limit, each entry of $G^{K^{-1}}$ is manifestly $O(1)$, and there is no longer any dependence on $n_0,n_1,m$ in the self-consistent equations (S209)-(S228). We also point out explicitly where the $n_0,n_1,m\to\infty$ limit is taken. We believe this revision should address the reviewer’s concerns.
> &nbsp;
> 3. *pg 18: Formula (S200) is crucial. Please give a reference to the precise section and theorem in Mingo--Speicher '17.*
> &nbsp;
> Done. Thank you for the suggestion.
> &nbsp;
> 4. *pg 19: The R-transforms in (S202)--(S208) depend on indices of $G^{K^{-1}}$ that go out of range. For example there is a (3,15) entry in (S203).*
> &nbsp;
> This has been corrected. (Originally these components were meant to refer to the hermitianization $\bar{G}$, which is double the size of $G$. We now express all components equivalently in terms of $G$.)
> &nbsp;
> 5. *pg 19: The details of R-transform computations are omitted. I find this acceptable, but please cite the specific chapter in Mingo--Speicher that describes the technique for computing them.*
> &nbsp;
> Done.
> &nbsp;
> 6. *pg 19: How is (S200) used to derive equations (S209)--(S228)? I would appreciate if in the discussion period the authors can explain how to derive (S209) from (S200) and the various R-transforms. In particular, I would like to understand the details of why the expectation over $\mu$ shows up and what happened to all of the random matrices.*
> &nbsp;
> We thank Reviewer ot31 for bringing these presentational issues to our attention. The updated manuscript now includes the following clarification: By eq S200 we have already eliminated all reference to random matrices. Specifically, $\bar{G}^{K^{-1}}$ and $\bar{R}^{K^{-1}}_{W,X,\Theta}(\bar{G}^{K^{-1}})$ are both scalar-entried 18x18 matrices. One then expands the inverse on the right-hand side of S200 to obtain an 18x18 block matrix whose blocks involve various rational functions of $\Sigma,\Sigma_\beta$ and the scalar entries of $G, R$. Finally one computes the normalized traces of these blocks, giving scalar values and eliminating all reference to random matrices. In the limit, traces of rational functions of $\Sigma,\Sigma_\beta$ tend toward expectations of the corresponding rational functions over the LJSD $\mu$.
> &nbsp;
> 7. *pg 19: The appearance of μ in (S209)--(S228) suggests that limits with respect to $n_0,n_1,m$ have been taken. So how is it that the terms $n_1,n_0,m$ still appear in these expressions?*
> &nbsp;
> This issue is addressed in the second point above.
> &nbsp;
> 8. *pg 22: The calculation in Section F.4.4 involves the system (S312)--(S364) consisting of ~50 equations that spans 3 pages. It is hard to verify the end result. Are all of these equations necessary for the derivation? If a simpler explanation isn't possible, I recommend giving a computer-assisted proof citing Mathematica or other algebra software.*
> &nbsp;
> While we agree that solving such a large system of equations is tedious to complete by hand, it is not so large that it is infeasible to do so, especially given that many of the equations trivialize and decouple from the system. In our view, the accessibility afforded by maintaining a paper-based derivation outweighs the potential benefits of providing a computer-aided proof.

---

> ### Author Response · Authors · 2021-11-17
> **Response 1 to Reviewer ot31 [3/3]**
>
> *Minor issues with (ii)*
> 1. *pg 13: $K$ appears to be regularized in (S137), but later $\hat{K}$ is introduced that has the same formula. Should $K$ be non-regularized in Section F?*
> &nbsp;
> $K = \frac{1}{n_1}F^\top F + \gamma I_m$  is $m\times m$ while $\hat{K}=\frac{1}{n_1}FF^\top + \gamma I_{n_1}$ is $n_1\times n_1$ and is built from the other Gram matrix. We have added a note calling attention to this difference.
> &nbsp;
> 2. pg 13: $\theta$ is not defined and seems to sometimes be written as $\Theta$ or $\Theta_F$.
> &nbsp;
> We have added a sentence indicating that $\theta$ on page 13 is a standard normal vector, and that the training features are linearized via $F\to F^{lin}=\frac{\sqrt{\rho}}{\sqrt{n_0}} W X + \sqrt{\eta - \zeta} \Theta$. To streamline the notation, we have removed all mentions of $\Theta_F$.
> &nbsp;
> 3. *pg 18: "Here the normalized trace... acts on the space of 18x18 matrices." Should it be 9x9 matrices?*
> &nbsp;
> The matrices are 18x18 rather than 9x9 because of the hermitianizations which double the matrix dimensions in (S196) and (S199). The updated text should make this clearer.
> &nbsp;
> 4. *For readability, I suggest re-organizing lists of equations with indices listed in lexicographic order ( for example in (S172)--(S194) ).*
> &nbsp;
> Lists of equations are now either row- or column-major ordered (in certain cases we first show an ordered subset of equations which close, followed by a list of remaining components that are defined in terms of previous ones.)
> &nbsp;
> 5. *pg 19: The terms in (S209, S210) have no expectation over μ, but the remaining ones do. Is this a typo?*
> &nbsp;
> These terms do not have an expectation over $\mu$ because the corresponding block in the inverse matrix on the rhs of (S200) does not involve $\Sigma,\Sigma_\beta$ - only the scalars entries of $\bar{G}^{K^{-1}}$ and $\bar{R}^{K^{-1}}_{W,X,\Theta}(\bar{G}^{K^{-1}})$.
>
> **Minor issues/typos in the main text:**
> &nbsp;
> 1. *pg 3, Eq (5, 6): The expectation over $\beta$ appears and then disappears; for consistency please make a choice whether or not to include the expectation over $\beta$ in the error metric.*
> &nbsp;
> The expectation over $\beta$ now only appears in the first equality of eqs. (5, 6) for thoroughness, and is thereafter suppressed for clarity. This is noted in the text immediately following eqs. (5, 6).
> &nbsp;
> 2. *pg 6, Figure 1: What do the blue dashed/solid lines indicate in (a)? What do the gold dashed/solid lines indicate in (b, c)? Are these necessary?*
> &nbsp;
> These lines indicate the locations of the horizontal and vertical slices shown in (d,e,f). We’ve updated the caption to indicate this.
> &nbsp;
> 3. *pg 8, Figure 3: In (b, c), it is hard to see the color gradient. Different textures for each curve would improve this.*
> &nbsp;
> We’ve increased the lineweight which should make it easier to see the color gradient.
> &nbsp;
> 4. We thank Reviewer ot31 for pointing out other small problem spots which have been fixed.

---

> > ### Comment · Reviewer_ot31 · 2021-11-24
> > **Thanks**
> >
> > Thanks very much to the authors for the diligent, detailed reply and revision. With the improved explanations and commentary, it is easier to follow the free probability techniques in G.4.1. I'm also grateful to the authors for acknowledging the use of a computer algebra system to compute the linear pencil. Is computer algebra used also in parts of G.4.4? If so, then mentioning this would further demystify the calculations in that subsection.
> >
> > Regarding the linearization, the discussion with reviewer 3RfG helped a lot as well as the careful literature review and comparison in the Section 1.2 and Appendix A. I do find it unfortunate though that the rigorous proof of the linearization invariance required for this work cannot be accessed by us.
> >
> > I accept that Theorem 3.1 is not the main contribution of this work, but since the observed phenomenology depends on these formulas, I think their correctness is quite crucial. That being said, my confidence has improved regarding the correctness of the free probability calculations, and given the surrounding literature/conjectures, it seems that the linearization invariance likely holds as well (though it is not carefully justified in this work).
> >
> > Thanks also for including the empirical simulations. They seem to match well with the formulas. Could you say more explicitly what is being plotted in the crosses? I guess it is an average of $(y(x) - \hat y(x))^2$ over several trials with m = 4000 samples, but I wasn't clear on the details.
> >
> > In the remainder of the discussion period, I will update my review to take into account the authors' responses and revisions.

---

> > > ### Author Response · Authors · 2021-11-24
> > > **Response 2 to Reviewer ot31 [1/1]**
> > >
> > > *Thanks very much to the authors for the diligent, detailed reply and revision. With the improved explanations and commentary, it is easier to follow the free probability techniques in G.4.1. I'm also grateful to the authors for acknowledging the use of a computer algebra system to compute the linear pencil. Is computer algebra used also in parts of G.4.4? If so, then mentioning this would further demystify the calculations in that subsection.*
> > >
> > > Yes, in the final version we will include a statement similar to the one in G.4.1 to indicate the use of computer algebra to construct the linear pencils in section G.4.4 as well. Moreover, we will indicate that computer algebra was also used to obtain (S370)-(S374).
> > >
> > > *Regarding the linearization, the discussion with reviewer 3RfG helped a lot as well as the careful literature review and comparison in the Section 1.2 and Appendix A. I do find it unfortunate though that the rigorous proof of the linearization invariance required for this work cannot be accessed by us.*
> > >
> > > We recognize that the inaccessibility of this reference makes evaluating some aspects of the current manuscript difficult. Of course, we do not ask the reviewer to accept the conclusions of an anonymous reference without the opportunity to review it. We would be happy to add comments to the manuscript that emphasize our reliance on the linearization, and that a rigorous proof of the linearization was not publicly available at the time of writing. If those caveats seem appropriate, we would encourage the reviewer to evaluate the manuscript through that lens.
> > >
> > > *I accept that Theorem 3.1 is not the main contribution of this work, but since the observed phenomenology depends on these formulas, I think their correctness is quite crucial. That being said, my confidence has improved regarding the correctness of the free probability calculations, and given the surrounding literature/conjectures, it seems that the linearization invariance likely holds as well (though it is not carefully justified in this work).*
> > >
> > > Yes, we agree that the correctness of the formulas is paramount. For one additional datapoint -- as we mentioned to reviewer 3RfG, it appears our formula for the total test error agrees with that of d’Ascoli et al. (2021), at least when $\mu$ is a finite sum of delta masses. We are considering adding a section to the appendix showing this to be the case.
> > >
> > > *Thanks also for including the empirical simulations. They seem to match well with the formulas. Could you say more explicitly what is being plotted in the crosses? I guess it is an average of $(y(x)−\hat{y}(x))^2$ over several trials with $m = 4000$ samples, but I wasn't clear on the details.*
> > >
> > > Yes, in all panels where simulation results are plotted (ie. Fig. 1d,e and the additional figure linked in our first response), traces indicate theoretical predictions and crosses show results from simulations. In Fig. 1d, for example, $n_0/m=\phi=10/9$ (this is fixed for the whole figure) and the horizontal axis gives the value of $n_1/m=\phi/\psi$ for each cross. Choosing $m=4000$ then determines the sizes $n_0,n_1$ used for simulation. For each cross, we repeatedly generated random inputs, features and outputs according to the model in eqs. (1), (2) and numerically computed the prediction $\hat{y}$ on a new test example using eq. (4). Blue crosses representing total error indicate the average value of $(y-\hat{y})^2$ over all repetitions. A similar procedure was used to estimate the bias. The variance was computed by subtracting the bias from the total error.
> > >
> > > We thank the reviewer for bringing this issue to our attention. A more detailed explanation of how simulations were carried out will appear in the final version.

---

> > > > ### Comment · Reviewer_ot31 · 2021-11-24
> > > > **Re: Response 2 to Reviewer ot31**
> > > >
> > > > Thanks for the message. Regarding your note,
> > > >
> > > > *We recognize that the inaccessibility of this reference...*
> > > >
> > > > I appreciate the thought into this. As long as it is clear in the main text that Theorem 3.1 relies on either a proof of the linearization result in (Anon, 2021) or conjectures from the neighboring literature, then I will evaluate the paper based on the conclusions that it makes from there.

---

### Official Review · Reviewer_6qzE · 2021-11-03

**Correctness:** 3
**Technical Novelty And Significance:** 2
**Empirical Novelty And Significance:** 3
**Recommendation:** 6
**Confidence:** 3

**Main Review:**

In general, the paper is written clearly with enough technical details to understand the arguments. The motivation to study data with anisotropic covariances is good. The findings on the error curve structures and the correspondence between phase transition and induced kernel matrix spectrum are interesting. In all, I think the significance and contribution of this paper meet the bar of ICLR.

However, one thing that may weaken the significance of this work is that, as a theoretical paper, the proof techniques of the main results is almost the same as previous works studying the case of identity covariance. The only difference is the changing in the covariance, which only changes some calculations. Thus, from the technical perspective, the paper does not provide much new insight.

Questions and detailed comments:

1. What is the motivation to consider random model weights and what is a good practical implication of this setting and related results (e.g., the covariance alignment)? I notice that the analysis in Liao et. al NeurIPS 2020 also does not require that the data has identity covariance matrix, and do not assume random weights either. I think that setting could be more practical. How does the results in this paper compare with theirs?

2. The analytical figures are plotted using the data model described in Section 2.3. Does the observations hold for more general data distributions?

3. What is s in equation (12)? In Figure 3 c), what is the optimal regularization parameter lambda? Seems that it is not described.

4. In Figure 2 a), the multiple descent is more obvious for small \omega, which is the linear model. Is that correct? Then why is it stated as an observation for random feature regression?

Some plots in Figure 2 and 3 needs to be better clarified. I'm a little bit confused about some plots and find it hard to get what the curves represent. There is a typo in the caption of Figure 2, 3rd line.

**Summary Of The Paper:**

The paper studies random feature regression in high dimensional setting, when the data and model weights are assumed to be normal with anisotropic covariances. By an error decomposition argument, the exact formulas for the bias, variance and total error are provided. The results motivates the discussion on the multiple descent effect of the sample-wise test error curve, and the authors also prove that the error against overparametirazation only exhibits double descent. The phase transition is associated with the spectral structure of the approximated kernel matrix. Also, the error is improved when the data covariance aligns more with the model weight covariance.

**Summary Of The Review:**

In the sub-field of high-dimensional random feature regression, the paper present good contribution and results to consider the anisotropic data distribution, though the proof techniques mostly follow from previous works. Therefore, I think the paper has its pros and cons, depending on the "weight" of technical novelty. In all, I feel the paper is above the borderline of acceptance.

---

> ### Author Response · Authors · 2021-11-17
> **Response 1 to Reviewer 6qzE [1/2]**
>
> **Main Review:**
>
> *In general, the paper is written clearly with enough technical details to understand the arguments. The motivation to study data with anisotropic covariances is good. The findings on the error curve structures and the correspondence between phase transition and induced kernel matrix spectrum are interesting. In all, I think the significance and contribution of this paper meet the bar of ICLR.*
>
> *However, one thing that may weaken the significance of this work is that, as a theoretical paper, the proof techniques of the main results is almost the same as previous works studying the case of identity covariance. The only difference is the changing in the covariance, which only changes some calculations. Thus, from the technical perspective, the paper does not provide much new insight.*
>
> The reviewer is absolutely correct that this paper does not introduce new techniques for obtaining the results. However, the calculations themselves are quite nontrivial, and even applying known methods to derive the explicit results is a substantial technical contribution. Nevertheless, the resulting asymptotic formulas (Thm 3.1) only constitute one of the five main contributions that we list in Sec 1.1. The other four contributions relate to insights derived from these results, including: the benefit of overparameterization; a definition of and the effect of weight-data alignment; an analysis of anisotropy and its connection to multiple descent; and the existence of steep cliffs in the learning curves induced by phase transitions in the kernel spectrum. In our view, these insights are nontrivial and valuable to the community, regardless of whether the techniques used to derive the main theorem are novel.
>
> **Questions and detailed comments:**
> 1. *What is the motivation to consider random model weights and what is a good practical implication of this setting and related results (e.g., the covariance alignment)? I notice that the analysis in Liao et. al NeurIPS 2020 also does not require that the data has identity covariance matrix, and do not assume random weights either. I think that setting could be more practical. How does the results in this paper compare with theirs?*
> &nbsp;
> We first note that the assumption of random weights is actually quite common in the high-dimensional regression setting, see e.g. Dobriban and Wager (2018). We invoked this assumption for technical convenience and to mirror prior results that make this kind of assumption, but there is no major hurdle preventing an extension to deterministic $\beta$. We believe our current formulas extend to the case of deterministic $\beta$ upon replacement of $\Sigma_\beta$ with $\beta\beta^\top$, so long as the activation function is centered (i.e. it has zero Gaussian mean). From this perspective, the overlap coefficients in Eq. (8) are $q_i  = (v_i^\top \beta)^2$, i.e. they are simply the components of the weight vector in the eigenbasis of the data covariance. Interpreting the results in this light should give a sense for the practical implications of our setup.
> &nbsp;
> 2. *While there are many valuable aspects of the analysis of Liao et. al, a significant drawback is the restriction to random Fourier features, meaning they cannot change the nonlinearity and therefore cannot examine the effects of the nonlinearity itself. As can be seen in e.g. Fig 2, many of the phenomena we study require varying the nonlinearity, and are therefore uniquely accessible from our setting. The analytical figures are plotted using the data model described in Section 2.3. Does the observations hold for more general data distributions?*
> &nbsp;
> Yes. We use the simple d-scale model of Section 2.3 for the purposes of illustration, but all our results hold for the general setup described in Sections 2.1 and 2.2. The sample-wise multiple descent illustrated in Fig. 2 is a nonlinear generalization of the multiple descent curves reported in (We & Xu 2020; Mel & Ganguli 2021) and holds more generally, so long as the data distributions have well-separated scales. We show in Section E.1 of the supplement that cliffs in the error curves as a function of overparameterization (illustrated in Fig. 3) are the consequence of large spectral gaps.
> &nbsp;
> 3. *What is $s$ in equation (12)? In Figure 3 c), what is the optimal regularization parameter lambda? Seems that it is not described.*
> &nbsp;
> $s$ is defined as the asymptotic covariance scale $s= \lim_{n_0 \to \infty} \bar{\mathrm{tr}}(\Sigma)$ in 2.2. We agree it would be helpful to refresh the reader at this point in the text. The definition of s is now reproduced immediately after eq 12. A note explaining how optimal regularization curves were obtained now appears in the caption of Fig. 3.
> &nbsp;

---

> ### Author Response · Authors · 2021-11-17
> **Response 1 to Reviewer 6qzE [2/2]**
>
> 4. *In Figure 2 a), the multiple descent is more obvious for small $\omega$, which is the linear model. Is that correct? Then why is it stated as an observation for random feature regression?*
> &nbsp;
> As observed by Bartlett et al. (2021) and others, the nonlinearity in random feature regression can act as an effective regularizer through the constant $\omega$. As such, it may not be surprising that peaks in the error curves are attenuated for large $\omega$ and are more pronounced for small $\omega$. Moreover, it is worth highlighting two important aspects of the nonlinearity constant $\omega$. First, in the overparameterized regime, the rank of the random feature matrix is discontinuous in $\omega$ at $\omega=0$. As such, even very small (but nonzero) values of $\omega$ can reveal phenomena that are truly unique to nonlinear random feature regression. Second, even when $\omega=0$, the (linear) random feature model is distinct from basic linear regression, owing to the random weight matrix $W$ because the extra randomness induced by $W$ has a nontrivial effect on the variance.
> &nbsp;
> To further elaborate on the effect of $\omega$, note that, as Figure 2(a) shows, multiple descent is a general feature of random feature regression and can be observed for truly nonlinear activation functions (ie. $\omega>0$). Even for modest values of $\omega$ (ie. stronger nonlinearities) where only one peak is visible, Figure 2(a) shows that it wouldn’t be accurate to refer to this as ordinary sample-wise double descent. There is only one peak, but it is a different peak than that of double descent: it occurs at a different critical value of m (ie. $m/n_0=1/3$, rather than $m/n_0=1$), and indeed is part of a distinct feature of the error surface than the classical peak observed for the linear model at $m=n_0$.

---

### Author Response · Authors · 2021-11-30
**Response to feedback on revision**

We thank the reviewers for their additional feedback and suggestions. As a small update, we wanted to note that we have now mapped out the correspondence between our quantities and the observables defined by d’Ascoli et al. ‘21; Loureiro et al. ‘21; etc. The final manuscript will include a detailed derivation of this equivalence in the case of square loss. We look forward to exploring additional directions that these connections facilitate, such as the effect of alignment for other loss functions, as reviewer 3RfG mentions.

Additionally, the final version will include an explicit discussion of the linearization assumption, highlighting how and when we rely on the results of Anonymous, 2021. It will also expand the discussion of Loureiro et al. ‘21, which we have now more fully digested, and which we agree is a key prior work. In particular, it will emphasize the rigor of their results, modulo the linearization.

---

### Decision · Program_Chairs · 2022-01-20

**Decision:**

Accept (Poster)

**Comment:**

This paper extends recent and very active literature on analyzing learning algorithms in the simplified setting of Gaussian data and model weights, with the main generalization being to allow for non-isotropic covariance matrices.  The main technical results seem to be correct and slightly novel, though reviewers feel they are not innovative or unexpected enough to stand on their own.  However, the main contributions of the paper are then to interpret these results to give phenomenological results (regarding double descent, etc.), and reviewers were unanimously happy with these. In the end, all reviewers were positive about the paper.

The largest reviewer criticisms of the paper were technical issues (ot31) and lack of context of recent literature (3RfG). Both of these concerns were mostly addressed by the revision/rebuttal. The reviewers still had specific comments on improvement, but found no major faults.